# What can a Single Attention Layer Learn?
# A Study Through the Random Features Lens

**Hengyu Fu**[*]
Peking University
2100010881@stu.pku.edu.cn

**Tianyu Guo**[*]
UC Berkeley
tianyu_guo@berkeley.edu

**Yu Bai**
Salesforce AI Research
yu.bai@salesforce.com

**Song Mei**
UC Berkeley
songmei@berkeley.edu

## Abstract

Attention layers—which map a sequence of inputs to a sequence of outputs—are core building blocks of the Transformer architecture which has achieved significant breakthroughs in modern artificial intelligence. This paper presents a rigorous theoretical study on the learning and generalization of a single multi-head attention layer, with activation function replaced by ReLU, which have recently shown comparable performance with the original Softmax activation. We consider the *random feature* setting where the attention layer has a large number of heads, with randomly sampled frozen query and key matrices, and trainable value matrices. We show that such a random-feature attention layer can express a broad class of target functions that are permutation invariant to the key vectors. We further provide quantitative excess risk bounds for learning these target functions from finite samples, using random feature attention with finitely many heads.

Our results feature several implications unique to the attention structure compared with existing random features theory for neural networks, such as (1) Advantages over standard fully connected random-feature models; (2) Concrete and natural classes of functions that can be learned efficiently by a random-feature attention layer. Additionally, we show that the sampling distribution of the *query-key* matrix (the product of the query and key matrix) matters—A *biased* Gaussian random matrix results in better sample complexities over the standard zero-mean counterpart for learning certain natural target functions.Experiments on simulated data corroborate our theoretical findings and further illustrate the interplay between the sample size and the complexity of the target function.

## 1 Introduction

The transformer architecture [86] has achieved remarkable recent successes in many areas of artificial intelligence (AI) such as vision, language, speech, graph processing, reinforcement learning, and more recently general AI capabilities [29–31, 19, 73, 101, 22, 77, 65, 20]. A central building block in transformers is the *attention layers* [10]—sequence-to-sequence mappings that allow each token within the input sequence to "attend to" other tokens that are most relevant to the present token, and produce outputs based on those tokens. Attention layers implement this mechanism in a compact way that allows them to handle sequences of arbitrary length using a fixed set of parameters, a crucial reason behind their success in handling long input sequences.

---

[*]Equal contributions.

37th Conference on Neural Information Processing Systems (NeurIPS 2023).

Despite its wide applicability, the theoretical properties of attention layers are less well understood. While multi-layer attention networks (transformers) have been shown to be universal approximators for certain classes of functions, such as equivariant sequence-to-sequence functions [102], their results only focus on the expressive power and do not account for learning from finite samples. Another line of work derives generalization bounds for learning with *multi-layer* transformers in terms of the number of layers, heads, and weight norms [91, 35], yet the results are either instantiated on specific target functions such as sparse boolean functions [35], or generic but arguably elusive function classes such as Turing machines [91]. Understandings about the more basic building block—a *single* attention layer— remain largely open. This is in stark contrast with the situation for fully connected neural networks, where there is by now a decent understanding of the learning and generalization in the important basic case of *two-layer* neural networks on generic and natural function classes (e.g., [8, 51, 7, 40] and many other results along the line). This motivates the following open question:

*What function classes can be learned by a **single attention layer** with benign sample complexities?*

This work makes progress on this problem by studying the learning and generalization with a single attention layer in the *random feature* setting [74, 76, 28, 100], in which the query and key matrices are frozen at their random initialization, and the value matrices remain to be learnable parameters. Motivated by the attention structure in practical architectures, we consider attention layers that take in a single *query token* $\mathbf{x}_0 \in \mathbb{R}^d$ and $N$ *key tokens* $\{\mathbf{x}_i\}_{i\in[N]}$ as the input, and produce a scalar-valued output—A simplified setting capturing the essence (the interaction between the query and keys) of attention models. We study the sample complexity of learning certain target functions (of $\mathbf{x}_{0:N}$) using an attention layer with a large but finite number of heads, and finitely many samples.

Our contributions are summarized as follows.

- We show that a Random Feature Attention layer (the `RFA` model) with a sufficiently large number of heads can express a broad class of target functions that are averages over a generic function of two tokens, which are in particular permutation invariant with respect to the key tokens (Section 3.1). We give several natural examples of target functions in this class (Section 3.3) with concrete bounds on the number of heads and weight norms.

- We derive an $\widetilde{\mathcal{O}}(\sqrt{B(f_\star)/n})$ excess risk bound for learning with the `RFA` model with sufficiently many heads, where $B(f_\star)$ is an inherent complexity measure of the target function $f_\star$ and $n$ is the sample size (Section 3.2). When instantiated on the aforementioned examples, the bounds only depend on the input dimension and not the number of key tokens, improving over a naive two-layer random feature neural network model (`RFMLP`). Such improvement is expected due to the permutation invariance structure of target functions, aligning with the attention mechanism.

- Towards moving beyond standard random feature settings, we study a *biased* `RFA` model where the query-key matrices (product of transposed query matrices and key matrices) are drawn from a distribution with a non-zero mean (more precisely the identity matrix as the mean), motivated by a similar observation on learned attention layers in practice. We show that this model achieves provably superior sample complexity than the standard zero-mean `RFA` for learning certain functions of the *correlations* between the query token and key tokens (Section 4).

- Experiments on simulated data verify our theoretical findings in realistic settings of learning from finite samples using a `RFA` layer with a mild number of heads, and characterize the interplay between the complexity of the target function and the sample size (Section 5).

## 1.1 Related work

**Transformers** The Transformer architecture, initially proposed by [86], brought about a revolutionary change in natural language processing and has been widely adopted in large language models such as GPT and BERT [72, 29, 19]. At the core of transformers lie the *attention layers*, which were originally introduced as neural network modules for machine translation tasks [10, 49, 68].

A recent line of work investigated the capabilities of transformers by viewing transformers to be function approximators [102], computational models [91, 70, 99, 15, 56], or algorithms [82], and using transformers to perform synthetic reasoning tasks [103]. Among these works, the closest to our work is [102], which shows that multi-layer transformers can approximate any permutation-equivariant sequence-to-sequence function, and any function if positional encodings are added. Our

paper instead uses a single attention layer to approximate sequence-to-scalar functions and focuses on the generalization property with quantitative bounds.

In terms of generalization properties of transformers, [35] analyzed the generalization bound of a single attention network through the Rademacher complexity and showed that a single self-attention head could efficiently represent a sparse function of the input sequence. Besides, several works studied the sample complexities of vision transformers [48, 50], and prompt-tuning using attention [67] with special target function classes. Our paper also studies the generalization bound of a single attention network, but from the different perspective of kernel methods, and for a more general class of target functions. The kernel limit of transformers was derived in [43, 98], which shows that multi-head attention architectures behave as Gaussian processes as the number of heads tends to infinity. However, they do not study the representation power of the limiting kernel.

Besides approximation and generalization capabilities, recent work also studied the limitations [42, 14], internal working mechanisms [36, 81, 94, 64], and in-context learning capabilities [19, 96, 37, 88, 3, 26, 41, 52] of Transformer models.

**Theory of random features and neural tangent kernels** A recent line of work [28, 51, 33, 32, 6, 7, 104, 66, 25] studied the training dynamics of overparametrized neural networks under certain random initialization, and showed that it converges to a kernel estimator, which corresponds to the "neural tangent kernel" (NTK). These works suggested that one could use kernel or random-feature models [74] to study the properties of deep neural networks.

For NTK of MLPs and their corresponding random-feature models, there is a vast number of literature that studies their approximation power [12, 71, 8], as well as their generalization properties [13, 21, 93, 92, 54, 55, 76, 79, 100, 57, 60, 40, 39]. More recently, a line of work studies the NTK beyond MLPs, including convolution networks [53, 17, 59, 62, 18, 16], residual networks [45, 83, 4], graph networks [97, 47], and transformers [43, 98].

Although the kernel approach is a powerful tool for studying neural networks, it received criticism since it does not capture the feature learning of neural networks. Going beyond the kernel regime, a series of works used the mean field method to establish the evolution of the network parameters via a Wasserstein gradient flow [58, 9, 24, 78]. Several other mechanisms have been proven to obtain superior results over the NTK, including the Quadratic NTK [5, 11, 23, 63], regularization [90], Neural Tangent Hierarchy [34, 44], representation learning [27], and staircase-like mechanisms [1, 2].

## 2 Preliminaries

We consider a sequence of $N+1$ input tokens $\mathbf{x}_{0:N} = (\mathbf{x}_0, \{\mathbf{x}_i\}_{i \in [N]}) \in \mathcal{X} = (\mathbb{R}^d)^{N+1}$, where each $\{\mathbf{x}_i\}_{i \in [N]} \subseteq \mathbb{R}^d$ represents a sequence of *key vectors*, and $\mathbf{x}_0$ represents the *query vector*. This model simplifies standard self-attention, which maps $N$ input tokens to $N$ output tokens, where the output $i$ only uses input $i$ as the query token and all of $[N]$ as key tokens. Results obtained in this model can be directly mapped back to full self-attention, by simply concatenating $N$ outputs generated by our model with $\mathbf{x}_0$ ranging over $\{\mathbf{x}_i\}_{i \in [N]}$. In addition, throughout the paper, we consider scalar-valued attention models, which take the sequence $\mathbf{x}_{0:N}$ as input and give a scalar output in $\mathbb{R}$.

**Attention layer** We consider a scalar-valued, $M$-head, multiplicative attention layer that takes $\mathbf{x}_{0:N} = (\mathbf{x}_0, \{\mathbf{x}_i\}_{i \in [N]})$ as the input. The attention layer first applies affine (linear with bias) transformations to the input vectors to obtain {query, keys, values} at each head $m \in [M]$:

$$\mathbf{q}_{m,0} = \mathbf{Q}_m[\mathbf{x}_0; 1] =: \mathbf{Q}_m \widetilde{\mathbf{x}}_0 \in \mathbb{R}^d, \qquad \mathbf{k}_{m,i} = \mathbf{K}_m[\mathbf{x}_i; 1] =: \mathbf{K}_m \widetilde{\mathbf{x}}_i \in \mathbb{R}^d,$$
$$v_{m,i} = \mathbf{v}_m^\top[\mathbf{x}_i; 1] = \mathbf{v}_m^\top \widetilde{\mathbf{x}}_i \in \mathbb{R}, \quad i \in [N], \tag{1}$$

where $\mathbf{Q}_m, \mathbf{K}_m \in \mathbb{R}^{(d+1) \times d}$, $\mathbf{v}_m \in \mathbb{R}^{d+1}$ are the parameters of the attention layer, and $\widetilde{\mathbf{x}}_i := [\mathbf{x}_i; 1]$ for a more compact display. Then, it computes the output value by an attention mechanism

$$f(\mathbf{x}_{0:N}) = \sum_{m=1}^{M} \frac{1}{N} \sum_{i=1}^{N} f_{m,i}(\mathbf{x}_0, \mathbf{x}_i), \qquad f_{m,i}(\mathbf{x}_0, \mathbf{x}_i) = \sigma(\langle \mathbf{q}_{m,0}, \mathbf{k}_{m,i} \rangle) \cdot v_{m,i} \in \mathbb{R}. \tag{2}$$

Above, $\sigma : \mathbb{R} \to \mathbb{R}$ is an activation function applied entry-wisely to each attention score $\langle \mathbf{q}_{m,0}, \mathbf{k}_{m,i} \rangle$. We choose $\sigma$ to be the ReLU activation $\sigma(t) = \max\{t, 0\}$ throughout this paper. Notice that

this choice of the attention non-linearity is different from standard transformers [86] with softmax-attention. We remark that we choose to study the (normalized) ReLU attention for theoretical convenience, and this replacement does not change the essence of the attention mechanism. Such a choice is also recently explored in the literatures such as [80] and [95], which show that transformers with ReLU-attention perform as well as standard softmax-attention transformers in certain NLP and CV tasks. Moreover, our results can extend to other activation functions such as the exponential activation $\sigma(t) = \exp(t)$, which is more similar to the standard Softmax activation. We refer to Section B.5 for a short discussion.

Simplifying the expression, we reparametrize the attention layer (1) and (2) using parameters $\{(\mathbf{W}_m, \mathbf{v}_m)\}_{m \in [M]} \subseteq \mathbb{R}^{(d+1) \times (d+1)} \times \mathbb{R}^{d+1}$:

$$f_{i,m}(\mathbf{x}_{0:N}) = \sigma\Big(\widetilde{\mathbf{x}}_0^\top \mathbf{Q}_m \mathbf{K}_m \widetilde{\mathbf{x}}_i\Big) \cdot \langle \mathbf{v}_m, \widetilde{\mathbf{x}}_i \rangle = \sigma\big(\langle \mathbf{W}_m, \widetilde{\mathbf{x}}_0 \widetilde{\mathbf{x}}_i^\top \rangle\big) \cdot \langle \mathbf{v}_m, \widetilde{\mathbf{x}}_i \rangle. \tag{3}$$

For technical convenience, we assume all input tokens have unit norm throughout the rest of the paper: $\|\mathbf{x}_i\|_2 \equiv 1$ so that $\|\widetilde{\mathbf{x}}_i\|_2 \equiv \sqrt{2}$, for all $i \in \{0\} \cup [N]$.

**Random-feature attention models**  We consider a random-feature version[2] of the multiplicative attention mechanism (3), where the weight matrices $\{\mathbf{W}_m\}_{m \in [M]}$ have i.i.d. Gaussian entries[3]:

$$(\mathbf{W}_m)_{ij} \sim_{\text{iid}} \mathsf{N}(0, 1/4), \quad (m, i, j) \in [M] \times [d+1]^2. \tag{4}$$

The variance is chosen to be $1/4$ without loss of generality: this choice of variance is such that $\langle \mathbf{W}_m, \widetilde{\mathbf{x}}_0 \widetilde{\mathbf{x}}_i^\top \rangle \sim \mathsf{N}(0, 1)$ has a unit variance. The weight matrices $\{\mathbf{W}_m\}_{m \in [M]}$ are then held to be fixed during the entire learning process, whereas the value vectors $\{\mathbf{v}_m\}_{m \in [M]}$ are the learnable parameters. The random-feature attention model with input $\mathbf{x}_{0:N}$ is thus given by

$$f_M^{\mathbf{W}}(\mathbf{x}_{0:N}; \mathbf{V}) = \sum_{m=1}^M \frac{1}{N} \sum_{i=1}^N \sigma\big(\langle \mathbf{W}_m, \widetilde{\mathbf{x}}_0 \widetilde{\mathbf{x}}_i^\top \rangle\big) \langle \mathbf{v}_m, \widetilde{\mathbf{x}}_i \rangle. \tag{5}$$

Notice that random-feature attention model is linear in the parameter $\mathbf{V}$, so training this model with a convex loss function gives a convex optimization problem.

**Additional notation**  For any $\mathbf{x} \in \mathbb{R}^{d_1}$ and $\mathbf{y} \in \mathbb{R}^{d_2}$, let $\mathbf{x} \otimes \mathbf{y} \in \mathbb{R}^{d_1 \times d_2}$ denote their tensor product (outer product), and $\mathbf{x}^{\otimes n} := \mathbf{x} \otimes \cdots \otimes \mathbf{x}$ denote the $n$-fold self tensor product of $\mathbf{x}$. For a tensor $\mathbf{A}$, we use $\|\mathbf{A}\|_{\mathsf{Fr}}$ to denote its Frobenius norm. For a function $f : \mathcal{X} \to \mathbb{R}$, we use $\|f\|_\infty$ to denote its $L^\infty$ norm. We use $\mathcal{O}(\cdot)$ (resp. $\Theta(\cdot)$) for standard Big-O (resp. Big-Theta) relations. We use $\widetilde{\mathcal{O}}(\cdot)$ for hiding the multiplicative terms that are logarithmic in problem parameters, including $(M, d, n, N, \delta^{-1})$. We use $\mathrm{Poly}(p)$ to denote a polynomial of $p$ that is less than $p^C$ for some universal constant $0 < C < \infty$.

## 3 Learning with random-feature attention models

In this section, we study the expressivity and generalization of random-feature attention models. We will consider a broad class of target functions that can be well approximated and is efficiently learnable by random-feature attention models.

### 3.1 Expressivity of random-feature attention

Consider a broad class of permutation invariant[4] target functions $f_\star : \mathcal{X} \to \mathbb{R}$ that takes form

$$f_\star(\mathbf{x}_{0:N}) = \frac{1}{N} \sum_{i=1}^N F(\mathbf{x}_0, \mathbf{x}_i). \tag{6}$$

---

[2]A different and closely related model is the Neural Tangent Kernel [46, 32], which is however similar in essence to the random feature model in terms of the sample complexity of learning, e.g. [40].

[3]Another feasible choice for (4) is to sample the key matrix and the query matrix separately with independent Gaussian entries, which however will produce a mean-zero product matrix similar to (4) in many aspects.

[4]A function $f(\mathbf{x}_0, \mathbf{x}_1, ..., \mathbf{x}_N)$ is permutation invariant (with respect to $\mathbf{x}_{1:N}$) if $f(\mathbf{x}_0, \mathbf{x}_1, ..., \mathbf{x}_N) = f(\mathbf{x}_0, \mathbf{x}_{\sigma(1)}, ..., \mathbf{x}_{\sigma(N)})$ for any permutation $\sigma : [N] \to [N]$. We consider permutation invariant target functions since attention layers can only fit these functions due to the structure of attention models.

Assume that there exists symmetric tensors $\{\mathbf{f}_{rs} \in \mathbb{R}^{d^{r+s}}\}_{r,s \geq 0}$ such that $F : \mathbb{R}^{2d} \to \mathbb{R}$ admits representation

$$F(\mathbf{x}_0, \mathbf{x}_i) = \sum_{r,s \geq 0}^{\infty} \left\langle \mathbf{x}_0^{\otimes r} \otimes \mathbf{x}_i^{\otimes s}, \mathbf{f}_{rs} \right\rangle. \tag{7}$$

Note that such an expression allows $F(\mathbf{x}_0, \mathbf{x}_i)$ to be any general nonlinear function that admits convergent Taylor expansions. In particular, any polynomials of $[\mathbf{x}_0, \mathbf{x}_i]$ (e.g., $\boldsymbol{\beta}^\top \mathbf{x}_0$, $\boldsymbol{\beta}^\top \mathbf{x}_i$, and $\langle \mathbf{x}_0, \mathbf{S} \mathbf{x}_i \rangle$ for some $\boldsymbol{\beta} \in \mathbb{R}^d$ and $\mathbf{S} \in \mathbb{R}^{d^2}$) are within this function class. We will discuss more specific target functions in Section 3.3.

**Theorem 1** (Expressivity of `RFA` model). *Suppose function $f_\star : \mathcal{X} \to \mathbb{R}$ takes form (6). Then for any input distribution $P$ on $\mathcal{X}$, with probability at least $1 - \delta$ (over $\{\mathbf{W}_m\}_{m \in [M]}$ sampled from (4)), there exists an $M$-head `RFA` model (5) with coefficients $\mathbf{V} = \{\mathbf{v}_m\}_{m \in [M]} \subseteq \mathbb{R}^{d+1}$ that approximates $f_\star$ in $L^2(P)$ up to error*

$$\mathbb{E}_{\mathbf{x}_{0:N} \sim P}\left[\left(f_\star(\mathbf{x}_{0:N}) - f_M^{\mathbf{W}}(\mathbf{x}_{0:N}; \mathbf{V})\right)^2\right] \leq \mathcal{O}\left(\frac{(d^2 + \log M)B(f_\star)\delta^{-1}}{M}\right). \tag{8}$$

*In addition, the norms of the weight of this random-feature attention model are bounded as*

$$\sum_{m=1}^{M} \|\mathbf{v}_m\|_2 \leq \mathcal{O}\left(\sqrt{B(f_\star)} + \sqrt{\frac{B(f_\star)\delta^{-1}}{M}}\right), \qquad \sum_{m=1}^{M} \|\mathbf{v}_m\|_2^2 \leq \mathcal{O}\left(\frac{B(f_\star)\delta^{-1}}{M}\right). \tag{9}$$

*Here $B(f_\star)$ is a complexity measure of $f_\star$ defined as*

$$B(f_\star) = \sum_{k=0}^{\infty} C_k \sum_{\max\{r,s\}=k} \|\mathbf{f}_{rs}\|_{\mathsf{Fr}}^2, \qquad C_k = k^{4.5} 4^k \vee 1. \tag{10}$$

*In case where $f_\star$ admits multiple representations of the form (7), $B(f_\star)$ is the infimum of the right-hand-side over all such representations.*

The proof of Theorem 1 is contained in Appendix B.1. Our proof relies on standard analyses of infinite-width random feature model with ReLU-Gaussian kernel, combined with a sampling argument to obtain approximation with finite-width.

This theorem is applicable to general functions with a finite $B(f_\star)$ norm. The $4^k$ scaling of $C_k$ in the summand of equation (10) seemingly confines the target function class to those with exponentially fast decaying $\|\mathbf{f}_{rs}\|_{\mathsf{Fr}}$, which suggests a relatively narrow target function class. However, as we will demonstrate in the forthcoming examples, this class includes a diverse range of functions.

## 3.2 Generalization and sample complexity of learning

Given $n$ samples $\{\mathbf{x}_{0:N}^{(j)}, y_j\}_{j \in [n]} \sim_{\text{iid}} P$, where $\mathbf{x}_{0:N}^{(j)} = \{\mathbf{x}_i^{(j)}\}_{0 \leq i \leq N}$ is the $j$-th token sequence with length $N + 1$, and $y_j$ is the label corresponding to the $i$-th token sequence. Assume that we are given a loss function $\ell(\widehat{y}, y)$ that is 1-Lipschitz in $\widehat{y}$, and $\ell(0, y) \leq 1$ for any $y$. The population risk is then given by $L_D(f) = \mathbb{E}_{(\mathbf{x}_{0:N}, y) \sim P}[\ell(f(\mathbf{x}_{0:N}), y)]$. We consider the empirical risk minimization (ERM) over the `RFA` model (5),

$$\widehat{\mathbf{V}} = \arg\min_{\mathbf{V} \in \mathcal{V}_M} \widehat{L}_D(f_M^{\mathbf{W}}(\cdot; \mathbf{V})), \qquad \widehat{L}_D(f) = \frac{1}{n} \sum_{j=1}^{n} \ell(f(\mathbf{x}_{0:N}^{(j)}), y_j), \tag{11}$$

where the constrained class $\mathcal{V}_M$ is given by

$$\mathcal{V}_M = \left\{ \mathbf{V} = \{\mathbf{v}_m\}_{m=1}^{M} : \sum_{m=1}^{M} \|\mathbf{v}_m\|_2 \leq K_1, \sum_{m=1}^{M} \|\mathbf{v}_m\|_2^2 \leq K_2/M \right\}, \tag{12}$$

with $K_1$ and $K_2$ being two constants. Theorem 2 below provides the excess risk bound for the empirical risk minimizer.

**Theorem 2.** *Assume $M > \delta^{-1}$ and $n > \log(dM)$. Let $f_\star$ be the minimizer of the population risk $L_D(f)$ within the target function class (6) (7). Let $\widehat{f}_M^{\mathbf{W}} = f_M^{\mathbf{W}}(\cdot; \widehat{\mathbf{V}})$ be the empirical risk minimizer given by (11), where in (12) we choose $K_1 = C\sqrt{B(f_\star)}$ and $K_2 = CB(f_\star)\delta^{-1}$, with $C$ being a constant. Then for any joint distribution $P$, with probability at least $1 - \delta$ over $\{\mathbf{W}_m\}_{m \in [M]}$ sampled according to (4) and $\{(\mathbf{x}_{0:N}^{(j)}, y_j)\}_{j \in [n]} \sim_{iid} P$, the excess risk is bounded by*

$$L_D(\widehat{f}_M^{\mathbf{W}}) - L_D(f_\star) \leq \widetilde{\mathcal{O}}\left(\sqrt{B(f_\star)}\left[\sqrt{\frac{1}{n}} + \sqrt{\frac{d^2\delta^{-1}}{M}}\right]\right). \tag{13}$$

The proof of Theorem 2 is contained in Appendix B.2. The proof mostly uses the Rademacher complexity bound for the supremum of empirical process. The main non-trivial technical challenge lies in showing the concentration of $\sup_{f \in \mathcal{V}_M} |\widehat{L}_D(f) - L_D(f)|$, which cannot be simply controlled due to the unboundedness of the infinity norm of functions in the target function class $\mathcal{V}_M$. We dealt with this subtlety by a carefully decomposition of $\sup_{f \in \mathcal{V}_M} |\widehat{L}_D(f) - L_D(f)|$. The seemingly unnatural constraint set (12) is used in bounding different terms in this decomposition.

### 3.3 Examples and comparison

We next give the sample complexity for learning several examples of target functions using the random-feature attention model. We will compare its sample complexity for learning these functions with that of the standard random-feature model [74] (thereafter, we call it the random-feature MLP model, in short RFMLP model). In the RFMLP model, we view $\mathbf{x}_{0:N}$ as an input vector instead of a sequence of vectors denoted as $\text{vec}(\mathbf{x}_{0:N}) = [\mathbf{x}_0; \mathbf{x}_1; \ldots; \mathbf{x}_N; 1] \in \mathbb{R}^{d(N+1)+1}$. The RFMLP is given by

$$f_M^{\text{MLP}}(\mathbf{x}_{0:N}; \mathbf{v}) = \sum_{m=1}^{M} \sigma\big(\langle \mathbf{w}_m, \text{vec}(\mathbf{x}_{0:N}) \rangle\big) \cdot v_m, \quad \{\mathbf{w}_m\}_{m \in [M]} \sim_{\text{iid}} \mathsf{N}(\mathbf{0}, \mathbf{I}/(N+2)). \quad (14)$$

We choose the variance of random weights $\mathbf{w}_m$ to be $1/(N+2)$ to ensure that $\langle \mathbf{w}_m, \text{vec}(\mathbf{x}_{0:N}) \rangle \sim \mathsf{N}(0,1)$ has unit variance. The generalization and approximation properties of the random-feature MLP model have been well-studied in the literature, for example, [7, 8, 60].

We instantiate Theorem 2 on three concrete examples of target functions (calculations of the excess risks in Appendix B.4, where the result for RFMLP are adapted[5] from Arora et al. [7]). In all three cases, the target functions are permutation invariant with respect to $\{\mathbf{x}_i\}_{i \in [N]}$, by which we naturally expect RFA to achieve better sample complexity than RFMLP in accordance with this structure. Although the comparsion between RFA and RFMLP is based on comparing upper bounds on the sample complexity of both models, existing work has also derived lower bounds on the sample complexity of RFMLP, which aligns with the upper bound for RFMLP we used. We do not invoke these lower bounds, as they apply to a special case with a uniform distributional assumption on the input tokens.

**Example 1** (Functions of $\mathbf{x}_0$)**:** We consider functions of $\mathbf{x}_0$ (no dependence on $\mathbf{x}_{1:N}$) of the form

$$f_\star(\mathbf{x}_{0:N}) = \sum_{k=0}^{\infty} \langle \mathbf{x}_0^{\otimes k}, \mathbf{A}_k \rangle, \quad \mathbf{A}_k \in \mathbb{R}^{d^k}, \quad \text{with } B(f_\star) = \sum_{k=0}^{\infty} C_k \|\mathbf{A}_k\|_{\text{Fr}}^2 \text{ by (10)}.$$

By Theorem 2, setting $M = \Theta(d^2 n)$, the excess risk bound gives $\widetilde{\mathcal{O}}(\sqrt{\sum_{k=0}^{\infty} k^{4.5} 4^k \|\mathbf{A}_k\|_{\text{Fr}}^2 / n})$. $\diamond$

As a special case, consider $f_\star(\mathbf{x}_{0:N}) = (\boldsymbol{\beta}^\top \mathbf{x}_0)^p$, which corresponds to taking $\mathbf{A}_k = \boldsymbol{\beta}^{\otimes p}$ for $k = p$ and $\mathbf{A}_k = \mathbf{0}$ for $k \neq p$. The above excess risk of RFA model and the RFMLP model scales as

$$\text{RFA} : \widetilde{\mathcal{O}}\Big(\text{Poly}(p)\sqrt{4^p \|\boldsymbol{\beta}\|_2^{2p}/n}\Big), \quad \text{RFMLP} : \widetilde{\mathcal{O}}\Big(\text{Poly}(p)\sqrt{(N+2)^p \|\boldsymbol{\beta}\|_2^{2p}/n}\Big).$$

Compared to the RFMLP model, the RFA model significantly reduces the necessary sample size by a factor of $(N/4)^p$.

**Example 2** (Average of functions of $\mathbf{x}_i$)**:** We consider average of functions of $\mathbf{x}_i$ of the form

$$f_\star(\mathbf{x}_{0:N}) = \frac{1}{N} \sum_{i=1}^{N} \sum_{k=0}^{\infty} \langle \mathbf{x}_i^{\otimes k}, \mathbf{A}_k \rangle, \quad \mathbf{A}_k \in \mathbb{R}^{d^k}, \quad \text{with } B(f_\star) = \sum_{k=0}^{\infty} C_k \|\mathbf{A}_k\|_{\text{Fr}}^2 \text{ by (10)}.$$

Theorem 2 then gives an $\widetilde{\mathcal{O}}(\sqrt{\sum_{k=0}^{\infty} k^{4.5} 4^k \|\mathbf{A}_k\|_{\text{Fr}}^2 / n})$ excess risk, same as Example 1. $\diamond$

As a specific example, consider $f_\star = \frac{1}{N} \sum_{i=1}^{N} \psi(\langle \boldsymbol{\beta}, \mathbf{x}_i \rangle)$ with $\psi(z) = z \arctan(z/\eta)$ for some $\eta > 2$, $\|\boldsymbol{\beta}\|_2 = 1$. Using the power series expansion of $\psi$, the excess risk bound of RFA model and the RFMLP model scale as

$$\text{RFA} : \widetilde{\mathcal{O}}\Big(\sqrt{\sum_{k=1}^{\infty} k^{4.5}(2/\eta)^{2k}/n}\Big) = \widetilde{\mathcal{O}}(\sqrt{1/n}), \quad \text{RFMLP} : \widetilde{\mathcal{O}}\Big(\sqrt{\sum_{k=1}^{\infty} k^{4.5}[(N+2)/(2\eta)]^{2k}/n}\Big).$$

---

[5]By deriving the corresponding results for Random Features instead of Neural Tangent Kernels.

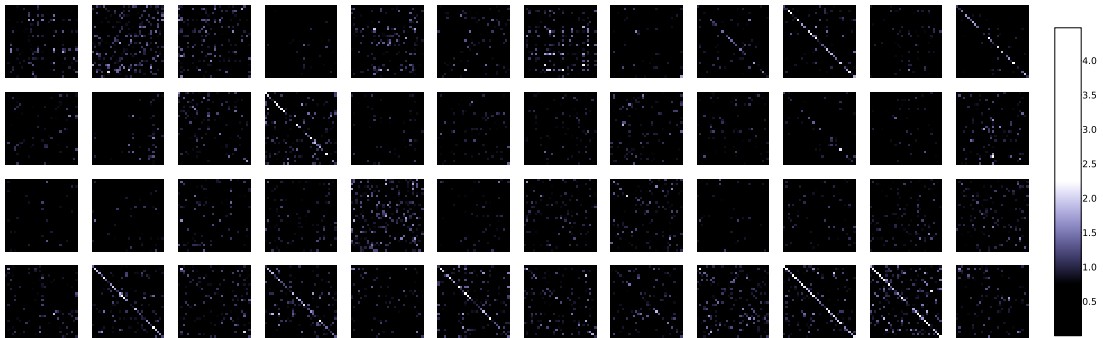

Figure 1: Visualization of weight matrices of the 2nd, 5th, 8th, and 11th layers of the BERT-Base model. Each row contains weight matrices of a layer. All matrices are clipped to the top-left $32 \times 32$ block. Lighter color indicates a larger absolute value.

The latter diverges whenever $\eta \leq (N+2)/2$, in which case the bound is meaningless.

**Example 3** (Correlation-weighted functions)**:** $f_\star$ is the following function:

$$f_\star(\mathbf{x}_{0:N}) = \frac{1}{N}\sum_{i=1}^N F(\langle \mathbf{x}_0, \mathbf{S}\mathbf{x}_i\rangle)G(\mathbf{x}_i), \quad F(t) = \sum_{k=0}^\infty a_k \cdot t^k, \quad G(\mathbf{x}_i) = \sum_{k=0}^\infty \left\langle \mathbf{x}_i^{\otimes k}, \mathbf{G}_k\right\rangle,$$

for $\mathbf{S} \in \mathbb{R}^{d \times d}$, $\{a_k\}_{k\geq 0} \subseteq \mathbb{R}$, $\mathbf{G}_k \in \mathbb{R}^{d^k}$. This target function fully exploits the representation power of the attention layer. Eq. (10) gives $B(f_\star) = \mathcal{O}(\sum_{k=0}^\infty C_k(\sum_{r+s=k} a_r^2 \|\mathbf{S}\|_{\mathsf{Fr}}^{2r}\|\mathbf{G}_s\|_{\mathsf{Fr}}^2))$. $\diamond$

As a specific example, consider $f_{1,\star} = \frac{1}{N}\sum_{i=1}^N \langle \mathbf{x}_0, \mathbf{x}_i\rangle^p$, corresponding to taking $\mathbf{S} = \mathbf{I}_d$, $F(t) = t^p$, and $G \equiv 1$. The excess risk bound of RFA (by Theorem 2) and RFMLP scale as

$$\texttt{RFA} : \widetilde{\mathcal{O}}\Big(\mathrm{Poly}(p)\sqrt{(4d)^p/n}\Big), \qquad \texttt{RFMLP} : \widetilde{\mathcal{O}}\Big(\mathrm{Poly}(p)\sqrt{[(N+2)d]^p/n}\Big).$$

As another example, consider $f_{2,\star} = \frac{1}{N}\sum_{i=1}^N \cos(\langle \mathbf{x}_0, \mathbf{x}_i\rangle)\langle \mathbf{x}_i^{\otimes p}, \mathbf{G}\rangle$ with $\|\mathbf{G}\|_{\mathsf{Fr}} = 1$. Then the excess risk bound of RFA and RFMLP scale as

$$\texttt{RFA} : \widetilde{\mathcal{O}}\Big(\mathrm{Poly}(pd)\sqrt{e^{4\sqrt{d}}4^p/n}\Big), \qquad \texttt{RFMLP} : \widetilde{\mathcal{O}}\Big(\mathrm{Poly}(pNd)\sqrt{e^{2(N+2)\sqrt{d}}N^p/n}\Big).$$

RFA reduces the required sample size by factors of $(N/4)^p$ for $f_{1,\star}$ and $\exp(N\sqrt{d})$ for $f_{2,\star}$.

## 4 Expressivity of biased random-feature attention model

We now move beyond the Gaussian weight assumption by exploring alternative possibilities for the weight distribution in the attention heads. We observe empirically that the weight matrices in transformer architectures learned in practice are often more similar to the identity matrix than a mean-zero matrix (Figure 1; see the details in Appendix D.1. This is also observed in a recent and concurrent work [84]).

Towards understanding this effect, we consider an alternative attention model with *biased* random weights, where the bias is a fixed matrix $\mathbf{W}_0 \in \mathbb{R}^{(d+1)\times(d+1)}$:

$$f_M^{\mathbf{W},\mathbf{W}_0}(\mathbf{x}_{0:N}; \mathbf{V}) = \sum_{m=1}^M \frac{1}{N}\sum_{i=1}^N \sigma\big(\langle \mathbf{W}_0 + \mathbf{W}_m, \widetilde{\mathbf{x}}_0\widetilde{\mathbf{x}}_i^\top\rangle\big)\langle \mathbf{v}_m, \widetilde{\mathbf{x}}_i\rangle. \tag{15}$$

Here $\{\mathbf{W}_m\}_{m\in[M]}$ are again Gaussian random matrices sampled according to (4). The biased random-feature attention model is similar to (5) except that a bias weight $\mathbf{W}_0$ is added. Motivated by our observation, we choose $\mathbf{W}_0 = [\mathbf{I}_{d\times d}, \mathbf{0}_{d\times 1}; \mathbf{0}_{1\times d}, 0] \in \mathbb{R}^{(d+1)\times(d+1)}$, so that the diagonal elements of $\mathbf{W}_0 + \mathbf{W}_m$ will be on average larger than the off-diagonal elements.

## 4.1 Expressivity of biased random-feature attention

Given the formulation of biased random-feature attention models (thereafter, we call it the biased random-feature attention model, in short BRFA model), a natural conjecture is that this model can better fit functions that are the average of function of $\langle \mathbf{x}_0, \mathbf{x}_i \rangle$. We here show that this is indeed the case. In particular, we consider a broad class of target functions $g_\star : \mathcal{X} \to \mathbb{R}$ that take forms

$$g_\star(\mathbf{x}_{0:N}) = \frac{1}{N} \sum_{i=1}^N F(\langle \mathbf{x}_0, \mathbf{x}_i \rangle) G(\mathbf{x}_0, \mathbf{x}_i)$$
$$F(t) = \sum_{k=0}^\infty a_k t^k, \quad G(\mathbf{x}_0, \mathbf{x}_i) = \langle \widetilde{\mathbf{x}}_i^{\otimes 3} \otimes \widetilde{\mathbf{x}}_0^{\otimes 2}, \mathbf{A}_\star \rangle. \tag{16}$$

Here the scalars $\{a_k\}_{k \geq 0} \subseteq \mathbb{R}$ and the tensor $\mathbf{A}_\star \in \mathbb{R}^{d^5}$ parameterizes $g_\star$. As we will explain in Section 4.3, confining $G$ to be a degree-$(3, 2)$ polynomial in $(\mathbf{x}_i, \mathbf{x}_0)$ is essential to our theoretical results. Our next theorem provides the excess risk of learning target function $g_\star$ using the BRFA model (15).

**Theorem 3.** *Given the same setting and assumptions as in Theorem 2, when the population risk minimizer gives $f_\star = g_\star$, with probability at least $1 - \delta$, we have*

$$L_D(\widehat{f}_M^{\mathbf{W}, \mathbf{W}_0}) - L_D(g_\star) = \widetilde{\mathcal{O}}\left( \inf_L \left[ \sqrt{B(g_\star, L)} \left( \sqrt{\frac{1}{n}} + \sqrt{\frac{d^2 \delta^{-1}}{M}} \right) + \varepsilon_L \|g_\star\|_\infty \right] \right), \tag{17}$$

*where $\varepsilon_L = 1/[2^{L+1}(L+1)!]$ and*

$$B(g_\star, L) = \|\mathbf{A}_\star\|_{\mathsf{Fr}}^2 \cdot \left( \sum_{k=0}^\infty |a_k| \cdot C_k \right)^2, \quad \text{with } C_k = (2L+k)^{(k+3)/2} 8^{L+k/2}. \tag{18}$$

The proof of Theorem 3 is contained in Appendix C. We provide the intuitions of the result and an overview of the proof technique in Section 4.3.

## 4.2 Examples and comparison

Compared to the target functions (7) discussed in Section 3.1, functions in (16) may not express the average of arbitrary functions of $\mathbf{x}_0$ and $\mathbf{x}_i$, but are well-suited to express functions of correlations. Consequently, we anticipate that the BRFA model will outperform the RFA model in learning functions of correlations. We will now present three concrete examples of target functions (16), and compare the excess risk of the BRFA model to that of the RFA model. The proof of excess risk is contained in Appendix C.3.

**Example 4** (Low degree polynomials)**:** Consider average of polynomials of $\mathbf{x}_i$ and $\mathbf{x}_0$,

$$g_\star = \frac{1}{N} \sum_{i=1}^N \langle \mathbf{x}_i^{\otimes 3} \otimes \mathbf{x}_0^{\otimes 2}, \mathbf{A} \rangle, \quad \text{with } B(g_\star, L) = \|\mathbf{A}\|_{\mathsf{Fr}}^2 L^3 8^{2L} \text{ by } (18).$$

For any $\eta > 0$, if we take $n \geq \exp(\exp(\Theta(1/\eta)))$, $L = \Theta((1 + \log\log n)^{-1} \log n)$, and $M = \Theta(d^2 n)$, the excess risk will scale as $\widetilde{\mathcal{O}}(\sqrt{\|\mathbf{A}\|_{\mathsf{Fr}}^2 / n^{1-\eta}})$. $\diamond$

Compared with the excess risk of the RFA model as detailed in Example 2, the excess risk bound of the BRFA model loses a factor of $n^{-\eta/2}$.

**Example 5** (Functions of correlations)**:** Consider a special case of functions of correlations,

$$g_\star = \frac{1}{N} \sum_{i=1}^N \langle \mathbf{x}_0, \mathbf{x}_i \rangle^p \langle \boldsymbol{\beta}, \mathbf{x}_i \rangle, \quad \boldsymbol{\beta} \in \mathbb{S}^{d-1}, \text{ with } B(g_\star, L) = (2L+p)^{p+3} 8^{2L+p} \text{ by } (18).$$

For any $\eta > 0$, choosing the same parameters $(n, L, M)$ as Example 4, the excess risk bound scales as $\widetilde{\mathcal{O}}(\sqrt{(\log n + p)^{(p+3)} 8^p / n^{1-\eta}})$. $\diamond$

Consider the required sample size $n_\star$ to reach an accuracy of 0.01. The BRFA model requires $n_\star = \widetilde{\mathcal{O}}((8p + 48)^{p+3})$, whereas the RFA model requires $n_\star = \widetilde{\mathcal{O}}((4d)^p)$. Thus, in comparison to the RFA model, the BRFA model can reduce the required sample size by a factor of $\widetilde{\mathcal{O}}([d/(2p + 12)]^p)$.

**Example 6** (Correlation-weighted functions)**:** Consider the function

$$g_\star = \frac{1}{N} \sum_{i=1}^N \cos(\langle \mathbf{x}_0, \mathbf{x}_i \rangle) \langle \mathbf{x}_i^{\otimes 3}, \mathbf{G} \rangle, \quad \text{with } \|\mathbf{G}\|_{\mathsf{Fr}}^2 \leq 1 \text{ and } B(g_\star, L) = \Theta((8e)^{2L}),$$

where $B(g_\star, L)$ is bounded through the Taylor expansion of $\cos(t)$ and (18). For any $\eta > 0$, choosing the same parameters as Example 4, the excess risk bound scales as $\widetilde{\mathcal{O}}(\sqrt{1/n^{1-\eta}})$. $\diamond$

Consider the required sample size $n_\star$ to reach an accuracy of $0.01$. The BRFA model requires $n_\star = \widetilde{\mathcal{O}}(1)$, whereas the RFA model requires $n_\star = \widetilde{\mathcal{O}}(\mathrm{Poly}(d)\exp(\sqrt{d}))$. Thus, in comparison to the RFA model, the BRFA model can reduce the required sample size by a factor of $\widetilde{\mathcal{O}}(\mathrm{Poly}(d)\exp(\sqrt{d}))$.

### 4.3 Overview of techniques

Here we provide the intuition and an overview of the technique of Theorem 3, with the proof details in Appendix C. To show the sample complexity of learning with the BRFA model, the first step is to derive the kernel $K_{\text{BRFA}}(\mathbf{x}_{0:N}, \mathbf{x}'_{0,N})$ associated with the infinite-width BRFA model. This kernel has a natural feature map, given by $\{\Psi_k : \mathcal{X} \to \mathbb{R}^{d^{2k+1}}\}_{k \geq 0}$, where

$$\Psi_k(\mathbf{x}_{0:N}) = \sum_{i=1}^N \phi(\langle \mathbf{x}_0, \mathbf{x}_i\rangle) \cdot \mathrm{He}_{k-2}(\langle \mathbf{x}_0, \mathbf{x}_i\rangle) \cdot \widetilde{\mathbf{x}}_i^{\otimes k+1} \otimes \widetilde{\mathbf{x}}_0^{\otimes k}, \quad \forall k \geq 2.$$

Here $\phi(t) = (2\pi)^{-1/2}e^{-t^2/2}$ is the Gaussian density function, and $\mathrm{He}_k(z)$ denotes the $k$-th probabilist's Hermite polynomial, with detailed expression and properties given in Appendix A.1. This feature map implies the learnability of the following target function class by the BRFA model,

$$\widetilde{g}_\star(\mathbf{x}_{0:N}) = \frac{1}{N}\sum_{i=1}^N \phi(\langle \mathbf{x}_0, \mathbf{x}_i\rangle) \sum_{k=2}^\infty \mathrm{He}_{k-2}(\langle \mathbf{x}_0, \mathbf{x}_i\rangle) \langle \widetilde{\mathbf{x}}_i^{\otimes k+1} \otimes \widetilde{\mathbf{x}}_0^{\otimes k}, \mathbf{A}_k\rangle, \tag{19}$$

whose RKHS norm associated with kernel $K_{\text{BRFA}}$ is bounded by $B(\widetilde{g}_\star) = \sum_{k=2}^\infty (k-2)!k^2 4^k \|\mathbf{A}_k\|_{\text{Fr}}^2$.

Notice that $\widetilde{g}_\star$ bears similarities to, but also distinct differences from, $g_\star$ as presented in (16). The key difference lies in the $\phi(\langle \mathbf{x}_0, \mathbf{x}_i\rangle)$ factor in $\widetilde{g}_\star$, which is hard to interpret and analyze. To obtain the excess risk bound for learning $g_\star$, we can use $\widetilde{g}_\star$ to approximate $g_\star$ in the $L^\infty$ norm. The excess risk for learning $g_\star$ can be bounded by the summation of the excess risk for learning $\widetilde{g}_\star$ and the approximation error. Acquiring this approximation error bound necessitates a truncation argument of the Taylor expansion of $1/\phi(\cdot)$.

## 5 Numerical experiments

We test our theory by experimentally approximating two types of target functions using the three models under investigation RFA (5), BRFA (15), and RFMLP (14). We choose the target functions to be of form

$$f_{1,p}(\mathbf{x}_{0:N}) = \frac{1}{N}\sum_{i=1}^N \langle \boldsymbol{\beta}, \mathbf{x}_i\rangle^p, \qquad\qquad p \in \mathbb{N}, \quad \boldsymbol{\beta} \in \mathbb{S}^{d-1}, \tag{20}$$

$$f_{2,q}(\mathbf{x}_{0:N}) = \frac{1}{N}\sum_{i=1}^N \langle \mathbf{x}_0, \mathbf{x}_i\rangle^q \langle \boldsymbol{\beta}, \mathbf{x}_i\rangle, \quad q \in \mathbb{N}, \quad \boldsymbol{\beta} \in \mathbb{S}^{d-1}. \tag{21}$$

The first target function (20) is a specific instance of Example 2, whereas the second target function (21) has been considered in both Example 3 and 5.

In our experimental setup, we set the input dimension as $d = 16$ and the number of tokens as $N = 16$. We fix the width of RFA and BRFA to be $M_{\text{RFA}} = M_{\text{BRFA}} = M = 1000$, whereas the width of RFMLP is set as $M_{\text{RFMLP}} = M(d+1) = 17000$. This configuration ensures an equal number of parameters across all three models. To further accentuate the test risk difference between the BRFA and RFA, in BRFA we use a bias matrix of $\mathbf{W}_0 = 4[\mathbf{I}_{d\times d}, \mathbf{0}_{d\times 1}; \mathbf{0}_{1\times d}, 0] \in \mathbb{R}^{(d+1)\times(d+1)}$, which is four times the matrix investigated in our theory. The input distribution is selected as $\{\mathbf{x}_i\}_{0 \leq i \leq N} \sim_{\text{iid}} \mathsf{Unif}(\mathbb{S}^{d-1})$, and we take $y = f_\star(\mathbf{x}_{0:N})$ without any noise. We consider three representative target functions: $f_{1,p}$ for $p = 2, 4$, and $f_{2,p}$ for $p = 3$, as per (20) and (21). We examine a list of sample sizes $n$ from $2^4$ to $2^{12}$. Prior to training with RF models, we standardize the $y_i$'s to have zero mean and unit standard deviation, ensuring that the trivial risk equals 1. We train the RF models using square loss with ridge regularization, selecting the ridge parameter to minimize the test error. The experimental results are displayed in Figure 2.

The left and middle panels of Figure 2 demonstrate a noticeable separation between RFMLP and the other two random-feature attention models for learning these target functions. RFMLP can hardly approximate the target function, whereas RFA and BRFA exhibit significantly better performance. This observation is consistent with our sample complexity analysis detailed in Example 2, where the

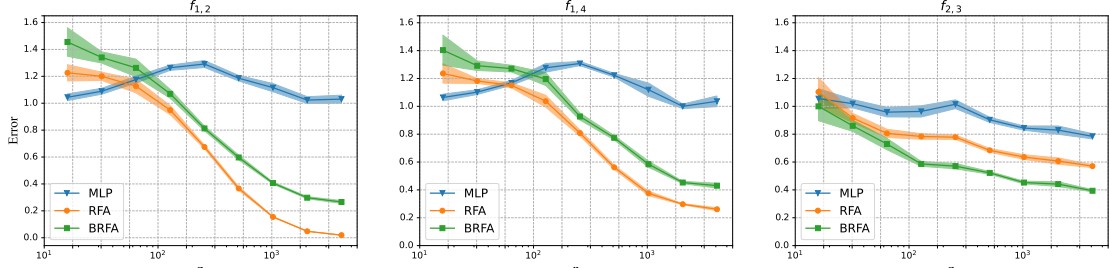

Figure 2: Test error of three RF models for learning $f_{1,2}$ (left), $f_{1,4}$ (mid), and $f_{2,3}$ (right), as per (20) and (21). We set $d, N = 16$, $M_{\texttt{RFA}} = M_{\texttt{BRFA}} = 1000$, and $M_{\texttt{RFMLP}} = 17000$. We train the RF models using square loss with ridge regularization, with the ridge parameter selected to minimize test error. The test error is calculated using $n_{\text{test}} = 1000$ fresh samples. The figure reports the mean and normalized standard error of the test error, based on 5 independent experimental instances.

sample complexity bound of `RFMLP` for learning average of functions of $\mathbf{x}_i$ is found to be $\mathcal{O}((N/4)^p)$ times greater than that of `RFA`.

The performance comparison between `RFA` and `BRFA` depends on the target functions. `RFA` outperforms `BRFA` in learning $f_{1,2}$ and $f_{1,4}$, whereas `BRFA` outperforms `RFA` in learning $f_{2,3}$. The latter phenomenon is as we expected: as demonstrated in Example 3 and 5, `BRFA` is more powerful than `RFA` in approximating the correlation-weighted functions.

We have conducted further experiments with various other target functions, detailed in Appendix D.

## 6 Conclusion

In this work, we introduced and examined the expressivity of two random-feature attention models, namely `RFA` (5) and `BRFA` (15). For general classes of functions that are invariant to the permutation of key tokens $\mathbf{x}_{1:N}$, the excess risk of `RFA` (5) can avoid the dependence on sequence length, in contrast to the standard random-feature model `RFMLP` (14). Moreover, for specific functions that adopt the form of correlation-weighted polynomials (6), the excess risk of `BRFA` can avoid the polynomial dependence on the dimension. These insights enhance our understanding of the attention mechanism within a simplified context. Finally, our work left open many interesting questions for future work, such as the expressivity of softmax attention, the influence of positional encoding in expressivity, and the expressivity of multi-layer transformers.

## Acknowledgment

S. Mei is supported in part by NSF DMS-2210827 and NSF CCF-2315725.

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

# A  Technical tools

## A.1  Basics on Hermite Polynomials

In this section, we briefly review Hermite polynomials and their properties. Let $\mathrm{He}_n$ be the probabilists' Hermite polynomial of degree $n$:

$$\mathrm{He}_n(x) := (-1)^n e^{\frac{x^2}{2}} \frac{d^n}{dx^n} e^{-\frac{x^2}{2}}.$$

Here are some basic properties of Hermite polynomials $\{\mathrm{He}_n\}_{n \geq 0}$:

- Let $(x, y) \sim \mathsf{N}(\mathbf{0}, [1, \rho; \rho, 1])$. Then $\mathbb{E}[\mathrm{He}_m(x)\mathrm{He}_n(y)] = n! \rho^n 1_{m=n}$ for $m, n \geq 0$.
- $\mathrm{He}_n(-x) = (-1)^n \mathrm{He}_n(x)$;
- $\mathrm{He}_{n+1}(x) = x\mathrm{He}_n(x) - \mathrm{He}_n'(x)$.
- $\mathrm{He}_n'(x) = n\mathrm{He}_{n-1}(x)$.

It can be shown that Hermite polynomials are a complete orthogonal basis of the following Hilbert space with $\phi(x) = (2\pi)^{-1/2} e^{-x^2/2}$:

$$L^2(\mathbb{R}, \phi(x)) := \left\{ f : \mathbb{R} \to \mathbb{R} : \quad \int_{-\infty}^{\infty} f(x)^2 \phi(x)\mathrm{d}x < \infty \right\}.$$

For two functions $f, g : \mathbb{R} \to \mathbb{R}$, we define their inner product $\langle f, g \rangle$ as:

$$\langle f, g \rangle := \mathbb{E}_{x \sim \mathsf{N}(0,1)}[f(x)g(x)] = \int_{-\infty}^{\infty} f(x)g(x)\phi(x)\mathrm{d}x.$$

For any function $f \in L^2(\mathbb{R}, \phi(x))$, we can derive its Hermite expansion:

$$f(x) = \sum_{n=0}^{\infty} \frac{a_n}{n!} \mathrm{He}_n(x),$$

where $a_n = \langle f, \mathrm{He}_n \rangle = \int_{-\infty}^{\infty} f(x)\mathrm{He}_n(x)\phi(x)\mathrm{d}x$. Then for another function $g(x) = \sum_{n=0}^{\infty} \frac{b_n}{n!} \mathrm{He}_n(x)$, the inner product of $f$ and $g$ gives:

$$\langle f, g \rangle = \sum_{n=0}^{\infty} \frac{a_n b_n}{n!}.$$

Here are some formulae for Hermite expansion of certain functions.

**Lemma A.1** (See e.g., [38]). *Let $f, g \in L^2(\mathbb{R}, \phi(x))$. Then, for any unit vectors $u, v \in \mathbb{R}^d$, we have that*

$$\mathbb{E}_{x \sim \mathcal{N}(0, \mathbf{I}_d)} \left[ f(u^\top x)g(v^\top x) \right] = \sum_{n=0}^{\infty} \frac{a_n b_n}{n!} \langle u, v \rangle^n,$$

*where $f(x) = \sum_{n=0}^{\infty} \frac{a_n}{n!} \mathrm{He}_n(x)$ and $g(x) = \sum_{n=0}^{\infty} \frac{b_n}{n!} \mathrm{He}_n(x)$.*

**Lemma A.2** (Inverse explicit expression [69]). *Hermite expansion of $x^n$ gives:*

$$x^n = n! \sum_{m=0}^{\lfloor \frac{n}{2} \rfloor} \frac{1}{2^m m!(n-2m)!} \mathrm{He}_{n-2m}(x).$$

**Lemma A.3.** *Let $\sigma_c(x) := \sigma(x+c)$ be the shifted ReLU function. Then the Hermite expansion of $\sigma_c$ gives:*

$$\sigma_c(x) = c\Phi(c) + \phi(c) + \Phi(c)\mathrm{He}_1(x) + \sum_{n=2}^{\infty} \frac{(-1)^n}{n!} \phi(c)\mathrm{He}_{n-2}(c)\mathrm{He}_n(x),$$

*where $\phi(x), \Phi(x)$ are the PDF and CDF of standard Gaussian.*

*Proof of Lemma A.3.* Denote $a_{n,i} := \int_{-c}^{\infty} (x+c)^i \mathrm{He}_n(x)\phi(x)dx$. It is easy to obtain that

$$a_{0,0} = \Phi(c), \qquad a_{0,1} = c\Phi(c) + \phi(c), \qquad a_{1,0} = \phi(c), \quad \text{and} \quad a_{1,1} = \Phi(c).$$

Then using the two different formulae for $a_{n,0}$,

$$a_{n,0} = \int_{-c}^{\infty} \mathrm{He}_n(x)\phi(x)\mathrm{d}x = -\frac{\mathrm{He}_{n+1}(-c)}{n+1}\phi(-c) + \int_{-c}^{\infty} \frac{\mathrm{He}_{n+1}(x)}{n+1} x\phi(x)\mathrm{d}x$$

$$= -\frac{\mathrm{He}_{n+1}(-c)}{n+1}\phi(-c) + \frac{1}{n+1}a_{n+1,1} - \frac{c}{n+1}a_{n+1,0}, \quad n \geq 0, \quad \text{and}$$

$$a_{n,0} = \int_{-c}^{\infty} (x\mathrm{He}_{n-1}(x) - (n-1)\mathrm{He}_{n-2}(x))\phi(x)\mathrm{d}x$$

$$= a_{n-1,1} - ca_{n-1,0} - (n-1)a_{n-2,0}, \quad n \geq 2,$$

we obtain that

$$a_{n,1} = (-1)^n \mathrm{He}_n(c)\phi(c) + ca_{n-1,1} + (n-c^2)a_{n-1,0} - (n-1)ca_{n-2,0}, \quad \text{and}$$

$$a_{n,0} = a_{n-1,1} - ca_{n-1,0} - (n-1)a_{n-2,0}, \quad n \geq 2.$$

Then it is easy to prove by induction that

$$a_{n,1} = (-1)^n \mathrm{He}_{n-2}(c)\phi(c), \quad n \geq 2,$$

$$a_{n,0} = (-1)^{n-1} \mathrm{He}_{n-1}(c)\phi(c), \quad n \geq 1.$$

This completes the proof. $\qquad\square$

### A.2 Basics on Random Features

In this section, we give some basic properties of the random feature model considered in our work, which can be seen as an extension of the standard random feature model (e.g. of Rahimi and Recht [74, 75]) to the *vector-valued case*.

Given a functional $\boldsymbol{\sigma}(x; w) : \mathcal{X} \times \mathcal{W} \to \mathbb{R}^d$. Denote $\mu$ as a probability measure on $\mathcal{W}$. We define the (infinite-width) random feature model as:

$$\mathcal{F} = \{f : f(x) = \langle \mathbf{v}, \boldsymbol{\sigma}(x; \cdot) \rangle_{\mathcal{H}_{\mathcal{W}}}, \quad \mathbf{v} \in \mathcal{H}_{\mathcal{W}}\}, \tag{22}$$

where $\mathcal{H}_{\mathcal{W}} = \{\mathbf{v}(w) : \int_{\mathcal{W}} \mathbf{v}(w)^\top \mathbf{v}(w)\mu(\mathrm{d}w) < \infty\}$ is a Hilbert space with norm $\|\mathbf{v}\|_{\mathcal{H}_{\mathcal{W}}}^2 = \int_{\mathcal{W}} \mathbf{v}(w)^\top \mathbf{v}(w)\mu(\mathrm{d}w)$ and inner product $\langle \mathbf{v}, \mathbf{u} \rangle_{\mathcal{H}_{\mathcal{W}}} = \int_{\mathcal{W}} \mathbf{v}(w)^\top \mathbf{u}(w)\mu(\mathrm{d}w)$. Besides, we endow $\mathcal{F}$ with a norm $\|\cdot\|_{\mathcal{F}}$ and the corresponding inner product $\langle \cdot, \cdot \rangle_{\mathcal{F}}$ defined as:

$$\|f\|_{\mathcal{F}} = \inf_{f=\langle \mathbf{v}, \boldsymbol{\sigma}(\cdot) \rangle_{\mathcal{H}_{\mathcal{W}}}} \|\mathbf{v}\|_{\mathcal{H}_{\mathcal{W}}}, \quad \langle f, g \rangle_{\mathcal{F}} = \frac{\|f+g\|_{\mathcal{F}}^2 - \|f-g\|_{\mathcal{F}}^2}{4}.$$

We further define the corresponding reproducing kernel $K : \mathcal{X} \times \mathcal{X} \to \mathbb{R}$, s.t.

$$K(x, y) = \int_{\mathcal{W}} \boldsymbol{\sigma}(x; w)^\top \boldsymbol{\sigma}(y; w)\mu(\mathrm{d}w),$$

which is positive definite. Define the RKHS induced by this kernel as $\mathcal{H}_K$ with corresponding norm $\|\cdot\|_{\mathcal{H}_K}$ and the inner product $\langle \cdot, \cdot \rangle_{\mathcal{H}_K}$. Then we have the following proposition according to [61]:

**Proposition A.1.** *Given the above definition of $\mathcal{F}$ and $\mathcal{H}_K$, we have that $(\mathcal{F}, \|\cdot\|_{\mathcal{F}}) = (\mathcal{H}_K, \|\cdot\|_{\mathcal{H}_K})$.*

More generally [11], for any feature map $\phi : \mathcal{X} \to \mathcal{H}$ (where $\mathcal{H}$ is a Hilbert space) that induces the kernel $K$, i.e., $K(x, y) = \langle \phi(x), \phi(y) \rangle_{\mathcal{H}}$, we have for any function $f$ that

$$\|f\|_{\mathcal{H}_K} = \inf_{f = \langle \mathbf{u}, \phi(\cdot) \rangle_{\mathcal{H}}} \|\mathbf{u}\|_{\mathcal{H}}, \tag{23}$$

which shows the equivalence among different feature maps that generate the same kernel.

## A.3 Concentration inequalities

**Definition A.1** (Sub-Gaussian and Sub-Exponential random variables [87])**.** *For a random variable $X$, its sub-gaussian norm, denoted $\|X\|_{\psi_2}$, is defined as*

$$\|X\|_{\psi_2} = \inf \left\{ t > 0 : \mathbb{E} \exp \left( X^2/t^2 \right) \leq 2 \right\}.$$

*If $\sigma \equiv \|X\|_{\psi_2} < \infty$, we say that $X$ is $\sigma$-sub-Gaussian.*

*For a random variable $X$, its sub-exponential norm, denoted $\|X\|_{\psi_1}$, is defined as*

$$\|X\|_{\psi_1} = \inf \left\{ t > 0 : \mathbb{E} \exp \left( |X|/t \right) \leq 2 \right\}.$$

*If $\sigma \equiv \|X\|_{\psi_1} < \infty$, we say that $X$ is $\sigma$-sub-exponential.*

**Theorem A.1** (Gaussian concentration inequality (e.g., [89]))**.** *Let $(X_1, \ldots, X_n)$ be a vector of i.i.d. standard Gaussian variables, and let $f : \mathbb{R}^n \to \mathbb{R}$ be $L$-Lipschitz with respect to the Euclidean norm. Then the random variable $f(X) - \mathbb{E}[f(X)]$ is $L$-sub-Gaussian, and hence*

$$\mathbb{P}\big(|f(X) - \mathbb{E}[f(X)]| \geq t\big) \leq 2e^{-\frac{t^2}{2L^2}} \quad \text{for all } t \geq 0.$$

**Theorem A.2** (Bounded difference inequality (e.g., [89]))**.** *Consider a function $f(X) : \mathbb{R}^n \to \mathbb{R}$. Assume that for any $X = (X_1, \ldots, X_n)$ and $X^{i,'} = (X_1, \ldots, X_i', \ldots, X_n)$, we have difference bound $|f(X) - f(X^{i,'})| \leq L_i$. We further assume that the random vector $X = (X_1, X_2, \ldots, X_n)$ has independent components. Then*

$$\mathbb{P}\big(|f(X) - \mathbb{E}[f(X)]| \geq t\big) \leq 2e^{-\frac{2t^2}{\sum_{k=1}^{n} L_k^2}} \quad \text{for all } t \geq 0.$$

**Theorem A.3** (Matrix Bernstein Inequality (e.g., [85]))**.** *Consider a sequence $\{\boldsymbol{S}_k\}_{k \in [n]}$ of independent random matrices with common dimension $d_1 \times d_2$. Assume that*

$$\mathbb{E}\boldsymbol{S}_k = \mathbf{0} \quad \text{and} \quad \|\boldsymbol{S}_k\|_{\mathrm{op}} \leq L \text{ almost surely,} \quad \text{for each index } k.$$

*Introduce the random matrix*

$$\boldsymbol{Z} = \sum_{k=1}^{n} \boldsymbol{S}_k.$$

*Let $v(\boldsymbol{Z})$ be the matrix variance statistic of the sum*

$$v(\boldsymbol{Z}) = \max \left\{ \left\| \mathbb{E}\left( \boldsymbol{Z}\boldsymbol{Z}^\top \right) \right\|_{\mathrm{op}}, \left\| \mathbb{E}\left( \boldsymbol{Z}^\top \boldsymbol{Z} \right) \right\|_{\mathrm{op}} \right\}$$

$$= \max \left\{ \left\| \sum_{k=1}^{n} \mathbb{E}\left( \boldsymbol{S}_k \boldsymbol{S}_k^\top \right) \right\|_{\mathrm{op}}, \left\| \sum_{k=1}^{n} \mathbb{E}\left( \boldsymbol{S}_k^\top \boldsymbol{S}_k \right) \right\|_{\mathrm{op}} \right\}.$$

*Then we have*

$$\mathbb{E}\|\boldsymbol{Z}\|_{\mathrm{op}} \leq \sqrt{2v(\boldsymbol{Z}) \log(d_1 + d_2)} + \frac{1}{3} L \log(d_1 + d_2),$$

*and for all $t \geq 0$,*

$$\mathbb{P}\big(\|\boldsymbol{Z}\|_{\mathrm{op}} \geq t\big) \leq (d_1 + d_2) \exp\left( \frac{-t^2/2}{v(\boldsymbol{Z}) + Lt/3} \right).$$

**Theorem A.4** (Ledoux-Talagrand contraction inequality (e.g., [89]))**.** *Let $\{\xi_i\}_{i \in [n]} \sim_{\mathrm{iid}} \mathsf{Unif}(\{\pm 1\})$ be independent Rademacher random variables. For any set $\mathbb{T} \subset \mathbb{R}^n$ and any family of $L$-Lipschitz functions $\{\phi_j\}_{j \in [n]}$ with $\phi_i(0) = 0$, we have*

$$\mathbb{E}\left[ \sup_{\theta \in \mathbb{T}} \left| \sum_{i=1}^{n} \xi_i \phi_i(\theta_i) \right| \right] \leq 2L \cdot \mathbb{E}\left[ \sup_{\theta \in \mathbb{T}} \left| \sum_{i=1}^{n} \xi_i \theta_i \right| \right]. \tag{24}$$

# B Proofs for Section 3

Throughout this section, we use the notation $\lesssim$ to hide a universal constant $C$. Also, we use $\overline{\sigma}(\mathbf{W}, \mathbf{X})$ to denote a function of $\mathbf{W} \in \mathbb{R}^{(d+1)\times(d+1)}$ and $\mathbf{X} \in \{\overline{\mathbf{X}} \in \mathbb{R}^{(d+1)\times(d+1)} : \|\overline{\mathbf{X}}\|_{\mathsf{Fr}}^2 \leq 4\} \equiv \mathcal{D}$ that satisfies the following properties:

1. For any $\mathbf{X} \in \mathcal{D}$, we have that $\overline{\sigma}(\mathbf{W}, \mathbf{X})$ is $L_1$-Lipschitz with respect to $\mathbf{W}$.
2. For any $\mathbf{X} \in \mathcal{D}$, the expectation $\mathbb{E}_{\mathbf{W}}[\overline{\sigma}(\mathbf{W}, \mathbf{X})] \leq L_2$.
3. For any $\mathbf{X} \in \mathcal{D}$, we have $\overline{\sigma}(\mathbf{0}, \mathbf{X}) \leq L_3$.

Here $L_1$, $L_2$, and $L_3$ are universal constants. Let $\{\mathbf{W}_m\}_{m\in[M]}$ be sampled from Eq. (4). Let $\mathbf{v} : \mathbb{R}^{(d+1)\times(d+1)} \to \mathbb{R}$ with $\mathbb{E}_{\mathbf{W}}[\|\mathbf{v}(\mathbf{W})\|_2^2] \leq R^2$. Consider a random feature model associated with $\overline{\sigma}$ (hereafter, we will refer to it as RF model)

$$f_M^{\mathbf{W}}(\mathbf{x}_{0:N}; \mathbf{V}) = \sum_{m=1}^{M} \frac{1}{N} \sum_{i=1}^{N} \overline{\sigma}(\mathbf{W}_m, \widetilde{\mathbf{x}}_0 \widetilde{\mathbf{x}}_i^\top) \langle \mathbf{v}_m, \widetilde{\mathbf{x}}_i \rangle, \tag{25}$$

as well as the infinite-width version of the random feature model,

$$f_{\mathbf{v}}(\mathbf{x}_{0:N}) = \frac{1}{N} \sum_{i=1}^{N} \mathbb{E}_{\mathbf{W}}\left[\overline{\sigma}(\mathbf{W}, \widetilde{\mathbf{x}}_0 \widetilde{\mathbf{x}}_i^\top) \langle \mathbf{v}(\mathbf{W}), \widetilde{\mathbf{x}}_i \rangle\right]. \tag{26}$$

Note that both model RFA and BRFA correspond to special choices of $\overline{\sigma}$. Thus, all lemmas and theorems in this section are applicable to both model RFA and model BRFA.

## B.1 Proof of Theorem 1

To prove Theorem 1, we first state two auxilliary lemmas, Lemma B.1 and B.2.

**Lemma B.1** (From infinite-width RF model to finite-width RF model). *Consider $f_{\mathbf{v}}$ that takes the form as Eq. (26), with $\mathbb{E}_{\mathbf{W}}[\|\mathbf{v}(\mathbf{W})\|_2^2] \leq R^2$. Let $\mathbf{x}_{0:N} \sim P$. Define the $\|\cdot\|_{L^2(P)}^2$ norm by*

$$\|g\|_{L^2(P)}^2 = \int g(\mathbf{x}_{0:N})^2 P(\mathrm{d}\mathbf{x}_{0:N}).$$

*Then with probability at least $1 - \delta$, there exists a sequence of vectors $\{\mathbf{v}_m\}_{m=1}^{M} \subseteq \mathbb{R}^{d+1}$ and constant universal $C < \infty$ that only depends on $L_1$, $L_2$, and $L_3$ s.t.*

$$\left\| f_{\mathbf{v}} - \frac{1}{N} \sum_{i=1}^{N} \sum_{m=1}^{M} \overline{\sigma}(\mathbf{W}_m, \widetilde{\mathbf{x}}_0 \widetilde{\mathbf{x}}_i^\top) \langle \mathbf{v}_m, \widetilde{\mathbf{x}}_i \rangle \right\|_{L^2(P)}^2 \leq \frac{C(d^2 + \log M)R^2\delta^{-1}}{M} \tag{27}$$

$$\sum_{m=1}^{M} \left\| \mathbf{v}_m \right\|_2 \leq \sqrt{2}R + \sqrt{\frac{CR^2\delta^{-1}}{M}} \quad \text{and} \quad \sum_{m=1}^{M} \left\| \mathbf{v}_m \right\|_2^2 \leq \frac{CR^2\delta^{-1}}{M}. \tag{28}$$

**Lemma B.2.** *Under the setting of Theorem 1, let $f_\star$ be a target function of form (6) and (7). Then there exists an infinite-width RFA model (26) with $\mathbf{v} : \mathbb{R}^{(d+1)\times(d+1)} \to \mathbb{R}$ such that*

$$f_\star = \frac{1}{N} \sum_{i=1}^{N} \mathbb{E}_{\mathbf{W}}[\overline{\sigma}(\mathbf{W}, \widetilde{\mathbf{x}}_0 \widetilde{\mathbf{x}}_i^\top) \langle \mathbf{v}(\mathbf{W}), \widetilde{\mathbf{x}}_i \rangle], \tag{29}$$

*with*

$$\mathbb{E}_{\mathbf{W}}[\|\mathbf{v}(\mathbf{W})\|_2^2] \leq B(f_\star),$$

*where $B(f_\star)$ is as defined in (10).*

The proofs of Lemma B.1 and B.2 are given in Section B.1.1. Now we assume these two lemmas hold, and use them to prove Theorem 1.

*Proof of Theorem 1.* For any function that takes form (6) and (7), by Lemma B.2, it admits representation (29) with $\mathbb{E}_{\mathbf{W}}[\|\mathbf{v}(\mathbf{W})\|_2^2] \leq B(f_\star)$. Then by Lemma B.1, since RFA model is a special case of the RF model, there exists $\{\mathbf{v}_m\}_{m\in[M]}$ such that Eq. (27) and Eq. (28) hold with probability larger than $1 - \delta$. This proves Theorem 1. $\qquad\square$

### B.1.1 Proof of auxiliary lemmas

*Proof of Lemma B.1.*

**Step 1. Proof of Eq.** (27)**.** To prove Eq. (27), we use a truncation argument.

Fix a $R_{\mathbf{W}} > 0$ which we will choose its value later in the proof. Recall that we have $\mathbf{W}_1, \ldots, \mathbf{W}_m$ i.i.d. with $\mathbf{W}_{m,i,j} \sim_{\text{iid}} \mathsf{N}(0, 1/4)$. Define $\mathbf{v}_m = \mathbf{v}(\mathbf{W}_m) \mathbf{1}\{\|\mathbf{W}_m\|_{\mathsf{Fr}} \leq R_{\mathbf{W}}/2\}/M$ for $m = 1, \ldots, M$. Consider the truncated infinite-width random feature model $f_{\mathbf{v}}^{R_{\mathbf{W}}} = \frac{1}{N} \sum_{i=1}^{N} \mathbb{E}_{\mathbf{W}}[\overline{\sigma}(\mathbf{W}, \widetilde{\mathbf{x}}_0 \widetilde{\mathbf{x}}_i^{\top}) \langle \mathbf{v}(\mathbf{W}), \widetilde{\mathbf{x}}_i \rangle \mathbf{1}\{\|\mathbf{W}\|_{\mathsf{Fr}} \leq R_{\mathbf{W}}/2\}]$, we have

$$\mathbb{E}_{\mathbf{W}}\left[\left\|f_{\mathbf{v}}^{R_{\mathbf{W}}} - \frac{1}{N}\sum_{i=1}^{N}\sum_{m=1}^{M} \overline{\sigma}(\mathbf{W}, \widetilde{\mathbf{x}}_0 \widetilde{\mathbf{x}}_i^{\top}) \langle \mathbf{v}_m, \widetilde{\mathbf{x}}_i \rangle \right\|_{L^2(P)}^2\right]$$

$$\leq \mathbb{E}_{\mathbf{X}}\left(\mathbb{E}_{\mathbf{W}}\left[\left\|\frac{1}{N}\sum_{i=1}^{N}\overline{\sigma}(\mathbf{W}, \widetilde{\mathbf{x}}_0 \widetilde{\mathbf{x}}_i^{\top})\widetilde{\mathbf{x}}_i\right\|_2^2 \left\|\mathbf{v}(\mathbf{W})\right\|_2^2 \mathbf{1}\{\|\mathbf{W}\|_{\mathsf{Fr}} \leq R_{\mathbf{W}}/2\}\right]\right)/M$$

$$\leq \mathbb{E}_{\mathbf{X}}\left(\mathbb{E}_{\mathbf{W}}\left[\frac{2}{N}\sum_{i=1}^{N}\|\widetilde{\mathbf{x}}_i\|_2^2(L_1^2\|\mathbf{W}\|_2^2 + L_3^2)\|\mathbf{v}(\mathbf{W})\|_2^2 \mathbf{1}\{\|\mathbf{W}\|_{\mathsf{Fr}} \leq R_{\mathbf{W}}/2\}\right]\right)/M$$

$$\leq \frac{\widetilde{C}R_{\mathbf{W}}^2 R^2}{M},$$

where $\widetilde{C_1}$ only depends on $L_1$, $L_2$, and $L_3$. Then using Markov's inequality,

$$\left\|f_{\mathbf{v}}^{R_{\mathbf{W}}} - \frac{1}{N}\sum_{i=1}^{N}\sum_{m=1}^{M}\overline{\sigma}(\mathbf{W}, \widetilde{\mathbf{x}}_0 \widetilde{\mathbf{x}}_i^{\top}) \langle \mathbf{v}_m, \widetilde{\mathbf{x}}_i \rangle\right\|_{L^2(P)}^2 \leq \frac{3\widetilde{C}R_{\mathbf{W}}^2 R^2}{\delta M}$$

holds with probability at least $1 - \delta/3$. Next for the difference between $f_{\mathbf{v}}^{R_{\mathbf{W}}}$ and $f_{\mathbf{v}}$, we have

$$\|f_{\mathbf{v}} - f_{\mathbf{v}}^{R_{\mathbf{W}}}\|_{L^2(P)}^2 \leq \mathbb{E}_{\mathbf{X}}\left[\left(\mathbb{E}_{\mathbf{W}}\left[\frac{1}{N}\sum_{i=1}^{N}\overline{\sigma}(\mathbf{W}, \widetilde{\mathbf{x}}_0 \widetilde{\mathbf{x}}_i^{\top}) \langle \mathbf{v}, \widetilde{\mathbf{x}}_i \rangle \mathbf{1}\{\|\mathbf{W}\|_{\mathsf{Fr}} \geq R_{\mathbf{W}}/2\}\right]\right)^2\right]$$

$$\leq \mathbb{E}_{\mathbf{X}}\left[\frac{1}{N}\sum_{i=1}^{N}\mathbb{E}_{\mathbf{W}}\left[\langle \mathbf{v}, \widetilde{\mathbf{x}}_i\rangle^2\right]\sqrt{\mathbb{E}_{\mathbf{W}}[\overline{\sigma}(\mathbf{W}, \widetilde{\mathbf{x}}_0\widetilde{\mathbf{x}}_i^{\top})^4]}\sqrt{\mathbb{P}(\|\mathbf{W}\|_{\mathsf{Fr}} \geq R_{\mathbf{W}}/2)}\right]$$

$$\leq \widetilde{C_2}R^2 \mathbb{P}(\|\mathbf{W}\|_{\mathsf{Fr}} > R_{\mathbf{W}}/2)^{\frac{1}{2}}.$$

Here $\widetilde{C_2}$ is a constant that only depends on $L_1$ and $L_2$. By concentration of functions of Gaussian random vectors (Theorem A.1), $\|\overline{\sigma}(\mathbf{W}, \widetilde{\mathbf{x}}_0\widetilde{\mathbf{x}}_i^{\top})\|_{\psi_2} \leq L_1 + L_2$ for any $i$. So in the last inequality, we used the bound $(\mathbb{E}_{\mathbf{W}}[\overline{\sigma}(\mathbf{W}, \widetilde{\mathbf{x}}_0\widetilde{\mathbf{x}}_i^{\top})^4])^{1/2}$ by $\Theta((L_1 + L_2)^2)$. To bound $\mathbb{P}(\|\mathbf{W}\|_{\mathsf{Fr}} > R_{\mathbf{W}}/2)$, we use concentration of functions of Gaussian random vectors (Theorem A.1) again, and get that

$$\mathbb{P}\big(\|\mathbf{W}\|_{\mathsf{Fr}} - \mathbb{E}(\|\mathbf{W}\|_{\mathsf{Fr}}) \geq t/2\big) \leq \exp\big(-t^2/2\big).$$

Take $C = max(\widetilde{C_1}, \widetilde{C_2})$. Since $\mathbb{E}(\|\mathbf{W}\|_{\mathsf{Fr}}) \leq (\mathbb{E}\|\mathbf{W}\|_{\mathsf{Fr}}^2)^{1/2} \leq d + 1$, by choosing $R_{\mathbf{W}} = d + 1 + C\sqrt{\log M}$, the above probability is less than $1/M^2$. Then

$$\left\|f_{\mathbf{v}} - \frac{1}{N}\sum_{i=1}^{N}\sum_{m=1}^{M}\overline{\sigma}(\mathbf{W}, \widetilde{\mathbf{x}}_0\widetilde{\mathbf{x}}_i^{\top}) \langle \mathbf{v}_m, \widetilde{\mathbf{x}}_i\rangle\right\|_{L^2(P)}^2 \leq \frac{3CR_{\mathbf{W}}^2 R^2}{\delta M} + CR^2 \mathbb{P}(\|\mathbf{W}\|_{\mathsf{Fr}} > R_{\mathbf{W}}/2)^{\frac{1}{2}}$$

$$\leq \frac{C(\log M + d^2)R^2}{\delta M}$$

with probability larger than $1 - \delta/3$. This proves Eq. (27).

**Step 2. Proof of Eq.** (28)**.** By Chebyshev's inequality,

$$\mathbb{P}\left(\sum_{m=1}^{M}\|\mathbf{v}_m\|_2 - \mathbb{E}\left[\|\mathbf{v}(\mathbf{W})\|_2 \mathbf{1}\{\|\mathbf{W}_m\|_{\mathsf{Fr}} \leq R_{\mathbf{W}}/2\}\right] \geq \sqrt{\frac{6R^2}{\delta M}}\right)$$

$$\leq \frac{\mathbb{E}\Big[\,\|\mathbf{v}(\mathbf{W})\|_2^2\,\mathbf{1}\,\{\|\mathbf{W}\|_{\mathsf{Fr}} \leq R_{\mathbf{W}}/2\}\Big]\delta M}{6R^2 M} \leq \frac{\delta}{3}.$$

Combining with the fact that $\mathbb{E}\Big[\,\|\mathbf{v}(\mathbf{W})\|_2\,\mathbf{1}\,\{\|\mathbf{W}\|_{\mathsf{Fr}} \leq R_{\mathbf{W}}/2\}\Big] \leq \sqrt{\mathbb{E}\Big[\,\|\mathbf{v}(\mathbf{W})\|_2^2\Big]} \leq R$, we have

$$\mathbb{P}\bigg(\sum_{m=1}^{M} \|\mathbf{v}_m\|_2 \geq R + \sqrt{\frac{6R^2}{\delta M}}\bigg) \leq \frac{\delta}{3}.$$

For the second part of (28), Markov inequality gives

$$\mathbb{P}\bigg(\sum_{m=1}^{M} \|\mathbf{v}_m\|_2^2 \geq \frac{6R^2}{\delta M}\bigg) \leq \frac{\mathbb{E}\Big[\,\|\mathbf{v}(\mathbf{W})\|_2^2\,\mathbf{1}\,\{\|\mathbf{W}_m\|_{\mathsf{Fr}} \leq R_{\mathbf{W}}/2\}\Big]\delta M}{6R^2 M} \leq \frac{\delta}{3}.$$

This proves Eq. (28) and completes the proof of Lemma B.1. $\qquad\square$

*Proof of Lemma B.2.*

To get the kernel of the RFA model, we have

$$K(\mathbf{x}_{0:N}, \mathbf{x}'_{0:N}) = \frac{1}{N^2}\mathbb{E}_{\mathbf{W}}\Big[\sum_{i,j=1}^{N} \sigma(\langle\mathbf{W}, \widetilde{\mathbf{x}}_0\widetilde{\mathbf{x}}_i^\top\rangle)\sigma(\langle\mathbf{W}, \widetilde{\mathbf{x}}'_0(\widetilde{\mathbf{x}}'_j)^\top\rangle)\langle\widetilde{\mathbf{x}}_i, \widetilde{\mathbf{x}}'_j\rangle\Big].$$

We first consider a single component in the sum, which is

$$\mathbb{E}_{\mathbf{W}}\Big[\sigma\big(\langle\mathbf{W}, \widetilde{\mathbf{x}}_0\widetilde{\mathbf{x}}_i^\top\rangle\big)\sigma\big(\langle\mathbf{W}, \widetilde{\mathbf{x}}'_0(\widetilde{\mathbf{x}}_j)'\rangle\big)\Big]\langle\widetilde{\mathbf{x}}_i, \widetilde{\mathbf{x}}'_j\rangle. \tag{30}$$

Let $u_{i,j} = \langle\widetilde{\mathbf{x}}_0\widetilde{\mathbf{x}}_i^\top, \widetilde{\mathbf{x}}'_0(\widetilde{\mathbf{x}}'_j)^\top\rangle/4$. Let $\mathsf{N}_2(\rho)$ denote a bivariate normal distribution with marginals are $\mathsf{N}(0,1)$ and the correlation is $\rho \in [-1, 1]$. Then (30) can be expanded as follows:

$$\mathbb{E}_{\mathbf{W}}\Big[\sigma\big(\langle\mathbf{W}, \widetilde{\mathbf{x}}_0\widetilde{\mathbf{x}}_i^\top\rangle\big)\sigma\big(\langle\mathbf{W}, \widetilde{\mathbf{x}}'_0(\widetilde{\mathbf{x}}'_j)^\top\rangle\big)\Big]\langle\widetilde{\mathbf{x}}_i, \widetilde{\mathbf{x}}'_j\rangle$$

$$= \mathbb{E}_{Z_1, Z_2 \sim \mathsf{N}_2(u_{i,j})}\Big[\sigma(Z_1)\sigma(Z_2)\Big]\langle\widetilde{\mathbf{x}}_i, \widetilde{\mathbf{x}}'_j\rangle$$

$$= \frac{1}{2\pi}\Big(u_{i,j}(\pi/2 - \arccos u_{i,j}) + \sqrt{1 - u_{i,j}^2}\Big)\langle\widetilde{\mathbf{x}}_i, \widetilde{\mathbf{x}}'_j\rangle$$

$$= \frac{1}{2\pi}\bigg(1 + \frac{\pi}{2}u_{i,j} + \sum_{\ell=1}^{\infty}\frac{(2\ell-3)!!}{(2\ell)!!(2\ell-1)}u_{i,j}^{2\ell}\bigg)\langle\widetilde{\mathbf{x}}_i, \widetilde{\mathbf{x}}'_j\rangle$$

$$= \sum_{\ell\in\{0,1\}\cup\{2k\}_{k\geq1}} c_\ell\,\big\langle\widetilde{\mathbf{x}}_0\widetilde{\mathbf{x}}_i^\top, \widetilde{\mathbf{x}}'_0(\widetilde{\mathbf{x}}'_j)^\top\big\rangle^\ell\,4^{-\ell}\langle\widetilde{\mathbf{x}}_i, \widetilde{\mathbf{x}}'_j\rangle$$

$$= \sum_{\ell\in\{0,1\}\cup\{2k\}_{k\geq1}} c_\ell\Big\langle 2^{-\ell}(\widetilde{\mathbf{x}}_0\widetilde{\mathbf{x}}_i^\top)^{\otimes\ell}\otimes\widetilde{\mathbf{x}}_i, 2^{-\ell}(\widetilde{\mathbf{x}}'_0(\widetilde{\mathbf{x}}'_j)^\top)^{\otimes\ell}\otimes\widetilde{\mathbf{x}}'_j\Big\rangle.$$

Here the coefficients $\{c_\ell\}$ satisfy

$$c_0 = 1/(2\pi), \quad c_1 = 1/4, \quad \text{and} \quad c_{2\ell} = \frac{(2\ell-3)!!}{2\pi(2\ell)!!(2\ell-1)} = O(\ell^{-\frac{5}{2}}) \ \text{ for } \ell \geq 1.$$

Therefore, the kernel can be expressed as:

$$K(\mathbf{x}_{0:N}, \mathbf{x}'_{0:N})$$

$$= \frac{1}{N^2}\sum_{1\leq i,j\leq N}\sum_{\ell\in\{0,1\}\cup\{2k\}_{k\geq1}} c_\ell\Big\langle 2^{-\ell}(\widetilde{\mathbf{x}}_0\widetilde{\mathbf{x}}_i^\top)^{\otimes\ell}\otimes\widetilde{\mathbf{x}}_i, 2^{-\ell}(\widetilde{\mathbf{x}}'_0(\widetilde{\mathbf{x}}'_j)^\top)^{\otimes\ell}\otimes\widetilde{\mathbf{x}}'_j\Big\rangle$$

$$= \sum_{\ell\in\{0,1\}\cup\{2k\}_{k\geq1}}\Big\langle\frac{\sqrt{c_\ell}}{N}\sum_{i=1}^{N}2^{-\ell}(\widetilde{\mathbf{x}}_0\widetilde{\mathbf{x}}_i^\top)^{\otimes\ell}\otimes\widetilde{\mathbf{x}}_i, \frac{\sqrt{c_\ell}}{N}\sum_{j=1}^{N}2^{-\ell}(\widetilde{\mathbf{x}}'_0(\widetilde{\mathbf{x}}'_j)^\top)^{\otimes\ell}\otimes\widetilde{\mathbf{x}}'_j\Big\rangle.$$

Now we reformulate the target function as

$$
f_\star = \frac{1}{N} \sum_{i=1}^{N} \sum_{\ell=0}^{\infty} \sum_{\max\{r,s\}=\ell} \left\langle \widetilde{\mathbf{x}}_0^{\otimes r} \otimes \widetilde{\mathbf{x}}_i^{\otimes s}, \mathbf{f}_{rs} \right\rangle
$$

$$
= \frac{1}{N} \sum_{i=1}^{N} \sum_{\ell=2k,k\geq 0}^{\infty} \sum_{\max\{r,s\}=\ell \text{ or } \ell-1} \left\langle (\widetilde{\mathbf{x}}_0 \otimes \widetilde{\mathbf{x}}_i)^{\otimes \ell} \otimes \widetilde{\mathbf{x}}_i, \widetilde{\mathbf{f}}_{rs} \right\rangle
$$

$$
= \sum_{\ell=2k,k\geq 0}^{\infty} \left\langle \frac{2^\ell}{\sqrt{c_\ell}} \sum_{\max\{r,s\}=\ell \text{ or } \ell-1} \widetilde{\mathbf{f}}_{rs}, \frac{\sqrt{c_\ell}}{N} \sum_{i=1}^{N} 2^{-\ell} (\widetilde{\mathbf{x}}_0 \widetilde{\mathbf{x}}_i^\top)^{\otimes \ell} \otimes \widetilde{\mathbf{x}}_i \right\rangle,
$$

where $\widetilde{\mathbf{f}}_{rs}$ is a transpose of $\mathbf{f}_{rs} \otimes \mathbf{1}_{d+1}^{\otimes(2\ell-r-s+1)}$ with $\mathbf{1}_{d+1} = (0,\ldots,0,1)$, such that $\left\langle \widetilde{\mathbf{x}}_0^{\otimes r} \otimes \widetilde{\mathbf{x}}_i^{\otimes s}, \mathbf{f}_{rs} \right\rangle = \left\langle (\widetilde{\mathbf{x}}_0 \otimes \widetilde{\mathbf{x}}_i)^{\otimes \ell} \otimes \widetilde{\mathbf{x}}_i, \widetilde{\mathbf{f}}_{rs} \right\rangle$ for any $r \geq 0$ and $s \geq 0$. Then by the feature map equivalence property (23), the RKHS norm of $f^\star$ can be bounded as

$$
\|f^\star\|_{\mathcal{H}_K}^2 \leq \left\| \sum_{\ell=2k,k\geq 0}^{\infty} \left\langle \frac{2^\ell}{\sqrt{c_\ell}} \sum_{\max\{r,s\}=\ell \text{ or } \ell-1} \widetilde{\mathbf{f}}_{rs}, \frac{\sqrt{c_\ell}}{N} \sum_{i=1}^{N} 2^{-\ell} (\widetilde{\mathbf{x}}_0 \widetilde{\mathbf{x}}_i^\top)^{\otimes \ell} \otimes \widetilde{\mathbf{x}}_i \right\rangle \right\|_{\mathcal{H}_K}^2
$$

$$
= \sum_{\ell=2k,k\geq 0}^{\infty} \left\langle \frac{2^\ell}{\sqrt{c_\ell}} \sum_{\max\{r,s\}=\ell \text{ or } \ell-1} \widetilde{\mathbf{f}}_{rs}, \frac{2^\ell}{\sqrt{c_\ell}} \sum_{\max\{r,s\}=\ell \text{ or } \ell-1} \widetilde{\mathbf{f}}_{rs} \right\rangle
$$

$$
= \sum_{\ell=2k,k\geq 0}^{\infty} 4^\ell c_\ell^{-1} \left\| \sum_{\max\{r,s\}=\ell \text{ or } \ell-1} \widetilde{\mathbf{f}}_{rs} \right\|_{\mathrm{Fr}}^2
$$

$$
\leq \sum_{k=0}^{\infty} 4^k k^{4.5} \sum_{\max\{r,s\}=k} \left\| \widetilde{\mathbf{f}}_{rs} \right\|_{\mathrm{Fr}}^2.
$$

Thus, using again the property (23) with the original feature map of the random feature model, there exists $\mathbf{v} : \mathbb{R}^{(d+1)\times(d+1)} \to \mathbb{R}^{d+1}$ such that

$$
f_\star = \frac{1}{N} \sum_{i=1}^{N} \mathbb{E}_{\mathbf{W}} [\overline{\sigma}(\mathbf{W}, \widetilde{\mathbf{x}}_0 \widetilde{\mathbf{x}}_i^\top) \langle \mathbf{v}(\mathbf{W}), \widetilde{\mathbf{x}}_i \rangle], \quad \text{with}
$$

$$
\mathbb{E}_{\mathbf{W}} [\|\mathbf{v}(\mathbf{W})\|_2^2] \leq \sum_{k=0}^{\infty} 4^k k^{4.5} \sum_{\max\{r,s\}=k} \|\widetilde{\mathbf{f}}_{rs}\|_{\mathrm{Fr}}^2.
$$

Notice that $\|\widetilde{\mathbf{f}}_{rs}\|_{\mathrm{Fr}}^2 = \|\mathbf{f}_{rs}\|_{\mathrm{Fr}}^2$ by our construction of $\widetilde{\mathbf{f}}_{rs}$, so that the right-hand-side of the equation above coincides with Eq. (10). This proves Lemma B.2. $\qquad\square$

## B.2 Preliminary proposition for Theorem 2

To prove Theorem 2, we first present and prove the following proposition that gives a high probability bound for the difference between the empirical risk and the population risk. In the proposition and lemmas below, we denote $\mathbf{X} = \{\mathbf{x}_{0:N}^{(j)}\}_{j\in[n]}$ and $\mathbf{y} = \{y_j\}_{j\in[n]}$.

**Proposition B.1.** *Under the setting of Theorem 2. Consider the finite width* RF *model* (25):

$$
f_M^{\mathbf{W}}(\mathbf{x}_{0:N}; \mathbf{V}) = \sum_{m=1}^{M} \frac{1}{N} \sum_{j=1}^{n} \overline{\sigma}(\mathbf{W}_m, \widetilde{\mathbf{x}}_0 \widetilde{\mathbf{x}}_i^\top) \langle \mathbf{v}_m, \widetilde{\mathbf{x}}_i \rangle.
$$

*Then with probability at least* $1 - \delta$ *(w.r.t.* $\mathbf{W}$, $\mathbf{y}$, *and* $\mathbf{X}$*), we have*

$$
\sup_{\mathbf{V}\in\mathcal{V}_M} \left| \frac{1}{n} \sum_{j=1}^{n} \ell\left( f_M^{\mathbf{W}}\left(\mathbf{x}_{0:N}^{(j)}; \mathbf{V}\right), y_j \right) - \mathbb{E}_{\mathbf{x}_{0:N}, y} \ell\left( f_M^{\mathbf{W}}(\mathbf{x}_{0:N}; \mathbf{V}), y \right) \right|
$$

$$
\lesssim K_1 \sqrt{\frac{\log(dM)\log(nNM)}{n}} + \sqrt{\log\left(\frac{6}{\delta}\right)} \left( \frac{K_1}{\sqrt{n}} + \sqrt{\frac{K_2}{M}} \right). \tag{31}
$$

The main difficulty of the proof of Proposition B.1 comes from that $\ell\left(f_M^{\mathbf{W}}, y\right)$ might be unbounded and that $\ell(f_M^{\mathbf{W}}(\mathbf{x}_{0:N}^{(j)}), y_j)$ are not independent across $j$ (since they share the same $\{\mathbf{W}_m\}_{m\in[M]}$). So we begin with several lemmas below.

**Lemma B.3.** *Let* $\{\xi_j\}_{j\in[n]}$ *be a set of* $i.i.d.$ *Rademacher random variables. Under the setting of Proposition B.1,*

$$\mathbb{E}_{\mathbf{X},\mathbf{y},\mathbf{W},\boldsymbol{\xi}}\Big[\sup_{\mathbf{V}\in\mathcal{V}_M}\Big|\frac{1}{n}\sum_{j=1}^n \xi_j\ell\Big(f_M^{\mathbf{W}}\left(\mathbf{x}_{0:N}^{(j)};\mathbf{V}\right),y_j\Big)\Big|\Big] \lesssim K_1\sqrt{\frac{\log(dM)\log(nNM)}{n}}. \tag{32}$$

*Furthermore, any fixed* $\mathbf{X}$ *and* $\mathbf{y}$,

$$\mathbb{E}_{\mathbf{W},\boldsymbol{\xi}}\Big[\sup_{\mathbf{V}\in\mathcal{V}_M}\Big|\frac{1}{n}\sum_{j=1}^n \xi_j\ell\Big(f_M^{\mathbf{W}}\left(\mathbf{x}_{0:N}^{(j)};\mathbf{V}\right),y_j\Big)\Big|\Big] \lesssim K_1\sqrt{\frac{\log(dM)\log(nNM)}{n}}. \tag{33}$$

**Lemma B.4.** *Under the setting of Proposition B.1. With probability at least* $1-\delta/3$ *over* $\mathbf{X}$, $\mathbf{y}$, *and* $\mathbf{W}$,

$$\sup_{\mathbf{V}\in\mathcal{V}_M}\Big|\frac{1}{n}\sum_{j=1}^n \ell\Big(f_M^{\mathbf{W}}\left(\mathbf{x}_{0:N}^{(j)};\mathbf{V}\right),y_j\Big) - \mathbb{E}_{\mathbf{W}}\Big[\frac{1}{n}\sum_{j=1}^n \ell\Big(f_M^{\mathbf{W}}\left(\mathbf{x}_{0:N}^{(j)};\mathbf{V}\right),y_j\Big)\Big]\Big|$$

$$\lesssim \sqrt{\frac{K_2\log(6/\delta)}{M}} + K_1\sqrt{\frac{\log(dM)\log(nNM)}{n}}. \tag{34}$$

**Lemma B.5.** *Under the setting of Proposition B.1. With probability at least* $1-\delta/3$ *over* $\mathbf{X}$ *and* $\mathbf{y}$,

$$\sup_{\mathbf{V}\in\mathcal{V}_M}\Big|\mathbb{E}_{\mathbf{W}}\Big[\frac{1}{n}\sum_{j=1}^n \ell\Big(f_M^{\mathbf{W}}\left(\mathbf{x}_{0:N}^{(j)};\mathbf{V}\right),y_j\Big)\Big] - \mathbb{E}_{\mathbf{X},\mathbf{y},\mathbf{W}}\Big[\frac{1}{n}\sum_{j=1}^n \ell\Big(f_M^{\mathbf{W}}\left(\mathbf{x}_{0:N}^{(j)};\mathbf{V}\right),y_j\Big)\Big]\Big|$$

$$\lesssim K_1\sqrt{\frac{\log(dM)\log(nNM)}{n}} + K_1\sqrt{\frac{\log(6/\delta)}{n}}. \tag{35}$$

**Lemma B.6.** *Under the setting of Proposition B.1. With probability at least* $1-\delta/3$ *over* $\mathbf{W}$,

$$\sup_{\mathbf{V}\in\mathcal{V}_M}\Big|\mathbb{E}_{\mathbf{X},\mathbf{y}}\Big[\frac{1}{n}\sum_{j=1}^n \ell\Big(f_M^{\mathbf{W}}\left(\mathbf{x}_{0:N}^{(j)};\mathbf{V}\right),y_j\Big)\Big] - \mathbb{E}_{\mathbf{X},\mathbf{y},\mathbf{W}}\Big[\frac{1}{n}\sum_{j=1}^n \ell\Big(f_M^{\mathbf{W}}\left(\mathbf{x}_{0:N}^{(j)};\mathbf{V}\right),y_j\Big)\Big]\Big|$$

$$\lesssim \sqrt{\frac{K_2\log(6/\delta)}{M}} + K_1\sqrt{\frac{\log(dM)\log(nNM)}{n}}. \tag{36}$$

The proofs of Lemma B.3, B.4, B.5, and B.6 are contained in section B.2.1. Now assuming they hold, we proceed to prove Proposition B.1.

*Proof of Proposition B.1.*

Split the left-hand side of inequality (31), we have

$$\sup_{\mathbf{V}\in\mathcal{V}_M}\Big|\frac{1}{n}\sum_{j=1}^n \ell\Big(f_M^{\mathbf{W}}\left(\mathbf{x}_{0:N}^{(j)};\mathbf{V}\right),y_j\Big) - \mathbb{E}_{\mathbf{X},\mathbf{y}}\Big(\ell\Big(f_M^{\mathbf{W}}\left(\mathbf{x}_{0:N}^{(j)};\mathbf{V}\right),y\Big)\Big)\Big|$$

$$\leq \sup_{\mathbf{V}\in\mathcal{V}_M}\Big|\frac{1}{n}\sum_{j=1}^n \ell(f_M^{\mathbf{W}},y_j) - \mathbb{E}_{\mathbf{W}}\Big[\frac{1}{n}\sum_{j=1}^n \ell(f_M^{\mathbf{W}},y_j)\Big]\Big|$$

$$+ \sup_{\mathbf{V}\in\mathcal{V}_M}\Big|\mathbb{E}_{\mathbf{W}}\Big[\frac{1}{n}\sum_{j=1}^n \ell(f_M^{\mathbf{W}},y_j)\Big] - \mathbb{E}_{\mathbf{X},\mathbf{y},\mathbf{W}}\Big(\ell(f_M^{\mathbf{W}},y)\Big)\Big|$$

$$+ \sup_{\mathbf{V}\in\mathcal{V}_M}\Big|\mathbb{E}_{\mathbf{X},\mathbf{y}}\Big(\ell(f_M^{\mathbf{W}},y)\Big) - \mathbb{E}_{\mathbf{X},\mathbf{y},\mathbf{W}}\Big(\ell(f_M^{\mathbf{W}},y)\Big)\Big|$$

$$\lesssim K_1\sqrt{\frac{\log(dM)\log(nNM)}{n}} + \sqrt{\log\left(\frac{6}{\delta}\right)}\left(\frac{K_1}{\sqrt{n}} + \sqrt{\frac{K_2}{M}}\right)$$

with probability at least $1-\delta$. Here the last inequality uses Lemma B.4, B.5, and B.6. This proves Proposition B.1. $\qquad\square$

### B.2.1 Proof of auxiliary lemmas

*Proof of Lemma B.3.*

First using Rademacher contraction inequality, since $\ell(0, y) \leq 1$, we can center it and only pay an extra term $1/\sqrt{n}$ in the Rademacher complexity. Then by the Rademacher contraction property (Theorem A.4), the problem boils down to bounding the Rademacher complexity of $f_M^{\mathbf{W}}$, which is

$$\mathbb{E}_{\mathbf{X},\mathbf{y},\mathbf{W},\boldsymbol{\xi}}\Big[\sup_{\mathbf{V}\in\mathcal{V}_M}\Big|\frac{1}{n}\sum_{j=1}^{n}\xi_j f_M^{\mathbf{W}}\big(\mathbf{x}_{0:N}^{(j)};\mathbf{V}\big)\Big|\Big].$$

Fix $\mathbf{X}$, $\mathbf{y}$, and $\mathbf{W}$, we have

$$\mathbb{E}_{\boldsymbol{\xi}}\Big[\sup_{\mathbf{V}\in\mathcal{V}_M}\Big|\frac{1}{n}\sum_{j=1}^{n}\xi_j f_M^{\mathbf{W}}\big(\mathbf{x}_{0:N}^{(j)};\mathbf{V}\big)\Big|\Big]$$

$$=\mathbb{E}_{\boldsymbol{\xi}}\Big[\sup_{\mathbf{V}\in\mathcal{V}_M}\Big|\sum_{m=1}^{M}\Big\langle\mathbf{v}_m,\frac{1}{n}\sum_{j=1}^{n}\xi_j\Big[\frac{1}{N}\sum_{i=1}^{N}\overline{\sigma}(\mathbf{W}_m,\widetilde{\mathbf{x}}_0^{(j)}\widetilde{\mathbf{x}}_i^{(j)\top})\widetilde{\mathbf{x}}_i^{(j)}\Big]\Big\rangle\Big|\Big]$$

$$\leq K_1\mathbb{E}_{\boldsymbol{\xi}}\Big[\max_m\Big\|\frac{1}{n}\sum_{j=1}^{n}\xi_j\Big[\frac{1}{N}\sum_{i=1}^{N}\overline{\sigma}(\mathbf{W}_m,\widetilde{\mathbf{x}}_0^{(j)}\widetilde{\mathbf{x}}_i^{(j)\top})\widetilde{\mathbf{x}}_i^{(j)}\Big]\Big\|_2\Big]. \tag{37}$$

By matrix Bernstein inequality (Theorem A.3), for any fixed $m$,

$$\mathbb{P}\Big[\Big\|\frac{1}{n}\sum_{j=1}^{n}\xi_j\Big[\frac{1}{N}\sum_{i=1}^{N}\overline{\sigma}(\mathbf{W}_m,\widetilde{\mathbf{x}}_0^{(j)}\widetilde{\mathbf{x}}_i^{(j)\top})\widetilde{\mathbf{x}}_i^{(j)}\Big]\Big\|_2\geq\varepsilon\Big]\leq 2d\exp\Big(-\frac{n\varepsilon^2/2}{A^2+K\varepsilon/3}\Big),$$

where

$$A=\max_m\sqrt{\frac{1}{nN^2}\sum_{i,j,k}\Big[\overline{\sigma}(\mathbf{W}_m,\widetilde{\mathbf{x}}_0^{(j)}\widetilde{\mathbf{x}}_i^{(j)\top})\overline{\sigma}(\mathbf{W}_m,\widetilde{\mathbf{x}}_0^{(j)}\widetilde{\mathbf{x}}_k^{(j)})\big\langle\widetilde{\mathbf{x}}_i^{(j)},\widetilde{\mathbf{x}}_k^{(j)}\big\rangle\Big]}$$

$$\lesssim\max_{m,i,j}\overline{\sigma}(\mathbf{W}_m,\widetilde{\mathbf{x}}_0^{(j)}\widetilde{\mathbf{x}}_i^{(j)\top}),\text{ and}$$

$$K=\max_{i,m}\Big\|\frac{1}{N}\sum_{i=1}^{N}\overline{\sigma}(\mathbf{W}_m,\widetilde{\mathbf{x}}_0^{(j)}\widetilde{\mathbf{x}}_i^{(j)\top})\widetilde{\mathbf{x}}_i^{(j)}\Big\|_2.$$

Using the union bound, we have

$$\mathbb{P}\Big[\max_{m\in[M]}\Big\|\frac{1}{n}\sum_{j=1}^{n}\xi_j\Big[\frac{1}{N}\sum_{i=1}^{N}\overline{\sigma}(\mathbf{W}_m,\widetilde{\mathbf{x}}_0^{(j)}\widetilde{\mathbf{x}}_i^{(j)\top})\widetilde{\mathbf{x}}_i^{(j)}\Big]\Big\|_2\geq\varepsilon\Big]\leq 2dM\exp\Big(-\frac{n\varepsilon^2}{A^2+K\varepsilon/3}\Big).$$

Therefore, we can bound its expectation with

$$\mathbb{E}_{\boldsymbol{\xi}}\Big[\max_m\Big\|\frac{1}{n}\sum_{j=1}^{n}\xi_j\Big[\frac{1}{N}\sum_{i=1}^{N}\overline{\sigma}(\mathbf{W}_m,\widetilde{\mathbf{x}}_0^{(j)}\widetilde{\mathbf{x}}_i^{(j)\top})\widetilde{\mathbf{x}}_i^{(j)}\Big]\Big\|_2\Big]\lesssim\Big[\sqrt{\frac{\log(dM)}{n}}A+\frac{\log(dM)}{n}K\Big]. \tag{38}$$

Now take expectation over $\mathbf{X}$, $\mathbf{y}$, and $\mathbf{W}$. Since $n\geq\log(dM)$, we have

$$\mathbb{E}_{\mathbf{X},\mathbf{y},\mathbf{W}}\Big[\sqrt{\frac{\log(dM)}{n}}A+\frac{\log(dM)}{n}K\Big]$$

$$\lesssim\Big(\sqrt{\frac{\log(dM)}{n}}+\frac{\log(dM)}{n}\Big)\mathbb{E}_{\mathbf{X},\mathbf{y},\mathbf{W}}\Big[\max_{i,j,m}\big|\overline{\sigma}\big(\big\langle\mathbf{W}_m,\widetilde{\mathbf{x}}_0^{(j)}\widetilde{\mathbf{x}}_i^{(j)\top}\big\rangle\big)\big|\Big]$$

$$\lesssim\sqrt{\frac{\log(dM)\log(nNM)}{n}}.$$

Combine this with Eq. (37) and (38), we prove (32).

Fixing any $\mathbf{X}$ and $\mathbf{y}$, only taking expectation over $\mathbf{W}$, we get

$$\mathbb{E}_{\mathbf{W}}\Big[\sqrt{\frac{\log(dM)}{n}}A+\frac{\log(dM)}{n}K\Big]\lesssim\sqrt{\frac{\log(dM)\log(nNM)}{n}}.$$

Combine this with Eq. (37) and (38), we prove (33). This finishes the proof of Lemma B.3. $\qquad\square$

*Proof for Lemma B.4.*

Denote $\mathbf{X} = \{\mathbf{x}_{0:N}^{(j)}\}_{j \in [n]}$ and $\mathbf{y} = \{y_j\}_{j \in [n]}$ and denote

$$g(\mathbf{W}_{1:M}; \mathbf{X}, \mathbf{y}) = \sup_{\mathbf{V} \in \mathcal{V}_M} \left| \frac{1}{n} \sum_{j=1}^{n} \ell\left( f_M^{\mathbf{W}}\left( \mathbf{x}_{0:N}^{(j)}; \mathbf{V} \right), y_j \right) - \mathbb{E}_{\mathbf{W}} \left[ \frac{1}{n} \sum_{j=1}^{n} \ell\left( f_M^{\mathbf{W}}\left( \mathbf{x}_{0:N}^{(j)}; \mathbf{V} \right), y_j \right) \right] \right|.$$

Given $\mathbf{W}_{1:M} = \{\mathbf{W}_m\}_{m=1}^{M}$ and $\mathbf{W}_{1:M}' = \{\mathbf{W}_m'\}_{m=1}^{M}$, we define $\|\mathbf{W}_{1:M} - \mathbf{W}_{1:M}'\|_{\mathsf{Fr}} = (\sum_{m \in [M]} \|\mathbf{W}_m - \mathbf{W}_m'\|_{\mathsf{Fr}}^2)^{1/2}$. Then we have

$$g(\mathbf{W}_{1:M}; \mathbf{X}, \mathbf{y}) - g(\mathbf{W}_{1:M}'; \mathbf{X}, \mathbf{y})$$

$$\leq \sup_{\mathbf{V} \in \mathcal{V}_M} \left| \frac{1}{n} \sum_{j=1}^{n} \left[ \ell(f_M^{\mathbf{W}}, y_j) - \ell\left( f_M^{\mathbf{W}'}, y_j \right) \right] \right|$$

$$\leq \sup_{\mathbf{V} \in \mathcal{V}_M} \left| \frac{1}{Nn} \sum_{i,j,m} \left[ \overline{\sigma}(\mathbf{W}, \widetilde{\mathbf{x}}_0^{(j)} \widetilde{\mathbf{x}}_i^{(j)\top}) - \overline{\sigma}(\mathbf{W}', \widetilde{\mathbf{x}}_0^{(j)} \widetilde{\mathbf{x}}_i^{(j)\top}) \right] \langle \mathbf{v}_m, \widetilde{\mathbf{x}}_i \rangle \right|$$

$$\lesssim \sup_{\mathbf{V} \in \mathcal{V}_M} \left| \sum_m \|\mathbf{W}_m - \mathbf{W}_m'\|_{\mathsf{Fr}} \|\mathbf{v}_m\|_2 \right|$$

$$\lesssim \|\mathbf{W}_{1:M} - \mathbf{W}_{1:M}'\|_{\mathsf{Fr}} \sqrt{\frac{K_2}{M}}.$$

Since $\mathbf{W}_m$ has independent standard Gaussian entries, by Gaussian concentration inequality (Theorem A.1), we have that

$$\mathbb{P}\left[ \left| g(\mathbf{W}_{1:M}; \mathbf{X}, \mathbf{y}) - \mathbb{E}_{\mathbf{W}}\left[ g(\mathbf{W}_{1:M}; \mathbf{X}, \mathbf{y}) \mid \mathbf{X}, \mathbf{y} \right] \right| \geq \varepsilon \mid \mathbf{X}, \mathbf{y} \right] \leq 2 \exp\left( -\frac{M\varepsilon^2}{4K_2} \right). \quad (39)$$

For the conditional expectation, by symmetrization, we have

$$\mathbb{E}_{\mathbf{W}}\left[ g(\mathbf{W}_{1:M}; \mathbf{X}, \mathbf{y}) \mid \mathbf{X}, \mathbf{y} \right]$$

$$= \mathbb{E}_{\mathbf{W}}\left[ \sup_{\mathbf{V} \in \mathcal{V}_M} \left| \sum_{j=1}^{n} \ell\left( f_M^{\mathbf{W}}\left( \mathbf{x}_{0:N}^{(j)}; \mathbf{V} \right), y_j \right) - \mathbb{E}_{\mathbf{W}}\left[ \ell\left( f_M^{\mathbf{W}}\left( \mathbf{x}_{0:N}^{(j)}; \mathbf{V} \right), y_j \right) \right] \right| \mid \mathbf{X}, \mathbf{y} \right]$$

$$\leq 2 \mathbb{E}_{\mathbf{W}, \boldsymbol{\xi}}\left[ \sup_{\mathbf{V} \in \mathcal{V}_M} \left| \sum_{j=1}^{n} \xi_j \ell\left( f_M^{\mathbf{W}}\left( \mathbf{x}_{0:N}^{(j)}; \mathbf{V} \right), y_j \right) \right| \mid \mathbf{X}, \mathbf{y} \right]$$

$$\lesssim K_1 \sqrt{\frac{\log(dM) \log(nNM)}{n}} \quad (40)$$

for any $\mathbf{X}$ and $\mathbf{y}$, where the last inequality uses Lemma B.3. Combining (39) and (40) and taking $\varepsilon = 2\sqrt{K_2 \log(6/\delta)/M}$, we have

$$\sup_{\mathbf{V} \in \mathcal{V}_M} \left| \frac{1}{n} \sum_{j=1}^{n} \ell\left( f_M^{\mathbf{W}}\left( \mathbf{x}_{0:N}^{(j)}; \mathbf{V} \right), y_j \right) - \mathbb{E}_{\mathbf{W}}\left[ \frac{1}{n} \sum_{j=1}^{n} \ell\left( f_M^{\mathbf{W}}\left( \mathbf{x}_{0:N}^{(j)}; \mathbf{V} \right), y_j \right) \right] \right|$$

$$\lesssim \sqrt{\frac{K_2}{M}} \sqrt{\log \frac{6}{\delta}} + K_1 \sqrt{\frac{\log(dM) \log(nNM)}{n}}, \quad (41)$$

for any $\mathbf{X}$ and $\mathbf{y}$. Since the right-hand side is irrelevant to $\mathbf{X}$ and $\mathbf{y}$, we get (34). This proves Lemma B.4. $\qquad \square$

*Proof of Lemma B.5.*

Let

$$h(\mathbf{x}_{0:N}, \mathbf{y}) = \sup_{\mathbf{V} \in \mathcal{V}_M} \left| \mathbb{E}_{\mathbf{W}}\left[ \frac{1}{n} \sum_{j=1}^{n} \ell(f_M^{\mathbf{W}}, y_j) \right] - \mathbb{E}_{\mathbf{X}, \mathbf{y}, \mathbf{W}}\left[ \frac{1}{n} \sum_{j=1}^{n} \ell(f_M^{\mathbf{W}}, y_j) \right] \right|. \quad (42)$$

For each $i \in [n]$, let $\{\mathbf{X}', \mathbf{y}'\}$ differs with $\{\mathbf{X}, \mathbf{y}\}$ only on $i$-th data point. We have

$$h(\mathbf{X}, \mathbf{y}) - h(\mathbf{X}', \mathbf{y}')$$

$$\leq \sup_{\mathbf{V} \in \mathcal{V}_M} \Big| \frac{1}{n} \sum_{j=1}^{n} \Big[ \mathbb{E}_{\mathbf{W}} \Big[ \ell \Big( f_M^{\mathbf{W}} \big( \mathbf{x}_{0:N}^{(j)}; \mathbf{V} \big), y_j \Big) \Big] - \mathbb{E}_{\mathbf{W}} \Big[ \ell \Big( f_M^{\mathbf{W}} \big( \mathbf{x}_{0:N}^{(j)\prime}; \mathbf{V} \big), y_j' \Big) \Big] \Big] \Big|$$

$$\leq \sup_{\mathbf{V} \in \mathcal{V}_M} \mathbb{E}_{\mathbf{W}} \Big[ \frac{1}{n} \sum_{j=1}^{n} \Big| f_M^{\mathbf{W}} \big( \mathbf{x}_{0:N}^{(j)}; \mathbf{V} \big) - f_M^{\mathbf{W}} \big( \mathbf{x}_{0:N}^{(j)\prime}; \mathbf{V} \big) \Big| \Big]$$

$$\leq \sup_{\mathbf{V} \in \mathcal{V}_M} \mathbb{E}_{\mathbf{W}} \Big[ \frac{1}{nN} \sum_{m \in [M], k \in [N]} \Big[ \Big| \overline{\sigma}(\mathbf{W}_m, \widetilde{\mathbf{x}}_0^{(i)} \widetilde{\mathbf{x}}_k^{(i)\top}) \Big| + \Big| \overline{\sigma}(\mathbf{W}_m, \widetilde{\mathbf{x}}_0^{(i)\prime} \widetilde{\mathbf{x}}_k^{(i)\prime\top}) \Big| \Big] \|\mathbf{v}_m\|_2 \Big].$$

$$\lesssim \frac{K_1}{n}.$$

Therefore, $h(\mathbf{x}_{0:N}, \mathbf{y})$ satisfies the bounded difference property with the parameter $\{L_i\}_{i=1}^{n}$ uniformly bounded by $\Theta(K_1/n)$. By bounded difference inequality (Theorem A.2), there's a constant $\widetilde{C}$ such that

$$\mathbb{P}\Big[ \Big| h(\mathbf{X}, \mathbf{y}) - \mathbb{E}_{\mathbf{X}} \Big[ h(\mathbf{X}, \mathbf{y}) \Big] \Big| \geq \varepsilon \Big] \leq 2 \exp\left( -\frac{\widetilde{C} n \varepsilon^2}{K_1^2} \right).$$

Combining with Lemma B.3, we have

$$\mathbb{E}_{\mathbf{X}, \mathbf{y}} \Big[ h(\mathbf{X}, \mathbf{y}) \Big]$$

$$= \mathbb{E}_{\mathbf{X}, \mathbf{y}} \Big[ \sup_{\mathbf{V} \in \mathcal{V}_M} \Big| \mathbb{E}_{\mathbf{W}} \Big[ \frac{1}{n} \sum_{j=1}^{n} \ell(f_M^{\mathbf{W}}, y_j) \Big] - \mathbb{E}_{\mathbf{X}, \mathbf{y}, \mathbf{W}} \Big[ \frac{1}{n} \sum_{j=1}^{n} \ell(f_M^{\mathbf{W}}, y_j) \Big] \Big| \Big]$$

$$\leq 2 \mathbb{E}_{\mathbf{X}, \mathbf{y}, \boldsymbol{\xi}} \Big[ \sup_{\mathbf{V} \in \mathcal{V}_M} \Big| \frac{1}{n} \sum_{j=1}^{n} \xi_j \mathbb{E}_{\mathbf{W}} \Big[ \ell \Big( f_M^{\mathbf{W}} \big( \mathbf{x}_{0:N}^{(j)}; \mathbf{V} \big), y_j \Big) \Big] \Big| \Big]$$

$$\lesssim K_1 \sqrt{\frac{\log(dM) \log(nNM)}{n}}.$$

Therefore, by taking $\varepsilon = 2K_1 [\log(6/\delta)/(n\widetilde{C})]^{1/2}$, we have

$$\sup_{\mathbf{V} \in \mathcal{V}_M} \Big| \mathbb{E}_{\mathbf{W}} \Big[ \frac{1}{n} \sum_{j=1}^{n} \ell(f_M^{\mathbf{W}}, y_j) \Big] - \mathbb{E}_{\mathbf{X}, \mathbf{y}, \mathbf{W}} \Big[ \frac{1}{n} \sum_{j=1}^{n} \ell(f_M^{\mathbf{W}}, y_j) \Big] \Big|$$

$$\lesssim K_1 \sqrt{\frac{\log(dM) \log(nNM)}{n}} + K_1 \sqrt{\frac{\log(1/\delta)}{n}}$$

with probability at least $1 - \delta$. This proves Lemma B.5. $\qquad \square$

*Proof of Lemma B.6.*

Denote

$$\varphi(\mathbf{W}_{1:M}) = \sup_{\mathbf{V} \in \mathcal{V}_M} \Big| \mathbb{E}_{\mathbf{X}, \mathbf{y}} \Big[ \frac{1}{n} \sum_{j=1}^{n} \ell(f_M^{\mathbf{W}}, y_j) \Big] - \mathbb{E}_{\mathbf{X}, \mathbf{y}, \mathbf{W}} \Big[ \frac{1}{n} \sum_{j=1}^{n} \ell(f_M^{\mathbf{W}}, y_j) \Big] \Big|.$$

Similar to the proof of Lemma B.4. Given $\mathbf{W}_{1:M} = \{\mathbf{W}_m\}_{m=1}^{M}$ and $\mathbf{W}_{1:M}' = \{\mathbf{W}_m'\}_{m=1}^{M}$, define $\|\mathbf{W}_{1:M} - \mathbf{W}_{1:M}'\|_{\mathsf{Fr}} = \sqrt{\sum_m \|\mathbf{W}_m - \mathbf{W}_m'\|_{\mathsf{Fr}}^2}$. We have

$$\varphi(\mathbf{W}_{1:M}) - \varphi(\mathbf{W}_{1:M}')$$

$$\leq \sup_{\mathbf{V} \in \mathcal{V}_M} \Big| \frac{1}{n} \sum_{j=1}^{n} \mathbb{E}_{\mathbf{X}, \mathbf{y}} \Big[ \ell(f_M^{\mathbf{W}}, y_j) - \ell\Big( f_M^{\mathbf{W}'}, y_j \Big) \Big] \Big|$$

$$\leq \sup_{\mathbf{V} \in \mathcal{V}_M} \Big| \frac{1}{Nn} \sum_{i,j,m} \mathbb{E}_{\mathbf{X}, \mathbf{y}} \Big\{ \Big[ \overline{\sigma}(\mathbf{W}, \widetilde{\mathbf{x}}_0^{(j)} \widetilde{\mathbf{x}}_i^{(j)\top}) - \overline{\sigma}(\mathbf{W}', \widetilde{\mathbf{x}}_0^{(j)} \widetilde{\mathbf{x}}_i^{(j)\top}) \Big] \langle \mathbf{v}_m, \widetilde{\mathbf{x}}_i \rangle \Big\} \Big|$$

$$\leq \sup_{\mathbf{V} \in \mathcal{V}_M} \left| \sum_m 2\sqrt{2} \|\mathbf{W}_m - \mathbf{W}'_m\|_{\mathsf{Fr}} \|\mathbf{v}_m\|_2 \right|$$

$$\lesssim \|\mathbf{W}_{1:M} - \mathbf{W}_{1:M}\|_{\mathsf{Fr}} \sqrt{\frac{K_2}{M}}.$$

Since $\mathbf{W}_m$ has independent standard Gaussian entries, there is a constant $\widetilde{C}$ s.t.

$$\mathbb{P}\left[ \left| \varphi(\mathbf{W}_{1:M}) - \mathbb{E}_{\mathbf{W}}\left[\varphi(\mathbf{W}_{1:M})\right] \right| \geq \varepsilon \right] \leq 2 \exp\left( -\frac{\widetilde{C} M \varepsilon^2}{K_2} \right). \tag{43}$$

For the expectation, we have

$$\mathbb{E}_{\mathbf{W}}\left[ \varphi(\mathbf{W}_{1:M}) \right] = \mathbb{E}_{\mathbf{W}}\left[ \sup_{\mathbf{V} \in \mathcal{V}_M} \left| \frac{1}{n} \sum_{j=1}^n \ell\left( f_M^{\mathbf{W}}\left(\mathbf{x}_{0:N}^{(j)}; \mathbf{V}\right), y_j \right) - \mathbb{E}_{\mathbf{X},\mathbf{y},\mathbf{w}}\left[ \frac{1}{n} \sum_{j=1}^n \ell(f_M^{\mathbf{W}}, y_j) \right] \right| \right]$$

$$\leq 2\mathbb{E}_{\mathbf{X},\mathbf{y},\mathbf{w},\boldsymbol{\xi}}\left[ \sup_{\mathbf{V} \in \mathcal{V}_M} \left| \frac{1}{n} \sum_{j=1}^n \xi_j \ell\left( f_M^{\mathbf{W}}\left(\mathbf{x}_{0:N}^{(j)}; \mathbf{V}\right), y_j \right) \right| \right]$$

$$\lesssim K_1 \sqrt{\frac{\log(dM) \log(nNM)}{n}}, \tag{44}$$

where the last inequality is by Lemma B.3. Combining (43) and (44) and taking $\varepsilon = [K_2 \log(6/\delta)/(M\widetilde{C})]^{1/2}$, we get

$$\sup_{\mathbf{V} \in \mathcal{V}_M} \left| \mathbb{E}_{\mathbf{X},\mathbf{y}}\left[ \frac{1}{n} \sum_{j=1}^n \ell(f_M^{\mathbf{W}}, y_j) \right] - \mathbb{E}_{\mathbf{X},\mathbf{y},\mathbf{w}}\left[ \frac{1}{n} \sum_{j=1}^n \ell(f_M^{\mathbf{W}}, y_j) \right] \right|$$

$$\lesssim \sqrt{\frac{K_2 \log(1/\delta)}{M}} + K_1 \sqrt{\frac{\log(dM) \log(nNM)}{n}}$$

with probability at least $1 - \delta$. This proves Lemma B.6. $\qquad\square$

### B.3 Proof of Theorem 2

*Proof.* Given any target function $f_\star$, by Lemma B.1, with probability larger than $1 - \delta/2$ over $\mathbf{W}_{1:M}$, there exists $\widetilde{\mathbf{V}} = \{\widetilde{\mathbf{v}}_m\}_{m=1}^M$ such that

$$\|f_\star - f_M^{\mathbf{W}}(\mathbf{x}_{0:N}; \widetilde{\mathbf{V}})\|_{L^2(P)} \lesssim \sqrt{\frac{(d^2 + \log M) B(f_\star) \delta^{-1}}{M}}, \tag{45}$$

with

$$\sum_{m=1}^M \left\|\widetilde{\mathbf{v}}_m\right\|_2 \lesssim \sqrt{B(f_\star)} + \sqrt{\frac{B(f_\star)\delta^{-1}}{M}} \lesssim 2\sqrt{B(f_\star)} \text{ and } \sum_{m=1}^M \left\|\widetilde{\mathbf{v}}_m\right\|_2^2 \lesssim \frac{B(f_\star)\delta^{-1}}{M}. \tag{46}$$

By our choice of $K_1$ and $K_2$, let $f_{\widehat{\mathbf{v}},M}^{\mathbf{W}} = f_M^{\mathbf{W}}(\cdot; \widehat{\mathbf{V}})$ denote the model trained by (11), and let

$$f_{\mathbf{v}^*,M}^{\mathbf{W}} = \arg\min_{\mathbf{V} \in \mathcal{V}_M} L_D\left( f_M^{\mathbf{W}}(\cdot; \mathbf{V}) \right) \text{ and } f_{\widetilde{\mathbf{v}},M}^{\mathbf{W}} = f_M^{\mathbf{W}}(\mathbf{x}_{0:N}; \widetilde{\mathbf{V}}).$$

Then with probability at least $1 - \delta$ over $\mathbf{W}$, $\mathbf{X}$, and $\mathbf{y}$, we have

$$L_D\left(f_{\widehat{\mathbf{v}},M}^{\mathbf{W}}\right) - L_D(f^*) \tag{47}$$

$$\leq L_D\left(f_{\widehat{\mathbf{v}},M}^{\mathbf{W}}\right) - \widehat{L}_D\left(f_{\widehat{\mathbf{v}},M}^{\mathbf{W}}\right) + \widehat{L}_D\left(f_{\widehat{\mathbf{v}},M}^{\mathbf{W}}\right) - \widehat{L}_D\left(f_{\mathbf{v}^*,M}^{\mathbf{W}}\right) + \widehat{L}_D\left(f_{\mathbf{v}^*,M}^{\mathbf{W}}\right)$$

$$- L_D\left(f_{\mathbf{v}^*,M}^{\mathbf{W}}\right) + L_D\left(f_{\mathbf{v}^*,M}^{\mathbf{W}}\right) - L_D\left(f_{\widetilde{\mathbf{v}},M}^{\mathbf{W}}\right) + L_D\left(f_{\widetilde{\mathbf{v}},M}^{\mathbf{W}}\right) - L_D(f_\star) \tag{48}$$

$$\leq L_D\left(f_{\widehat{\mathbf{v}},M}^{\mathbf{W}}\right) - \widehat{L}_D\left(f_{\widehat{\mathbf{v}},M}^{\mathbf{W}}\right) + \widehat{L}_D\left(f_{\mathbf{v}^*,M}^{\mathbf{W}}\right) - L_D\left(f_{\mathbf{v}^*,M}^{\mathbf{W}}\right) + L_D\left(f_{\widetilde{\mathbf{v}},M}^{\mathbf{W}}\right) - L_D(f^*) \tag{49}$$

$$\leq 2 \sup_{f \in \mathcal{V}_M} \left| L_D(f) - \widehat{L}_D(f) \right| + \|f_{\widetilde{\mathbf{v}},M}^{\mathbf{W}} - f_\star\|_{L^2(P)} \tag{50}$$

$$\lesssim K_1 \sqrt{\frac{\log(dM)\log(nNM)}{n}} + \sqrt{\log\left(\frac{1}{\delta}\right)}\left(\frac{K_1}{\sqrt{n}} + \sqrt{\frac{K_2}{M}}\right) + \sqrt{\frac{(d^2 + \log M)B(f_\star)\delta^{-1}}{M}}$$
(51)

$$\lesssim \sqrt{\frac{B(f_\star)}{n}}\left(\sqrt{\log(dM)\log(nNM)} + \sqrt{\log(\delta^{-1})}\right) + \sqrt{\frac{(d^2+\log M)B(f_\star)\delta^{-1}}{M}}$$
(52)

$$\lesssim \sqrt{\frac{B(f_\star)[\log(dM)\log(nNM) + \log(\delta^{-1})]}{n}} + \sqrt{\frac{(d^2+\log M)B(f_\star)\delta^{-1}}{M}},$$
(53)

where from (48) to (49) we use the definition of $f_{\widehat{\mathbf{v}},M}^{\mathbf{W}}$ and $f_{\mathbf{v}^*,M}^{\mathbf{W}}$. From (49) to (50), we bound ≪Yu notes: fix≫ $L_D(f_{\widehat{\mathbf{v}},M}^{\mathbf{W}}) - \widehat{L}_D(f_{\widehat{\mathbf{v}},M}^{\mathbf{W}})$ and $L_D(f_{\mathbf{v}^*,M}^{\mathbf{W}}) - \widehat{L}_D(f_{\mathbf{v}^*,M}^{\mathbf{W}})$ by $\sup_f |L_D(f) - \widehat{L}_D(f)|$ and use the Lipschitzness of $\ell(f,y)$. From (50) to (51), we use Proposition B.1 and Lemma B.1. From (51) to (52), we insert the value of $K_1$ and $K_2$ into the equation. This proves Theorem 2. □

## B.4  Proof of Examples in Section 3

### B.4.1  Excess risk of RFMLP

Denote $\mathcal{F}$ to be the set of all functions in the function class (6) and (7), i.e.,

$$\mathcal{F} = \left\{ f_\star(\mathbf{x}_{0:N}) = \frac{1}{N}\sum_{i=1}^{N}\sum_{r,s\geq 0}^{\infty} \left\langle \mathbf{x}_0^{\otimes r}\otimes\mathbf{x}_i^{\otimes s}, \mathbf{f}_{rs}\right\rangle : \mathbf{f}_{rs}\in\mathbb{R}^{d^{r+s}} \text{ symmetric}, r,s\geq 0 \right\}.$$
(54)

Consider the RFMLP model

$$f_M^{\text{MLP}}(\mathbf{x}_{0:N};\mathbf{v}) = \sum_{m=1}^{M}\sigma\left(\left\langle \mathbf{w}_m, \text{vec}(\mathbf{x}_{0:N})\right\rangle\right)\cdot v_m, \quad \{\mathbf{w}_m\}_{m\in[M]}\sim_{\text{iid}}\mathsf{N}(\mathbf{0},\mathbf{I}/(N+2)),$$
(55)

where $\text{vec}(\mathbf{x}_{0:N}) = [\mathbf{x}_0;\mathbf{x}_1;\ldots;\mathbf{x}_N;1]\in\mathbb{R}^{dN+d+1}$. For target functions that take forms in (6) and (7), define $B_{\text{MLP}}(f_\star) = \sum_{k=0}^{\infty}\widetilde{C}_k\sum_{r+s=k}\|\mathbf{f}_{rs}\|_{\text{Fr}}^2$ with $\widetilde{C}_k = k^{3.5}(N+2)^{2k}$. In case where $f_\star$ admits multiple representations of the form (7), $B_{\text{MLP}}(f_\star)$ is the infimum of the right-hand-side over all such representations.

Then we consider the empirical risk minimizer over the RFMLP model:

$$\widehat{\mathbf{v}} = \arg\min_{\mathbf{v}\in\mathcal{V}_M}\widehat{L}_D(f_M^{\text{MLP}}(\cdot;\mathbf{v})), \qquad \widehat{L}_D(f) = \frac{1}{n}\sum_{j=1}^{n}\ell(f(\mathbf{x}_{0:N}^{(j)}), y_j),$$
(56)

where the constrained class $\mathcal{V}_M^{\text{MLP}}$ gives

$$\mathcal{V}_M^{\text{MLP}} = \left\{\mathbf{v} = \{v_m\}_{m=1}^{M} : \sum_{m=1}^{M}|v_m|\leq K_1, \sum_{m=1}^{M}v_m^2\leq K_2/M\right\}.$$
(57)

**Proposition B.2** (The sample complexity of RFMLP). *Let $f_\star = \arg\min_{f\in\mathcal{F}}L_D(f)$ be the population risk minimizer within the target function class* (54). *Assume $M > \delta^{-1}$ and $n > \log(dM)$. Take $K_1 = C\sqrt{B_{\text{MLP}}(f_\star)}$ and $K_2 = CB_{\text{MLP}}(f_\star)\delta^{-1}$ in* (57), *with $C$ being a constant. Let $\widehat{f}_M^{\text{MLP}} = f_M^{\text{MLP}}(\cdot;\widehat{\mathbf{v}})$ be empirical risk minimizer of model RFMLP. Then for any joint distribution $\mathsf{P}$, with probability at least $1-\delta$ over $\{\mathbf{w}_m\}_{m\in[M]}$ sampled according to* (55) *and $\{(\mathbf{x}_{0:N}^{(j)}, y_j)\}_{j\in[n]}\sim_{iid}\mathsf{P}$, the excess risk is bounded by*

$$L_D(\widehat{f}_M^{\text{MLP}}) - L_D(f_\star) \leq \widetilde{\mathcal{O}}\left(\sqrt{B_{\text{MLP}}(f_\star)}\left[\sqrt{\frac{1}{n}} + \sqrt{\frac{d^2\delta^{-1}}{M}}\right]\right).$$
(58)

*Proof of Proposition B.2.* The proof is basically the same as that of RFA model. We only give a sketch of the proof. Firstly, with a few modifications of the proof, we can show that Lemma B.1 also holds for RFMLP, with $B(f_\star)$ replaced with $B_{\text{MLP}}(f_\star)$. Lemma B.2 is slightly different, we have the kernel expansion:

$$K(\mathbf{x}_{0:N}, \mathbf{x}_{0:N}') = \sum_{\ell\in\{0,1\}\cup\{2k\}_{k\geq 1}} \left\langle \sqrt{c_\ell}(N+2)^{-\ell}\left[\text{vec}(\mathbf{x}_{0:N})\right]^{\otimes\ell}, \sqrt{c_\ell}(N+2)^{-\ell}\left[\text{vec}(\mathbf{x}_{0:N}')\right]^{\otimes\ell}\right\rangle.$$

Therefore we can rewrite $f_\star$ as

$$f_\star = \sum_{\ell \in \{0,1\} \cup \{2k\}_{k \geq 1}} \left\langle \frac{(N+2)^\ell}{N\sqrt{c_\ell}} \sum_{i=1}^{N} \sum_{r+s=\ell} \widetilde{\mathbf{f}}_{rs,i}, \sqrt{c_\ell}(N+2)^{-\ell}\left[\text{vec}(\mathbf{x}'_{0:N})\right]^{\otimes \ell} \right\rangle, \quad (59)$$

where $\langle \widetilde{\mathbf{f}}_{rs,i}, [\text{vec}(\mathbf{x}'_{0:N})]^{\otimes \ell} \rangle = \langle \mathbf{f}_{rs}, \mathbf{x}_0^{\otimes r} \otimes \mathbf{x}_i^{\otimes s} \rangle$. Thus,

$$\|f_\star\|_{\mathcal{H}_K}^2 \leq \sum_{k=0}^{\infty} \widetilde{C}_k \sum_{r+s=k} \|\mathbf{f}_{rs}\|_{\mathsf{Fr}}^2 \quad \text{with} \quad \widetilde{C}_k = k^{3.5}(N+2)^{2k}.$$

The right-hand-side gives the formulation of $B_{\text{MLP}}(f_\star)$. Then, a similar version of Proposition B.1 holds for RFMLP model $f_M^{\mathbf{W}}$, i.e., with probability at least $1 - \delta$ (w.r.t. $\mathbf{W}$, $\mathbf{y}$, and $\mathbf{X}$),

$$\sup_{\mathbf{V} \in \mathcal{V}_M} \left| \frac{1}{n} \sum_{j=1}^{n} \ell\left(f_M^{\mathbf{W}}\left(\mathbf{x}_{0:N}^{(j)}; \mathbf{V}\right), y_j\right) - \mathbb{E}_{(\mathbf{x}_{0:N}, y) \sim \mathsf{P}} \ell(f_M^{\mathbf{W}}(\mathbf{x}_{0:N}; \mathbf{V}), y) \right|$$

$$\lesssim K_1 \sqrt{\frac{(N+2)\log(dM)\log(nNM)}{n}} + \sqrt{(N+2)\log(1/\delta)}\left(\frac{K_1}{\sqrt{n}} + \sqrt{\frac{K_2}{M}}\right). \quad (60)$$

Then combining all of the above equations and following the proof in Section B.2, we get (58). $\square$

**Remark 1.** *The representation in (59) is not unique. With a more careful choice of the representation of the target function $f_\star$, we can get a better bound for $B_{\text{MLP}}(f_\star)$, which is given by*

$$B_{\text{MLP}}(f_\star) = \sum_{k=0}^{\infty} \widetilde{C}_k \sum_{r+s=k} \|\mathbf{f}_{rs}\|_{\mathsf{Fr}}^2 \quad \text{with} \quad \widetilde{C}_k = k^{3.5}[(N+2)/2]^{2k}. \quad (61)$$

### B.4.2 Proofs of Examples

**Proposition B.3** (Restatement of Example 1). *For functions of $\mathbf{x}_0$ of the form*

$$f_\star(\mathbf{x}_{0:N}) = \sum_{k=0}^{\infty} \langle \mathbf{x}_0^{\otimes k}, \mathbf{A}_k \rangle, \quad \mathbf{A}_k \in \mathbb{R}^{d^k},$$

*we have that $B(f_\star) = \sum_{k=0}^{\infty} C_k \|\mathbf{A}_k\|_{\mathsf{Fr}}^2$. Setting $M = \Theta(d^2 n)$, the excess risk bound gives $\widetilde{\mathcal{O}}(\sqrt{\sum_{k=0}^{\infty} k^{4.5} 4^k \|\mathbf{A}_k\|_{\mathsf{Fr}}^2/n})$. Moreover, consider $f_\star(\mathbf{x}_{0:N}) = (\boldsymbol{\beta}^\top \mathbf{x}_0)^p$. The above excess risk of RFA model and the RFMLP model scales as*

$$\text{RFA} : \widetilde{\mathcal{O}}\left(\text{Poly}(p)\sqrt{4^p \|\boldsymbol{\beta}\|_2^{2p}/n}\right), \qquad \text{RFMLP} : \widetilde{\mathcal{O}}\left(\text{Poly}(p)\sqrt{((N+2))^p \|\boldsymbol{\beta}\|_2^{2p}/n}\right).$$

*Proof of Proposition B.3.* This follows by direct calculation. $\square$

**Proposition B.4** (Restatement of Example 2). *For $f_\star = \frac{1}{N} \sum_{i=1}^{N} \psi(\langle \boldsymbol{\beta}, \mathbf{x}_i \rangle)$ with $\psi(z) = z \arctan(z/\eta)$ with $\eta > 2$ and $\|\boldsymbol{\beta}\|_2 = 1$. The excess risk bound of RFA model and the RFMLP model are*

$$\text{RFA} : \widetilde{\mathcal{O}}\left(\sqrt{\sum_{k=1}^{\infty} k^{4.5}(2/\eta)^{2k}/n}\right) = \widetilde{\mathcal{O}}(\sqrt{1/n}), \quad \text{RFMLP} : \widetilde{\mathcal{O}}\left(\sqrt{\sum_{k=1}^{\infty} k^{4.5}[(N+2)/2\eta]^{2k}/n}\right).$$

*Proof of Proposition B.4.* Use the power series expansion of $\psi$, we have

$$f_\star = \frac{1}{N} \sum_{i=1}^{N} \sum_{k=1}^{\infty} (-1)^k \frac{\langle \boldsymbol{\beta}, \mathbf{x}_i \rangle^{2k}}{(2k-1)\eta^{2k-1}}.$$

Plug it into the formula of $B(f_\star)$ (10) and $B_{\text{MLP}}(f_\star)$ (61), we get

$$B(f_\star) = \sum_{k=1}^{\infty} (2k)^{4.5} 4^{2k} \|\boldsymbol{\beta}/\eta\|_2^{4k} = \Theta\left(\sum_{k=1}^{\infty} k^{4.5}(2/\eta)^{2k}\right),$$

and $B_{\text{MLP}}(f_\star) = \Theta(\sum_{k=1}^{\infty} k^{4.5}[(N+2)/2\eta]^{2k})$. Therefore, by Theorem 2 and Proposition B.2, we get their excess risk. This proves Proposition B.4. $\square$

**Proposition B.5** (Restatement of Example 3). *For* $f_{1,\star} = \frac{1}{N}\sum_{i=1}^{N}\langle \mathbf{x}_0, \mathbf{x}_i\rangle^p$, *the excess risk bound of* RFA *(by Theorem 2) and* RFMLP *scale as*

$$\text{RFA} : \widetilde{\mathcal{O}}\Big(\text{Poly}(p)\sqrt{(4d)^p/n}\Big), \qquad \text{RFMLP} : \widetilde{\mathcal{O}}\Big(\text{Poly}(p)\sqrt{[(N+2)d]^p/n}\Big).$$

*For* $f_{2,\star} = \frac{1}{N}\sum_{i=1}^{N}\cos(\langle \mathbf{x}_0, \mathbf{x}_i\rangle)\langle \mathbf{x}_i^{\otimes p}, \mathbf{G}\rangle$ *with* $\|\mathbf{G}\|_{\mathsf{Fr}} = 1$. *Then the excess risk bound of* RFA *and* RFMLP *scale as*

$$\text{RFA} : \widetilde{\mathcal{O}}\Big(\text{Poly}(pd)\sqrt{e^{4\sqrt{d}}4^p/n}\Big), \qquad \text{RFMLP} : \widetilde{\mathcal{O}}\Big(\text{Poly}(pNd)\sqrt{e^{2(N+2)d}(N+2)^p/n}\Big).$$

*Proof of Proposition B.5.* For $f_{1,\star}$, direct calculation gives the value of $B(f_{1,\star})$ and the excess risk follows. For $f_{2,\star}$, using the Taylor expansion of $\cos(z)$, we get

$$f_{2,\star} = \frac{1}{N}\sum_{i=1}^{N}\sum_{k=0}^{\infty}(-1)^k\frac{\langle \mathbf{x}_0, \mathbf{x}_i\rangle^{2k}}{(2k)!}\langle \mathbf{x}_i^{\otimes p}, \mathbf{G}\rangle = \frac{1}{N}\sum_{i=1}^{N}\sum_{k=0}^{\infty}\Big\langle (\mathbf{x}_0\mathbf{x}_i^{\top})^{\otimes 2k}\otimes \mathbf{x}_i^{\otimes p}, \frac{(-1)^k}{(2k)!}\mathbf{I}_d^{\otimes 2k}\otimes \mathbf{G}\Big\rangle.$$

Plug it into the formula of $B(f_\star)$ (10) and $B_{\text{MLP}}(f_\star)$ (61), we get

$$B(f_{2,\star}) = 4^p\text{Poly}(p)\sum_{k=0}^{\infty}(2k)^{4.5}4^{2k}\frac{d^{2k}}{((2k)!)^2} = 4^p\text{Poly}(pd)\Theta\Big(\sum_{k=0}^{\infty}\frac{(4d)^{2k}}{((2k)!)^2}\Big).$$

Note that for any $z > 0$,

$$\sum_{k=0}^{\infty}\frac{(z)^{2k}}{((2k)!)^2} \leq \Big[\sum_{k=0}^{\infty}\frac{(\sqrt{z})^{2k}}{(2k)!}\Big]^2 = \Theta\Big(e^{2\sqrt{z}}\Big). \tag{62}$$

Plug $z = 4d$ into (62) for $B(f_{2,\star})$ gives the excess risk for RFA model. As for $B_{\text{MLP}}(f_{2,\star})$, we have

$$B_{\text{MLP}}(f_{2,\star}) = \text{Poly}(p)\sum_{k=0}^{\infty}(N+2)^p(4k)^{4.5}(N+2)^{4k}\frac{d^{2k}}{((2k)!)^2} = 4^p\text{Poly}(pdN)\Theta\Big(\sum_{k=0}^{\infty}\frac{((N+2)\sqrt{d})^{4k}}{((2k)!)^2}\Big).$$

Using similar argument, we get that $B_{\text{MLP}}(f_{2,\star})$ is bounded by $\Theta(\text{Poly}(pdN)4^p\exp(2(N+2)\sqrt{d}))$. Using Theorem 2 and Proposition B.2, we can get the excess risk bound. This proves Proposition B.5. $\qquad\square$

## B.5 Random Feature Attention with Exponential Activation

We give a brief discussion on our results with activation function replaced by the exponential function $\sigma'(t) = \exp(t)$. The random feature attention model with exponential activation (we call it ERFA) is given as follows.

$$f_M^{\mathbf{W}}(\mathbf{x}_{0:N}; \mathbf{V}) = \sum_{m=1}^{M}\frac{1}{N}\sum_{i=1}^{N}\sigma'\big(\big\langle \mathbf{W}_m, \widetilde{\mathbf{x}}_0\widetilde{\mathbf{x}}_i^{\top}\big\rangle\big)\langle \mathbf{v}_m, \widetilde{\mathbf{x}}_i\rangle. \tag{63}$$

We consider deriving the explicit form of the Kernel $K_{\text{ERFA}}(\mathbf{x}_{0:N}, \mathbf{x}'_{0:N})$ associated with ERFA model

$$K_{\text{ERFA}}(\mathbf{x}_{0:N}, \mathbf{x}'_{0:N}) = \frac{1}{N^2}\mathbb{E}_{\mathbf{W}}\Big[\sum_{i,j=1}^{N}\exp(\langle \mathbf{W}, \widetilde{\mathbf{x}}_0\widetilde{\mathbf{x}}_i^{\top} + \widetilde{\mathbf{x}}'_0(\widetilde{\mathbf{x}}'_j)^{\top}\rangle)\langle \widetilde{\mathbf{x}}_i, \widetilde{\mathbf{x}}'_j\rangle\Big] \tag{64}$$

For any $i, j \in [N]$, let $u_{i,j} = \big\langle \widetilde{\mathbf{x}}_0\widetilde{\mathbf{x}}_i^{\top}, \widetilde{\mathbf{x}}'_0(\widetilde{\mathbf{x}}'_j)^{\top}\big\rangle/4$. For a single component in (64), we have

$$\mathbb{E}_{\mathbf{W}\sim\mathsf{N}(0,1/4)}\Big[\exp(\langle \mathbf{W}, \widetilde{\mathbf{x}}_0\widetilde{\mathbf{x}}_i^{\top} + \widetilde{\mathbf{x}}'_0(\widetilde{\mathbf{x}}'_j)^{\top}\rangle)\Big]\langle \widetilde{\mathbf{x}}_i, \widetilde{\mathbf{x}}'_j\rangle$$

$$= \exp\Big(\frac{1}{8}\big\|\widetilde{\mathbf{x}}_0\widetilde{\mathbf{x}}_i^{\top} + \widetilde{\mathbf{x}}'_0(\widetilde{\mathbf{x}}'_j)^{\top}\big\|_{\mathcal{F}}^2\Big)\langle \widetilde{\mathbf{x}}_i, \widetilde{\mathbf{x}}'_j\rangle$$

$$= \exp\big(1 + u_{i,j}\big)\langle \widetilde{\mathbf{x}}_i, \widetilde{\mathbf{x}}'_j\rangle$$

$$= e\Big(\sum_{\ell=0}^{\infty}\frac{1}{\ell!}u_{i,j}^{\ell}\Big)\langle \widetilde{\mathbf{x}}_i, \widetilde{\mathbf{x}}'_j\rangle$$

$$= \sum_{\ell=0}^{\infty} d_\ell \left\langle \widetilde{\mathbf{x}}_0 \widetilde{\mathbf{x}}_i^\top, \widetilde{\mathbf{x}}_0'(\widetilde{\mathbf{x}}_j')^\top \right\rangle^\ell 4^{-\ell} \langle \widetilde{\mathbf{x}}_i, \widetilde{\mathbf{x}}_j' \rangle$$

$$= \sum_{\ell=0}^{\infty} d_\ell \left\langle 2^{-\ell}(\widetilde{\mathbf{x}}_0 \widetilde{\mathbf{x}}_i^\top)^{\otimes \ell} \otimes \widetilde{\mathbf{x}}_i, 2^{-\ell}(\widetilde{\mathbf{x}}_0'(\widetilde{\mathbf{x}}_j')^\top)^{\otimes \ell} \otimes \widetilde{\mathbf{x}}_j' \right\rangle.$$

Here $d_\ell = e/\ell!$ for any $\ell \geq 0$. Therefore, the kernel can be expressed as:

$$K_{\texttt{ERFA}}(\mathbf{x}_{0:N}, \mathbf{x}_{0:N}') = \sum_{\ell=0}^{\infty} \left\langle \frac{\sqrt{d_\ell}}{N} \sum_{i=1}^{N} 2^{-\ell}(\widetilde{\mathbf{x}}_0 \widetilde{\mathbf{x}}_i^\top)^{\otimes \ell} \otimes \widetilde{\mathbf{x}}_i, \frac{\sqrt{d_\ell}}{N} \sum_{j=1}^{N} 2^{-\ell}(\widetilde{\mathbf{x}}_0'(\widetilde{\mathbf{x}}_j')^\top)^{\otimes \ell} \otimes \widetilde{\mathbf{x}}_j' \right\rangle.$$

Thus, for any target function $f_\star$ that in the form of (6), we can redefine its complexity measure as

$$B(f_\star) = \sum_{k=0}^{\infty} C_k \sum_{\max\{r,s\}=k} \|\mathbf{f}_{rs}\|_{\mathsf{Fr}}^2, \qquad C_k = k! 4^k k^2 \vee 1 \tag{65}$$

and obtain the corresponding results of Theorem 1 and Theorem 2 by repeating the proof of these two theorems.

## C  Proofs for Section 4

We consider the empirical risk minimizer over the $\texttt{BRFA}$ model (15),

$$\widehat{\mathbf{V}} = \arg\min_{\mathbf{V} \in \mathcal{V}_M} \widehat{L}_D(f_M^{\mathbf{W},\mathbf{W}_0}(\cdot\,; \mathbf{V})), \qquad \widehat{L}_D(f) = \tfrac{1}{n} \sum_{j=1}^{n} \ell(f(\mathbf{x}_{0:N}^{(j)}), y_j), \tag{66}$$

where the constrained class $\mathcal{V}_M$ gives (12), copied here for reader's convenience,

$$\mathcal{V}_M = \left\{ \mathbf{V} = \{\mathbf{v}_m\}_{m=1}^{M} :\ \sum_{m=1}^{M} \|\mathbf{v}_m\|_2 \leq K_1, \sum_{m=1}^{M} \|\mathbf{v}_m\|_2^2 \leq K_2/M \right\}. \tag{67}$$

Denote $\mathcal{G}$ to be the set of all functions in the function class (16), i.e.,

$$\mathcal{G} = \left\{ g_\star = \frac{1}{N} \sum_{i=1}^{N} F(\langle \mathbf{x}_0, \mathbf{x}_i \rangle) G(\mathbf{x}_0, \mathbf{x}_i) : F(t) = \sum_{k=0}^{\infty} a_k t^k, G = \langle \widetilde{\mathbf{x}}_i^{\otimes 3} \otimes \widetilde{\mathbf{x}}_0^{\otimes 2}, \mathbf{A}_\star \rangle \right\}. \tag{68}$$

We restate Theorem 3 in Theorem C.1 with detailed assumptions.

**Theorem C.1** (Restatement of Theorem 3)**.** *Assume $M > \delta^{-1}$, $n > \log(dM)$ and let $L \in \mathbb{Z}_{\geq 1}$. Let $g_\star = \arg\min_{g \in \mathcal{G}} L_D(g)$ be the population risk minimizer within the target function class $\mathcal{G}$ (68). Take $K_1 = C\sqrt{B(g_\star, L)}$ and $K_2 = CB(g_\star, L)\delta^{-1}$ in (67), with $C$ being a constant. Let $\widehat{f}_M^{\mathbf{W},\mathbf{W}_0} = f_M^{\mathbf{W},\mathbf{W}_0}(\cdot\,; \widehat{\mathbf{V}})$ be empirical risk minimizer given by (66). Then for any joint distribution $\mathsf{P}$, with probability at least $1 - \delta$ over $\{\mathbf{W}_m\}_{m \in [M]}$ sampled according to (4) and $\{(\mathbf{x}_{0:N}^{(j)}, y_j)\}_{j \in [n]} \sim_{iid} \mathsf{P}$, the excess risk is bounded by*

$$L_D(\widehat{f}_M^{\mathbf{W},\mathbf{W}_0}) - L_D(g_\star)$$

$$\lesssim \inf_L \left\{ \sqrt{B(g_\star, L)} \left[ \sqrt{\frac{\log(dM)\log(nNM) + \log(\delta^{-1})}{n}} + \sqrt{\frac{(d^2 + \log M)\delta^{-1}}{M}} \right] + \varepsilon_L \|g_\star\|_\infty \right\},$$

*where $\varepsilon_L = 1/[2^{L+1}(L+1)!]$ and*

$$B(g_\star, L) = \|\mathbf{A}_\star\|_{\mathsf{Fr}}^2 \cdot \left( \sum_{k=0}^{\infty} |a_k| \cdot C_k \right)^2, \quad \text{with } C_k = (2L+k)^{(k+3)/2} 8^{L+k/2}. \tag{69}$$

### C.1  Auxiliary results for the proof of Theorem 3

In this section, we give some auxiliary results used in the proof of Theorem 3. The proof will be given in Section C.2. We define the biased transformer with infinite width (informally corresponding to Eq. (15) with $M \to \infty$), given by

$$f_{\mathbf{v}}^{\mathbf{W}_0}(\mathbf{x}_{0:N}) = \frac{1}{N} \sum_{i=1}^{N} \mathbb{E}_{\mathbf{W}} \left[ \sigma\left( \langle \mathbf{W} + \mathbf{W}_0, \widetilde{\mathbf{x}}_0 \widetilde{\mathbf{x}}_i^\top \rangle \right) \langle \mathbf{v}(\mathbf{W}), \widetilde{\mathbf{x}}_i \rangle \right] \tag{70}$$

$$= \frac{1}{N} \sum_{i=1}^{N} \mathbb{E}_{\mathbf{W}} \big[ \sigma\big( \langle \mathbf{W}, \widetilde{\mathbf{x}}_0 \widetilde{\mathbf{x}}_i^\top \rangle + h_i \big) \langle \mathbf{v}(\mathbf{W}), \widetilde{\mathbf{x}}_i \rangle \big],$$

where $h_i := \langle \mathbf{W}_0, \widetilde{\mathbf{x}}_0 \widetilde{\mathbf{x}}_i^\top \rangle$ and $\mathbf{W}$ is sampled according to (4). Here we set $\mathbf{W}_0 = [\mathbf{I}_{d \times d}, \mathbf{0}_{d \times 1}; \mathbf{0}_{1 \times d}, 0]$. Then $h_i = \langle \mathbf{x}_0, \mathbf{x}_i \rangle$.

We consider a class of target functions $\widetilde{g}_\star : \mathcal{X} \to \mathbb{R}$ that takes form

$$\widetilde{g}_\star(\mathbf{x}_{0:N}) = \frac{1}{N} \sum_{i=1}^{N} \sum_{\ell=0}^{\infty} \langle u_\ell(\widetilde{\mathbf{x}}_0, \widetilde{\mathbf{x}}_i), \mathbf{D}_\ell \rangle, \tag{71}$$

for some coefficients $\{\mathbf{D}_\ell \in \mathbb{R}^{(d+1)^{2\ell+1}}\}_{\ell \geq 0}$ and

$$u_\ell(\widetilde{\mathbf{x}}_0, \widetilde{\mathbf{x}}_i) = \begin{cases} [h_i \Phi(h_i) + \phi(h_i)] \widetilde{\mathbf{x}}_i & (\ell = 0), \\ \Phi(h_i) \widetilde{\mathbf{x}}_0 \widetilde{\mathbf{x}}_i^\top \otimes \widetilde{\mathbf{x}}_i & (\ell = 1), \\ \phi(h_i) \mathrm{He}_{\ell-2}(h_i) (\widetilde{\mathbf{x}}_0 \widetilde{\mathbf{x}}_i^\top)^{\otimes \ell} \otimes \widetilde{\mathbf{x}}_i & (\ell \geq 2), \end{cases}$$

where $\phi$ and $\Phi$ are the PDF and CDF of the standard Gaussian random variable, respectively. Lemma C.1 below provides a counterpart of Lemma B.2 for the BRFA model and the target function (71).

**Lemma C.1.** *Any function $\widetilde{g}_\star$ of form (71) can be expressed exactly as an infinite-head random feature attention model (70)*

$$\widetilde{g}_\star(\mathbf{x}_{0:N}) = f_{\mathbf{v}}^{\mathbf{W}_0}(\mathbf{x}_{0:N}) = \frac{1}{N} \sum_{i=1}^{N} \mathbb{E}_{\mathbf{W}} \big[ \sigma\big( \langle \mathbf{W} + \mathbf{W}_0, \widetilde{\mathbf{x}}_0 \widetilde{\mathbf{x}}_i^\top \rangle \big) \langle \mathbf{v}(\mathbf{W}), \widetilde{\mathbf{x}}_i \rangle \big], \tag{72}$$

*where the coefficients $\mathbf{v}(\cdot)$ satisfy*

$$\mathbb{E}_{\mathbf{W}} \big[ \|\mathbf{v}(\mathbf{W})\|_2^2 \big] \leq \sum_{\ell \geq 0} 4^\ell \ell! \, \|\mathbf{D}_\ell\|_{\mathsf{Fr}}^2.$$

Given Lemma C.1, we can get a counterpart of Theorem 1 for the BRFA model as Proposition C.1 below.

**Proposition C.1** (Counterpart of Theorem 1 for BRFA)**.** *Suppose function $\widetilde{g}_\star : \mathcal{X} \to \mathbb{R}$ takes form (71). Then for any input distribution $P$ on $\mathcal{X}$, with probability at least $1 - \delta$ (over $\{\mathbf{W}_m\}_{m \in [M]}$ sampled from (4)), there exists an $M$-head BRFA model (15) with coefficients $\mathbf{V} = \{\mathbf{v}_m\}_{m \in [M]} \subseteq \mathbb{R}^{d+1}$ that approximates $\widetilde{g}_\star$ in $L^2(P)$ up to error*

$$\mathbb{E}_{\mathbf{x}_{0:N} \sim P} \big[ \big( \widetilde{g}_\star(\mathbf{x}_{0:N}) - f_M^{\mathbf{W}}(\mathbf{x}_{0:N}; \mathbf{V}) \big)^2 \big] \leq \mathcal{O}\bigg( \frac{(d^2 + \log M) B(\widetilde{g}_\star) \delta^{-1}}{M} \bigg). \tag{73}$$

*In addition, the norms of the weight of this random-feature attention model are bounded as*

$$\sum_{m=1}^{M} \|\mathbf{v}_m\|_2 \leq \mathcal{O}\bigg( \sqrt{B(\widetilde{g}_\star)} + \sqrt{\frac{B(\widetilde{g}_\star) \delta^{-1}}{M}} \bigg), \qquad \sum_{m=1}^{M} \|\mathbf{v}_m\|_2^2 \leq \mathcal{O}\bigg( \frac{B(\widetilde{g}_\star) \delta^{-1}}{M} \bigg). \tag{74}$$

*Here $B(\widetilde{g}_\star)$ is defined alternatively as*

$$B(\widetilde{g}_\star) = \sum_{k=0}^{\infty} C_k \|\mathbf{D}_k\|_{\mathsf{Fr}}^2, \quad \text{with} \quad C_k = k! 4^k \vee 1. \tag{75}$$

Furthermore, we could approximate the target function in the form (16) to any precision by a function that takes form (72), which is discussed in Lemma C.2.

**Lemma C.2.** *For any target function $g_\star$ in the form of (16), and for any precision $\varepsilon_\ell := \frac{1}{2^{\ell+1}(\ell+1)!}$, there exists a function $f_{\mathbf{v}}^{\mathbf{W}_0}$ in the form of (72) such that*

$$\big\| g_\star - f_{\mathbf{v}}^{\mathbf{W}_0} \big\|_\infty \leq \|g_\star\|_\infty \, \varepsilon_\ell,$$

*and*

$$\mathbb{E}_{\mathbf{W}} \big[ \|\mathbf{v}(\mathbf{W})\|_2^2 \big] \leq \bigg( \sum_{p=0}^{\infty} a_p 8^{\ell + \frac{p+3}{2}} (2\ell + p)^{\frac{p+3}{2}} \bigg)^2 \|\mathbf{A}_\star\|_{\mathsf{Fr}}^2,$$

*where the tensor $\mathbf{A}_\star \in \mathbb{R}^{d^5}$ parameterizes $g_\star$.*

### C.1.1 Proof of auxiliary lemmas

*Proof of Lemma C.1.*

The function (72) can be interpreted as a linear function of the feature map

$$\Psi(\mathbf{x}_{0:N}) = \frac{1}{N}\sum_{i=1}^{N}\sigma\big(\langle\mathbf{W}, \widetilde{\mathbf{x}}_0\widetilde{\mathbf{x}}_i^\top\rangle + h_i\big)\widetilde{\mathbf{x}}_i.$$

So the kernel w.r.t. the feature map takes the form that

$$
\begin{aligned}
&K(\mathbf{x}_{0:N}, \mathbf{x}'_{0:N})\\
&= \mathbb{E}_{\mathbf{W}}[\langle\Psi(\mathbf{x}_{0:N}), \Psi(\mathbf{x}'_{0:N})\rangle]\\
&= \frac{1}{N^2}\sum_{1\le i,j\le N}\mathbb{E}_{\mathbf{W}}\big[\sigma\big(\langle\mathbf{W}, \widetilde{\mathbf{x}}_0\widetilde{\mathbf{x}}_i^\top\rangle + h_i\big)\sigma\big(\langle\mathbf{W}, \widetilde{\mathbf{x}}'_0\widetilde{\mathbf{x}}'^\top_j\rangle + h'_j\big)\big]\langle\widetilde{\mathbf{x}}_i, \widetilde{\mathbf{x}}'_j\rangle,
\end{aligned}
$$

where $h_i = \langle\mathbf{x}_0, \mathbf{x}_i\rangle$ and $h'_j = \langle\mathbf{x}'_0, \mathbf{x}'_j\rangle$. Similar to the proof of Lemma B.2, we also consider a single component of the equation above first, which has the form of

$$\mathbb{E}_{\mathbf{W}}\big[\sigma\big(\langle\mathbf{W}, \widetilde{\mathbf{x}}_0\widetilde{\mathbf{x}}_i^\top\rangle + h_i\big)\sigma\big(\langle\mathbf{W}, \widetilde{\mathbf{x}}'_0\widetilde{\mathbf{x}}'^\top_j\rangle + h'_j\big)\big]\langle\widetilde{\mathbf{x}}_i, \widetilde{\mathbf{x}}'_j\rangle. \tag{76}$$

We expand $\sigma\big(\langle\mathbf{W}, \widetilde{\mathbf{x}}_0\widetilde{\mathbf{x}}_i^\top\rangle + h_i\big)$ by Hermite polynomials in the space $L^2(\mathbb{R}, \phi)$ using Lemma A.3,

$$
\begin{aligned}
&\mathbb{E}_{\mathbf{W}}\big[\sigma\big(\langle\mathbf{W}, \widetilde{\mathbf{x}}_0\widetilde{\mathbf{x}}_i^\top\rangle + h_i\big)\mathrm{He}_\ell\big(\langle\mathbf{W}, \widetilde{\mathbf{x}}_0\widetilde{\mathbf{x}}_i^\top\rangle\big)\big]\\
&= \mathbb{E}_{z\sim\mathsf{N}(0,1)}[\sigma(z + h_i)\mathrm{He}_\ell(z)]\\
&= \begin{cases} h_i\Phi(h_i) + \phi(h_i) & (\ell = 0),\\ \Phi(h_i) & (\ell = 1),\\ (-1)^\ell\phi(h_i)\mathrm{He}_{\ell-2}(h_i) & (\ell \ge 2). \end{cases}
\end{aligned}
$$

Therefore, we have

$$
\begin{aligned}
\sigma\big(\langle\mathbf{W}, \widetilde{\mathbf{x}}_0\widetilde{\mathbf{x}}_i^\top\rangle + h_i\big) ={}& h_i\Phi(h_i) + \phi(h_i) + \Phi(h_i)\mathrm{He}_1\big(\langle\mathbf{W}, \widetilde{\mathbf{x}}_0\widetilde{\mathbf{x}}_i^\top\rangle\big)\\
&+ \sum_{\ell=2}^{\infty}\frac{(-1)^\ell}{\ell!}\phi(h_i)\mathrm{He}_{\ell-2}(h_i)\mathrm{He}_\ell\big(\langle\mathbf{W}, \widetilde{\mathbf{x}}_0\widetilde{\mathbf{x}}_i^\top\rangle\big).
\end{aligned}
$$

Using Lemma A.1, we obtain the expansion:

$$
\begin{aligned}
&\mathbb{E}_{\mathbf{W}}\big[\sigma\big(\langle\mathbf{W}, \widetilde{\mathbf{x}}_0\widetilde{\mathbf{x}}_i^\top\rangle + h_i\big)\sigma\big(\langle\mathbf{W}, \widetilde{\mathbf{x}}'_0\widetilde{\mathbf{x}}'^\top_j\rangle + h'_j\big)\big]\langle\widetilde{\mathbf{x}}_i, \widetilde{\mathbf{x}}'_j\rangle\\
&= [h_i\Phi(h_i) + \phi(h_i)]\big[h'_j\Phi(h'_j) + \phi(h'_j)\big]\langle\widetilde{\mathbf{x}}_i, \widetilde{\mathbf{x}}'_j\rangle\\
&\quad + 2^{-2}\Phi(h_i)\Phi(h'_j)\big\langle\widetilde{\mathbf{x}}_0\widetilde{\mathbf{x}}_i^\top, \widetilde{\mathbf{x}}'_0\widetilde{\mathbf{x}}'^\top_j\big\rangle\langle\widetilde{\mathbf{x}}_i, \widetilde{\mathbf{x}}'_j\rangle\\
&\quad + \sum_{\ell=2}^{\infty}\bigg\{\frac{1}{\ell!}2^{-2\ell}\phi(h_i)\phi(h'_j)\mathrm{He}_{\ell-2}(h_i)\mathrm{He}_{\ell-2}\big(h'_j\big)\big\langle\widetilde{\mathbf{x}}_0\widetilde{\mathbf{x}}_i^\top, \widetilde{\mathbf{x}}'_0\widetilde{\mathbf{x}}'^\top_j\big\rangle^\ell\bigg\}\langle\widetilde{\mathbf{x}}_i, \widetilde{\mathbf{x}}'_j\rangle\\
&= \sum_{\ell=0}^{\infty}\big\langle\varphi_\ell(\widetilde{\mathbf{x}}_0, \widetilde{\mathbf{x}}_i), \varphi_\ell(\widetilde{\mathbf{x}}'_0, \widetilde{\mathbf{x}}'_j)\big\rangle,
\end{aligned}
$$

where

$$\varphi_\ell(\widetilde{\mathbf{x}}_0, \widetilde{\mathbf{x}}_i) = \sqrt{\frac{1}{\ell!}}2^{-\ell}u_\ell(\widetilde{\mathbf{x}}_0, \widetilde{\mathbf{x}}_i) = \begin{cases} [h_i\Phi(h_i) + \phi(h_i)]\widetilde{\mathbf{x}}_i & (\ell = 0),\\ 2^{-1}\Phi(h_i)\widetilde{\mathbf{x}}_0\widetilde{\mathbf{x}}_i^\top\otimes\widetilde{\mathbf{x}}_i & (\ell = 1),\\ \sqrt{\frac{1}{\ell!}}2^{-\ell}\phi(h_i)\mathrm{He}_{\ell-2}(h_i)(\widetilde{\mathbf{x}}_0\widetilde{\mathbf{x}}_i^\top)^{\otimes\ell}\otimes\widetilde{\mathbf{x}}_i & (\ell \ge 2). \end{cases}$$

Then by taking summation with respect to $i$ and $j$, we derive the expansion of the kernel:

$$K(\mathbf{x}_{0:N}, \mathbf{x}'_{0:N}) = \sum_{\ell=0}^{\infty}\Big\langle\frac{1}{N}\sum_{i=1}^{N}\varphi_\ell(\mathbf{x}_0, \mathbf{x}_i), \frac{1}{N}\sum_{j=1}^{N}\varphi_\ell(\mathbf{x}'_0, \mathbf{x}'_j)\Big\rangle.$$

Then for the target function that takes the form in (71), we have the RKHS norm of $\widetilde{g}_\star$ bounded by:

$$\|\widetilde{g}_\star\|_{\mathcal{H}_K}^2 \leq \sum_{\ell=0}^\infty \left(2^\ell \sqrt{\ell!}\right)^2 \|\mathbf{D}_\ell\|_{\mathsf{Fr}}^2 = \sum_{\ell=0}^\infty 4^\ell \ell! \, \|\mathbf{D}_\ell\|_{\mathsf{Fr}}^2$$

by the feature equivalence property (23). Thus, there exists coefficients $\mathbf{v}(\mathbf{W})$ such that

$$f_{\mathbf{v}}^{\mathbf{W}_0}(\mathbf{x}_{0:N}) = \frac{1}{N} \sum_{i=1}^N \mathbb{E}_{\mathbf{W}}\left[\sigma\left(\langle \mathbf{W} + \mathbf{W}_0, \widetilde{\mathbf{x}}_0 \widetilde{\mathbf{x}}_i^\top \rangle\right) \langle \mathbf{v}(\mathbf{W}), \widetilde{\mathbf{x}}_i \rangle\right]$$

and

$$\mathbb{E}_{\mathbf{W}}\left[\|\mathbf{v}(\mathbf{W})\|_2^2\right] \leq \sum_{\ell \geq 0} 4^\ell \ell! \, \|\mathbf{D}_\ell\|_{\mathsf{Fr}}^2,$$

which proves Lemma C.1. $\qquad\square$

*Proof of Proposition C.1.*

Note that Lemma B.1 is also applicable to the model BRFA. Combining it with Lemma C.1 proves Proposition C.1. $\qquad\square$

*Proof of Lemma C.2.*

Firstly, we consider approximating $x^p$ using the function class $\{\phi(x)\mathrm{He}_\ell(x)\}_{n\geq 0}$, which is equivalent to approximating $x^p \phi^{-1}(x)$ using Hermite polynomials. We take $p = 0$ first and compute the Hermite expansion of the $2\ell$-th-order term in the Taylor expansion of $\phi^{-1}(x) = e^{x^2/2}$, i.e. $\psi_\ell(x) = \sum_{n=0}^\ell \frac{x^{2n}}{2^n n!}$. Using Lemma A.2, we have that:

$$\begin{aligned}
\psi_\ell(x) &= \sum_{n=0}^\ell \frac{x^{2n}}{2^n n!} = \sum_{n=0}^\ell \frac{(2n)!}{2^n n!} \sum_{m=0}^n \frac{\mathrm{He}_{2n-2m}(x)}{2^m m! (2n - 2m)!} \\
&= \sum_{k=0}^\ell \mathrm{He}_{2k}(x) \sum_{m=0}^{\ell-k} \frac{(2m+2k)!}{2^{m+k}(m+k)!} \cdot \frac{1}{2^m m!(2k)!} \\
&=: \sum_{k=0}^\ell c_{\ell,k} \mathrm{He}_{2k}(x),
\end{aligned}$$

where $c_{\ell,k} = \frac{2^k}{(2k)!} \sum_{m=0}^{\ell-k} \frac{(2m+2k)!}{2^{2m+2k}(m+k)!m!}$. Similarly, for any $p \geq 0$, define $p_0 = \lfloor \frac{p}{2} \rfloor$ with $r = p - 2p_0$. We have that

$$\begin{aligned}
\psi_\ell(x)x^p &= \sum_{n=0}^\ell \frac{x^{2n+2p_0+r}}{2^n n!} = \sum_{n=0}^\ell \frac{(2n+2p_0+r)!}{2^n n!} \sum_{m=0}^{n+p_0} \frac{\mathrm{He}_{2n+2p_0+r-2m}(x)}{2^m m!(2n+2p_0+r-2m)!} \\
&= \sum_{k=0}^{\ell+p_0} \mathrm{He}_{2k+r}(x) \sum_{m=\max(p_0-k,0)}^{\ell+p_0-k} \frac{(2m+2k+r)!}{2^{m+k-p_0}(m+k-p_0)!} \cdot \frac{1}{2^m m!(2k+r)!} \\
&=: \sum_{k=0}^{\ell+p_0} c_{\ell,k,p_0} \mathrm{He}_{2k+r}(x),
\end{aligned}$$

where $c_{\ell,k,p_0} = \sum_{m=\max(p_0-k,0)}^{\ell+p_0-k} \frac{(2m+2k+r)!}{2^{m+k-p_0}(m+k-p_0)!} \cdot \frac{1}{2^m m!(2k+r)!}$. Then we can bound $c_{\ell,k,p_0}$ as follows:

$$\begin{aligned}
c_{\ell,k,p_0} &= \sum_{m=\max(p_0-k,0)}^{\ell+p_0-k} \frac{(2m+2k+r)!}{2^{m+k-p_0}(m+k-p_0)!} \cdot \frac{1}{2^m m!(2k+r)!} \\
&\leq \sum_{m=0}^{\ell+p_0-k} 2^{p_0} \frac{(m+k)!}{(m+k-p_0)!} \frac{(2m+2k+r)!}{2^{m+k}(m+k)!} \cdot \frac{1}{2^m m!(2k+r)!}
\end{aligned}$$

$$\leq 2^{p_0}(\ell+p_0)^{p_0}(2l+p)^r \sum_{m=0}^{\ell+p_0-k} \frac{(2m+2k)!}{2^{m+k}(m+k)!} \cdot \frac{1}{2^m m!(2k)!} \leq (2\ell+p)^{p_0+r} c_{\ell+p_0,k}.$$

Next, we give an upper bound of $c_{\ell,k}$:

$$c_{\ell,k} = \frac{2^k}{(2k)!} \sum_{m=0}^{\ell-k} \frac{(2m+2k)!}{2^{2m+2k}(m+k)!m!}$$

$$= \frac{2^k}{(2k)!} \sum_{m=0}^{\ell-k} \frac{(2m+2k)!}{2^{2m+2k}(m+k)!(m+k)!} \cdot \frac{(m+k)!}{m!}$$

$$\leq \frac{2^k}{(2k)!} \sum_{m=0}^{\ell-k} \sqrt{\frac{1}{m+k}} \cdot \frac{(m+k)!}{m!}$$

$$= \frac{2^k}{(2k)!} \sum_{m=0}^{\ell-k} \sqrt{m+k} \cdot \frac{(m+k-1)!}{m!}$$

$$\leq \frac{2^k}{(2k)!} \sqrt{\ell} \sum_{m=0}^{\ell-k} \frac{(m+k-1)!}{m!}$$

$$= \frac{2^k}{(2k)!} \sqrt{\ell} \frac{\ell!}{k(\ell-k)!}.$$

Here we use the inequality that

$$\sqrt{\frac{2}{\pi(2n+1)}} \leq \frac{(2n)!}{2^{2n}n!n!} \leq \sqrt{\frac{1}{2n}}, \forall n \geq 1.$$

and we will use it again in the following proof. Since the function

$$\frac{1}{N}\phi(\langle \mathbf{x}_0, \mathbf{x}_i \rangle)\psi_\ell(\langle \mathbf{x}_0, \mathbf{x}_i \rangle) \sum_{i=1}^{N} \langle \mathbf{x}_0, \mathbf{x}_i \rangle^p \left\langle \widetilde{\mathbf{x}}_i^{\otimes 3} \otimes \widetilde{\mathbf{x}}_0^{\otimes 2}, \mathbf{A}_\star \right\rangle$$

can be written as

$$\frac{1}{N} \sum_{i=1}^{N} \sum_{k=0}^{\ell+p_0} \left\langle u_{2k+r+2}(\widetilde{\mathbf{x}}_0, \widetilde{\mathbf{x}}_i), c_{\ell,k,p_0} \mathbf{A}_\star \otimes \mathbf{e}^{\otimes(4k+2r)} \right\rangle,$$

where $\mathbf{e} = \mathrm{diag}(0,0,...,1)$. By Lemma C.1, there exists $\mathbf{v}_{p,\ell}(\mathbf{W})$ s.t.

$$f_{\mathbf{v}_{p,\ell}}^{\mathbf{W}_0}(\mathbf{x}_{0:N}) = \phi(\langle \mathbf{x}_0, \mathbf{x}_i \rangle)\psi_\ell(\langle \mathbf{x}_0, \mathbf{x}_i \rangle)\frac{1}{N} \sum_{i=1}^{N} \langle \mathbf{x}_0, \mathbf{x}_i \rangle^p \left\langle \widetilde{\mathbf{x}}_i^{\otimes 3} \otimes \widetilde{\mathbf{x}}_0^{\otimes 2}, \mathbf{A}_\star \right\rangle$$

with

$$\mathbb{E}_{\mathbf{W}}\left[\|\mathbf{v}_{p,\ell}(\mathbf{W})\|_2^2\right] \leq \|\mathbf{A}_\star\|_{\mathsf{Fr}}^2 \sum_{k=0}^{\ell+p_0} 4^{2k+r+2}(2k+r+2)!c_{\ell,k,p_0}^2.$$

Notice that

$$\sum_{k=0}^{\ell+p_0} 4^{2k+r+2}(2k+r+2)!c_{\ell,k,p_0}^2$$

$$\leq \sum_{k=0}^{\ell+p_0} 4^{2k+r+2}(2k+r+2)!\left[(2\ell+p)^{p_0+r}c_{\ell+p_0,k}\right]^2$$

$$\leq (2\ell+p)^{2p_0+2r} \sum_{k=0}^{\ell+p_0} 4^{2k+r+2}(2k+r+2)!\left[\frac{2^k}{(2k)!}\sqrt{\ell+p_0}\frac{(\ell+p_0)!}{k(\ell+p_0-k)!}\right]^2$$

$$\leq 4(2\ell + p)^{2p_0+2r}(2\ell + 2p_0 + r + 2)\sum_{k=0}^{\ell_p} 4^{2k+2}(2k+2)!\left[\frac{2^k}{(2k)!}\sqrt{\ell_p}\frac{(\ell_p)!}{k(\ell_p)!}\right]^2,$$

where $\ell_p = \ell + p_0$, and by the inequality that

$$\sum_{k=0}^{\ell_p} 4^{2k+2}(2k+2)!\left[\frac{2^k}{(2k)!}\sqrt{\ell_p}\frac{(\ell_p)!}{k(\ell_p)!}\right]^2$$

$$= \sum_{k=0}^{\ell_p} 2^{4k+4}\ell_p\frac{(2k+2)(2k+1)}{k^2}\frac{2^{2k}}{(2k)!}\left[\frac{\ell_p!}{(\ell_p-k)!}\right]^2$$

$$\leq \sum_{k=0}^{\ell_p} 2^{4k+4}\ell_p\frac{(2k+2)(2k+1)}{k^2}\sqrt{\frac{\pi}{2}(2k+1)}\left[\frac{\ell_p!}{k!(\ell_p-k)!}\right]^2$$

$$\leq 8 \times 2^{4\ell_p+4}\ell_p^{\frac{3}{2}}\sum_{k=0}^{\ell}\left[\frac{\ell_p!}{k!(\ell_p-k)!}\right]^2$$

$$= 4\sqrt{2} \times 2^{4\ell_p+4}\ell_p^{\frac{3}{2}}\binom{2\ell_p}{\ell_p}$$

$$\leq 4\sqrt{2} \times 2^4\ell_p \cdot (64)^{\ell_p},$$

we obtain an upper bound:

$$\mathbb{E}_{\mathbf{W}}\left[\|\mathbf{v}_{p,\ell}(\mathbf{W})\|_2^2\right] \leq (2\ell + p)^{2p_0+2r}\|\mathbf{A}_\star\|_{\mathsf{Fr}}^2\sum_{k=0}^{\ell+p_0} 4^{2k+r+2}(2k+r+2)!c_{\ell+p_0,k}^2$$

$$\leq \|\mathbf{A}_\star\|_{\mathsf{Fr}}^2 \times 4(2\ell+p)^{2p_0+2r}(2\ell+2p_0+r+2) \times 4\sqrt{2} \times 2^4\ell_p \cdot (64)^{\ell_p}$$

$$\leq \|\mathbf{A}_\star\|_{\mathsf{Fr}}^2 (2\ell+p)^{p+r+1}8^{2\ell+2p_0+3}.$$

Finally, we consider the target function (16)

$$g_\star(\mathbf{x}_{0:N}) = \frac{1}{N}\sum_{i=1}^N F(\langle\mathbf{x}_0,\mathbf{x}_i\rangle)G(\mathbf{x}_0,\mathbf{x}_i), \quad F(t) = \sum_{p=0}^\infty a_p t^p, \quad G(\mathbf{x}_0,\mathbf{x}_i) = \langle\widetilde{\mathbf{x}}_i^{\otimes 3}\otimes\widetilde{\mathbf{x}}_0^{\otimes 2}, \mathbf{A}_\star\rangle.$$

By approximating $\frac{1}{N}\sum_{i=1}^N a_p\langle\mathbf{x}_0,\mathbf{x}_i\rangle^p G(\mathbf{x}_0,\mathbf{x}_i)$ separately and adding the approximation functions together, we obtain a $\mathbf{v}_\ell$ s.t. $f_{\mathbf{v}_\ell}^{\mathbf{W}_0}(\mathbf{x}_{0:N}) = \phi(\langle\mathbf{x}_0,\mathbf{x}_i\rangle)\psi_\ell(\langle\mathbf{x}_0,\mathbf{x}_i\rangle)g_\star(\mathbf{x}_{0:N})$ and

$$\mathbb{E}_{\mathbf{W}}\left[\|\mathbf{v}_{p,\ell}(\mathbf{W})\|_2^2\right] \leq \left(\sum_{p=0}^\infty a_p 8^{\ell+p/2+3/2}(2\ell+p)^{\frac{p+3}{2}}\right)^2\|\mathbf{A}_\star\|_{\mathsf{Fr}}^2 = 8^3 B(g_\star, L)$$

and $\left\|g_\star - f_{\mathbf{v}_\ell}^{\mathbf{W}_0}\right\|_\infty = \|(\phi\psi_\ell - 1)g_\star\|_\infty \leq \|\phi\psi_\ell - 1\|_\infty\|g_\star\|_\infty \leq \frac{e}{2^{\ell+1}(\ell+1)!}\|g_\star\|_\infty = \|g_\star\|_\infty\varepsilon_\ell.$
This finishes the proof of Lemma C.2. $\qquad\square$

## C.2    Proof of Theorem C.1

*Proof.* For any $L > 0$, using Lemma C.2, we can find a function $f_{\mathbf{v}_L}^{\mathbf{W}_0}$ such that $\left\|g_\star - f_{\mathbf{v}_L}^{\mathbf{W}_0}\right\|_\infty \leq \|g_\star\|_\infty\varepsilon_L$ with $B(f_{\mathbf{v}_L}^{\mathbf{W}_0}) \leq 8^3 B(g_\star, L)$. Follow the same manner as the proof of Theorem 2. We can get that

$$L_D(\widehat{f}_M^{\mathbf{W},\mathbf{W}_0}) - L_D(f_{\mathbf{v}_L}^{\mathbf{W}_0})$$
$$\lesssim \sqrt{B(g_\star, L)}\left[\sqrt{\frac{\log(dM)\log(nNM) + \log(\delta^{-1})}{n}} + \sqrt{\frac{(d^2 + \log M)\delta^{-1}}{M}}\right].$$

Therefore, we have that

$$L_D(\widehat{f}_M^{\mathbf{W},\mathbf{W}_0}) - L_D(g_\star)$$
$$= L_D(\widehat{f}_M^{\mathbf{W},\mathbf{W}_0}) - L_D(f_{\mathbf{v}_L}^{\mathbf{W}_0}) + L_D(f_{\mathbf{v}_L}^{\mathbf{W}_0}) - L_D(g_\star)$$
$$\lesssim \sqrt{B(g_\star, L)}\left[\sqrt{\frac{\log(dM)\log(nNM) + \log(\delta^{-1})}{n}} + \sqrt{\frac{(d^2 + \log M)\delta^{-1}}{M}}\right] + \varepsilon_L\|g_\star\|_\infty. \quad (77)$$

Taking infimum over $L$ proves Theorem 3. $\qquad\square$

## C.3 Proof of Examples in Section 4

**Proposition C.2** (Restatement of Example 4). *Consider the target function*

$$g_\star = \frac{1}{N} \sum_{i=1}^{N} \langle \mathbf{x}_i^{\otimes 3} \otimes \mathbf{x}_0^{\otimes 2}, \mathbf{A} \rangle.$$

*It has norm bound* $B(g_\star, L) = \|\mathbf{A}\|_{\mathsf{Fr}}^2 L^3 8^{2L}$. *So for any* $\eta > 0$, *if we take* $n \geq \exp(\exp(\Theta(1/\eta)))$, $L = \Theta((1 + \log\log n)^{-1} \log n)$, *and* $M = \Theta(d^2 n)$, *the excess risk will scale as* $\widetilde{\mathcal{O}}(\sqrt{\|\mathbf{A}\|_{\mathsf{Fr}}^2/n^{1-\eta}})$.

*Proof of Proposition C.2.* The value of $B(g_\star, L)$ can be obtained by definition and by direct calculation. As for the second part of the proposition, take $L = r \log n$, where $r > 0$ is a parameter that is to be chosen to minimize the excess risk. Eq. (17) becomes

$$\widetilde{\mathcal{O}}(\|\mathbf{A}\|_{\mathsf{Fr}} r^{3/2} n^{3r-1/2} + n^{-r\log(2/e) - r\log r - r\log\log n}), \tag{78}$$

where $\widetilde{\mathcal{O}}$ hides all the logarithm factors and constants of $n, d$, and $M$. To minimize the excess risk, we need to make the two terms in (78) have the same scale. So we set $3r - 1/2 = -r\log(2/e) - r\log r - r\log\log n$. Denoting the solution as $r_\star$, we get that

$$r_\star = \frac{1/2 - r_\star \log r_\star}{\log(2e^2) + \log\log n}.$$

So this gives $r_\star < (1/2 + 1/e)(\log(2e^2) + \log\log n)^{-1}$. Assume $r_\star < 1$ (otherwise $3r_\star > 1/2$ and the excess risk is meaningless), then $r_\star > (2\log(2e^2) + 2\log\log n)^{-1}$. Therefore, we get that $r_\star = C(1 + \log\log n)^{-1}$, with $C$ being a constant. As a result, when choosing $L = r_\star \log n = \Theta((1 + \log\log n)^{-1} \log n)$, the excess risk Eq. (78) scales as $\widetilde{\mathcal{O}}(\|\mathbf{A}\|_{\mathsf{Fr}} n^{3r_\star - \frac{1}{2}}) = \widetilde{\mathcal{O}}(\|\mathbf{A}\|_{\mathsf{Fr}} n^{C/(1+\log\log n) - \frac{1}{2}})$. As a result, let $n > \exp(\exp(C/\eta - 1))$ where $C$ is a constant, we get an excess risk that scales as $\widetilde{\mathcal{O}}(\sqrt{\|\mathbf{A}\|_{\mathsf{Fr}}^2/n^{1-\eta}})$. This finishes the proof of Proposition C.2. □

**Proposition C.3** (Restatement of Example 5). *Consider the target function*

$$g_\star = \frac{1}{N} \sum_{i=1}^{N} \langle \mathbf{x}_0, \mathbf{x}_i \rangle^p \langle \boldsymbol{\beta}, \mathbf{x}_i \rangle, \quad \boldsymbol{\beta} \in \mathbb{S}^{d-1}.$$

*It has* $B(g_\star, L) = (2L + p)^{p+3} 8^{2L+p}$. *For any* $\eta > 0$, *choosing the same parameters* $(n, L, M)$ *as Example 4, the excess risk bound scales as* $\widetilde{\mathcal{O}}(\sqrt{(\log n + p)^{(p+3)} 8^p/n^{1-\eta}})$.

*Furthermore, to reach an accuracy of* $0.01$, *the* BRFA *model requires* $n_\star = \widetilde{\mathcal{O}}((8p+48)^{p+3})$, *whereas the* RFA *model requires* $n_\star = \widetilde{\mathcal{O}}((4d)^p)$.

*Proof of Proposition C.3.* The value of $B(g_\star, L)$ can be obtained by direct calculation, and we use the same method as the proof of Proposition C.2 to get the $\widetilde{\mathcal{O}}(\sqrt{1/n^{1-\eta}})$ excess risk. To reach an accuracy of 0.01, we can set $L = 3$, note that $\varepsilon_L \|g_\star\|_\infty < 0.006$. Therefore we just need to choose $n_\star$ s.t. $n > \widetilde{\mathcal{O}}((6+p)^{p+3} 8^{6+p})$, and this gives $n_\star = \widetilde{\mathcal{O}}((8p+48)^{p+3})$ for the BRFA model. Direct calculation using Theorem 2 gives the value of $n_\star$ for the model RFA. This finishes the proof of Proposition C.3. □

**Proposition C.4** (Restatement of Example 6). *Consider the target function that has the form*

$$g_\star = \frac{1}{N} \sum_{i=1}^{N} \cos(\langle \mathbf{x}_0, \mathbf{x}_i \rangle) \langle \mathbf{x}_i^{\otimes 3}, \mathbf{G} \rangle.$$

*It has* $B(g_\star, L) = \Theta((8e)^{2L})$. *For any* $\eta > 0$, *choosing the same parameters as Example 4, the excess risk bound scales as* $\widetilde{\mathcal{O}}(\sqrt{1/n^{1-\eta}})$.

*Furthermore, to reach an accuracy of* $0.01$. *The* BRFA *model requires* $n_\star = \widetilde{\mathcal{O}}(1)$, *whereas the* RFA *model requires* $n_\star = \widetilde{\mathcal{O}}(\mathrm{Poly}(d)\exp(\sqrt{d}))$.

*Proof of Proposition C.4.* We use the expansion of the cos function,

$$g_\star = \frac{1}{N} \sum_{i=1}^{N} \sum_{k=0}^{\infty} \frac{(-1)^k}{(2k)!} \langle \mathbf{x}_0, \mathbf{x}_i \rangle^{2k} \langle \mathbf{x}_i^{\otimes 3}, \mathbf{G} \rangle .$$

Then use the Eq. (17), we get that

$$B(g_\star, L) = \sum_{k=0}^{\infty} \frac{(2L + 2k)^{k+3/2} 8^{L+k}}{(2k)!} \leq \sum_{k} \frac{(2k+3)^{k+3/2} 8^k}{(2k)!} (8e)^L = \Theta((8e)^L).$$

This gives the formula for $B(g_\star, L)$. To reach an accuracy of 0.01, we set $L = 3$. Then $B(g_\star, L) = \Theta(1)$. So `BRFA` needs $n_\star = \widetilde{\mathcal{O}}(1)$. And Theorem 2 and Example 3 show that `RFA` model would require $n_\star = \widetilde{\mathcal{O}}(\mathrm{Poly}(d) \exp(\sqrt{d}))$. This finishes the proof of Proposition C.4. $\qquad\square$

## D   Further experiments

In addition to Section 5, we perform further simulations to examine our theory upon the approximation power of `RFA` (5), `BRFA` (15), and `RFMLP` (14). Besides the target function (21), we consider two additional target functions of the form

$$f_{3,p}(\mathbf{x}_{0:N}) = \langle \boldsymbol{\beta}, \mathbf{x}_0 \rangle^p, \qquad p \in \mathbb{N}, \quad \boldsymbol{\beta} \in \mathbb{S}^{d-1}, \tag{79}$$

$$f_{4,\gamma}(\mathbf{x}_{0:N}) = \frac{1}{N} \sum_{i=1}^{N} \langle \mathbf{x}_0, \mathbf{S} \mathbf{x}_i \rangle^3 \langle \boldsymbol{\beta}, \mathbf{x}_i \rangle, \quad \boldsymbol{\beta} \in \mathbb{S}^{d-1}, \quad \mathbf{S} = \mathbf{Z} + \gamma I_d. \tag{80}$$

The target function (79) is a specific instance of Example 1, and the target function (80) is a specific instance of Example 3. In (80), we sample $Z_{ij} \sim_{\mathrm{iid}} \mathsf{N}(0, 1/d)$ for $(i, j) \in [d]^2$ and vary $\gamma$ in the experiment. Other experimental settings are the same as in Section 5.

Figure 3 demonstrates the effect of sequence length $N$ on the performance of three `RF` models when fitting the target function $f_{3,2}$ (79), which solely depends on $x_0$. We observe that a larger sequence length $N$ leads to a larger separation of the test error between `RFMLP` and the other two random-feature attention models. This result aligns with our sample complexity analysis detailed in Example 1, where the sample complexity bound of `RFMLP` for learning average of functions of $\mathbf{x}_i$ is found to be $\mathcal{O}((N/4)^p)$ times greater than that of `RFA`.

Figure 4 demonstrates the performance comparison between `RFA` and `BRFA` under different choices of the token dimension $d$. As we can see from the left panel of Figure 4, in the case of $d = 4$, `RFA` outperforms `BRFA` for large sample size, which may result from that `BRFA` has slower convergence rate with respect to the sample size $n$ as we state in Example 5. In the middle and the right panel (i.e., when $d$ is larger), `BRFA` exceeds `RFA`, and the largest separation lies in the case of $d = 32$. This result is consistent with our analyses that `BRFA` saves a Poly(d) factor in the sample complexity for $d \gg p$.

Figure 5 demonstrates that `BRFA` has no advantage in approximating the target function $f_{4,\gamma}$ for $\gamma = 0$ (left panel; i.e., $(S_{ij} \sim_{\mathrm{iid}} \mathsf{N}(0, 1/d)$, for $(i, j) \in [d]^2)$. However, as $\gamma$ increases, `BRFA` outperforms `RFA` and their separation increases with a larger $\gamma$ (middle and right panel). Notice that $\lim_{\gamma \to \infty} \langle \mathbf{x}_0, \mathbf{S} \mathbf{x}_i \rangle / \gamma = \langle \mathbf{x}_0, \mathbf{x}_i \rangle$. So this result also conforms to our analysis that `BRFA` is adept at approximating functions of correlations as in Example 5.

In conclusion, `RFA` and `BRFA` have similar performance in fitting functions without correlation structure, such as (20) and (79), and `RFA` may behave even better in some cases. However, `BRFA` is presumably more powerful than `RFA` in approximating functions of correlations.

### D.1   Weight matrices in BERT

As noted in Section 4, we plot the query-key matrices (weight matrices) of the BERT-Base model [29][6] and show that many query-key matrices are diagonally dominated. The BERT-Base model has 12 attention layers with 12 heads in each layer. Denote the query matrix in the $i$-th head of $j$-th layer as $Q_{ij} \in \mathbb{R}^{768 \times 64}$ and the key matrix as $K_{ij} \in \mathbb{R}^{768 \times 64}$. We compute $W_{ij} = \sqrt{768} \cdot Q_{ij} K_{ij}^\top$ for $i, j$ from 1 to 12, and then take the absolute value for each entry of the weight matrices $W_{ij}$. Figure 1

---

[6]Downloaded from `https://huggingface.co/bert-base-uncased`.

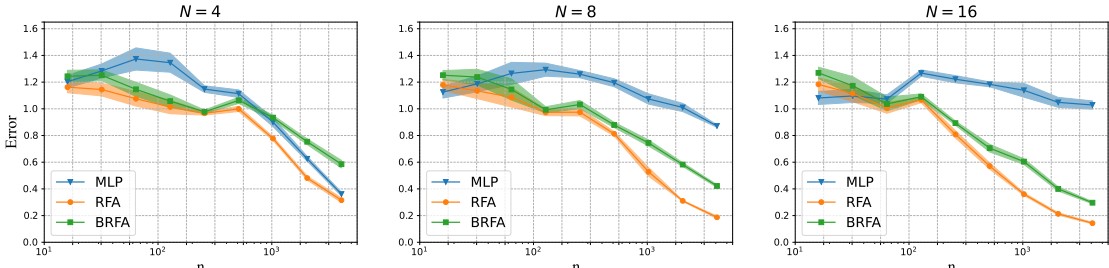

Figure 3: Test error of three RF models for learning $f_{3,2}$ (79). We fix $d = 16$ while varying $N = 4$ (left), 8 (middle), and 16 (right). The other settings are the same as in Figure 2.

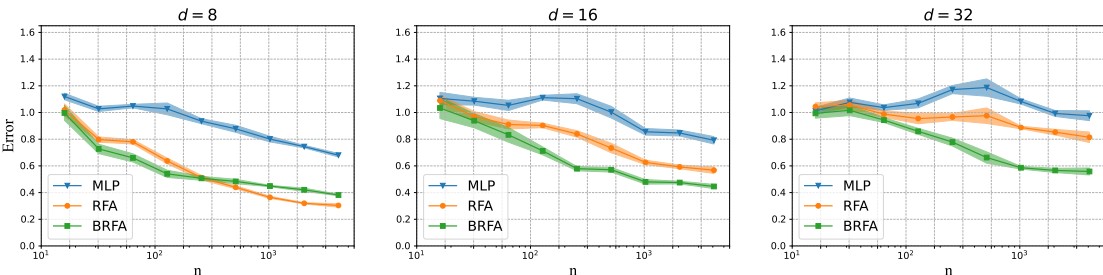

Figure 4: Test error of three RF models for learning $f_{2,3}$ (21). We fix $N = 16$ while vary $d = 8$ (left), 16 (middle), and 32 (right). For a fair comparison among RFA, BRFA, and RFMLP, the number of heads of RFMLP is taken to be $M_{\text{RFMLP}} = 9000$ (left), 17000 (middle), and 33000 (right). The other settings are the same as in Figure 2.

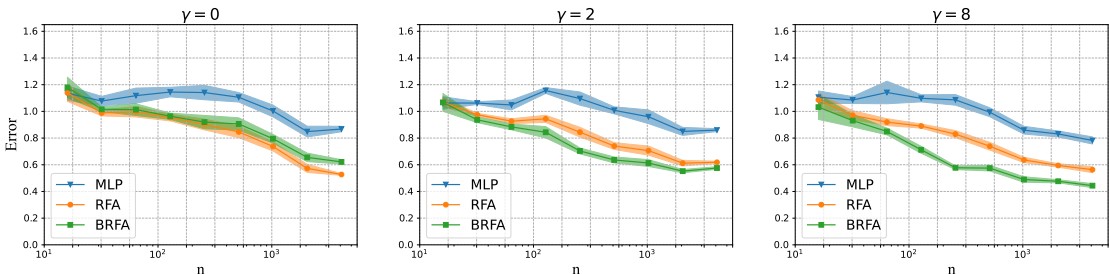

Figure 5: Test error of three RF models for learning $f_{4,\gamma}$ (80). We choose $\gamma = 0$ (left), 2 (middle), and 8 (right). The other settings are the same as in Figure 2.

shows the heat maps of weight matrices of the 2nd, 5th, 8th, and 11th layers, where all matrices are clipped to the top-left $32 \times 32$ block. As we can see, a lot of weight matrices are diagonally dominated. We remark that a very recent and concurrent paper [84] observed similar phenomena for the ViT-Tiny model. They further show that diagonally dominated weight initialization of self-attention layers allows training transformers faster and obtaining higher final accuracies on CIFAR-10 and ImageNet datasets.

