# OpenReview forum: "What can a Single Attention Layer Learn? A Study Through the Random Features Lens"
_NeurIPS.cc/2023/Conference — NeurIPS 2023 poster_

### Official Review · Reviewer_giTQ · 2023-06-25

**Soundness:** 3 good
**Presentation:** 4 excellent
**Contribution:** 2 fair
**Rating:** 5
**Confidence:** 3

**Summary:**

This paper explores the learning capabilities of a single attention layer, assuming keys and queries to be random and frozen (as in the random features model).

**Strengths:**

The paper is very well written, and the technical claims look formally supported.

The paper deals with an important problem, which is to theoretically better characterize the representation power of single attention layers.



**Weaknesses:**

Typo in line 95 "theorey"

The family of target functions considered in the discussion doesn't contain terms that consider the interaction between token $i$ and token $j$, with $i, j \neq 0$. Attention model success also hinges on capturing the relation between different tokens in the context. It looks like this work leverages on something orthogonal, and I wonder if the results are really representative of the attention layer power.

Typo in line 176 "$i$"

It is not clear to me how the permutation invariance of the input tokens represents a valuable propertyy over which attention should perform better than MLPRF. It could be that the toy-model analyzed has this power as a natural difference with respect the standard MLPRF, without really capturing the properties of attention.

**Questions:**

You consider single query token models. This semplify the derivation and I agree that is a reasonable simplification. Can you still argue more on this choice? Is this equivalent to consider a model with $N$ queries, after a left inner product with the canonical basis vector $e_1$? Is this method used in practice in single-output attention layers? Has it been considered in previous theoretical work?

Are your results optimization agnostic? There is any possible way to bridge the solutions you mention in Theorem1 with the solutions found by a GD algorithm? In this sense, are the assumptions over the constrained class $\mathcal V_M$ going in that direction?

It is not clear to me in Theorem2 if your minimizer $\hat V$ is unique, or if you don't need it, as any $\hat V$ would respect your claim.

Can you provide more intuition of why the permutation invariance is a property that is intrinsic of tasks where attention outperforms fully connected models?

**Limitations:**

No need

---

> ### Author Rebuttal · Authors · 2023-08-09
>
> We thank the reviewer for the valuable feedback. We respond to the questions as follows.
>
> > You consider single query token models... Has it been considered in previous theoretical work?
> The family of target functions considered in the discussion doesn't contain terms that consider the interaction between token $i$ and token $j$ with $i \neq j$.
>
> Our single-query token model is equivalent to **simply looking at the first output token** of a full self-attention (cf. Line 112-116), and abstracting out the query token to be $x_0$ and the key tokens to be $x_{1:N}$. This does not change the model at all, and merely avoids repeating our arguments $N$ times for the $N$ output tokens. Our results can be easily mapped back to full self-attention by repeating our arguments on all $N$ output tokens. In that case, the attention model would involve interaction between any $x_i$ and $x_j$ (instead of just $x_0$ and $x_j$).
>
> There exist many literature that constructs sequence-to-sequence transformers to implement a sequence-to-single-token function, by "reading out" from a certain output token (such as the last output token), for example Garg et al. (2022), Akyurek et al. (2022). Our model can be seen as a single-layer version of these transformers. We will add a discussion about this point in our revision.
>
> > Is this equivalent to consider a model with $N$ queries, after a left inner product with the canonical basis vector $e_1$? Is this method used in practice in single-output attention layers?
>
> It is equivalent to using only the first token as the query. It is not equivalent to applying $e_1^\top$ on the left to all $N$ query tokens.
>
> > Are your results optimization agnostic? There is any possible way to bridge the solutions you mention in Theorem1 with the solutions found by a GD algorithm? In this sense, are the assumptions over the constrained class $\mathcal{V}_M$ going in that direction?
>
> Our results are optimization agnostic. Any standard convex optimization algorithm (including GD) for the constrained problem on $\mathcal{V}_M$ (or an equivalent regularized problem) can find an approximate solution efficiently. In our experiments, we used Adam with weight decay, which we found was a good enough optimizer on all our problem instances.
>
> > in Theorem 2 if your minimizer $\hat{V}$ is unique, or if you don't need it
>
>  The minimizer $\hat{V}$ indeed may not be unique, and we don't need uniqueness in Theorem 2; the statement holds for any minimizer (due to the uniform concentration argument).
>
> > Can you provide more intuition of why the permutation invariance is a property that is intrinsic of tasks where attention outperforms fully connected models?
> It is not clear to me how the permutation invariance of the input tokens represents a valuable propertyy over which attention should perform better than MLPRF. It could be that the toy-model analyzed has this power as a natural difference with respect the standard MLPRF, without really capturing the properties of attention.
>
> Our attention models can only fit permutation invariant target functions due to the structure of attention heads. On such target functions, our random-feature attention (RFA) models do achieve better sample complexities than RFMLP as they exploit this structure.
>
> We remark that we also have results on comparing different weight distributions in RFA, where query-key matrices with non-zero means achieve better sample complexities than zero-mean ones for learning certain functions (Section 4). These results are more intrinstic to the attention structure and do not have analogues in RFMLP models to our best knowledge. See our Additional Response to All Reviewers for more details.

---

> > ### Comment · Reviewer_giTQ · 2023-08-12
> >
> > I thank the authors for their detailed rebuttal. I will follow up with additional questions.
> >
> >
> > -  It is not equivalent to applying $e_1^\top$ on the left to all $N$ query tokens.
> >
> > I'm a little confused on this. As the soft-max is applied row-wise, introducing the other $N$ tokens in the attention matrix wouldn't change the first row (the one generated by $q_0$). Looking only at this first row later (multiplying the attention matrix on the left with $e_1^\top$), should give the same output your model is considering.
> >
> > - On the permutation invariance of target functions.
> >
> > Sorry if I was not clear enough before. Your work proves that RFA models are better in learning permutation invariant target functions (or other functions defined in Section 4). Can you argue why this could be a reason why attention layers perform better than other architectures (e.g. RFMLP) in natural language tasks? At the moment, I do not see any connection between the target functions you study and what it could be a (very toysh) NLP-target function.
> >
> > Following up on this last point, I want to remark again my point raised as second weakness (relation between token $i$ and $j$), if you could elaborate a bit more.

---

> > > ### Author Response · Authors · 2023-08-16
> > > **Response to further questions**
> > >
> > > Thank you for the thoughtful response. We respond to the further questions as follows.
> > >
> > > > Is this equivalent to consider a model with $N$ queries, after a left inner product with the canonical basis vector $e_1$?
> > >
> > > We may have misunderstood your question in our original rebuttal (and we apologize for any confusion). Multiplying $e_1$ to the softmax matrix indeed leads to the same output as our formulation.
> > >
> > > > The family of target functions considered in the discussion doesn’t contain terms that consider the interaction between token $i$ and token $j$, with $i, j \neq 0$.
> > >
> > > There are two ways to incorporate the interaction between token $i$ and $j$ in the target function.
> > >
> > > 1. Currently, we are considering the (simplified) sequence-to-scalar attention models
> > >
> > >    $y_0=f(x_0, (x_i)_{i=1}^n)$
> > >
> > >    where all tokens only interact with $x_0$. When we map our results back to full sequence-to-sequence attention models
> > >
> > >    $(y_i) _{i=1}^n = f( (x_i) _{i=1}^n )$
> > >
> > >    (i.e., when we don’t multiply $e_1$ to the softmax matrix), the output tokens $y_i$ and $y_j$ will contain interactions between token $x_i$ and $x_j$. However, analyzing this full sequence-to-sequence model would be similar to our current sequence-to-scalar model (cf. the discussions in Line 112-116 of the main paper).
> > >
> > > 2. Going beyond our current setting, suppose we restrict to sequence-to-scalar target functions $y_0=f(x_0, (x_i)_{i=1}^n)$ but still want $f$ to involve interaction between all $x_i$ and $x_j$. One way to approximate such functions is to consider multi-layer attention networks instead, with full sequence-to-sequence self-attention as the intermediate layers, plus a final sequence-to-scalar layer. We believe this would be an interesting direction but beyond the scope of the current work.
> > >
> > > > Can you argue why this could be a reason why attention layers perform better than other architectures (e.g. RFMLP) in natural language tasks?
> > >
> > > To provide a toy example, consider a simple task where the input sequence is “aabbccada”, and our target function is to count the number of “a”s in the sequence (on this example, 4). By embedding letters as orthogonal vectors, the target function can be represented as a one-layer attention network $f_\star = \sum_{j=1}^N {\rm ReLU}(\langle x_0 W, x_j \rangle) \langle x_j,v\rangle$, which counts the numbers of tokens that are the same with token $x_0$. The number of parameters of this transformer is independent of the input sequence length $N$. However, using MLPs, the sample complexity would naturally depend linearly on the sequence length $N$. This illustrates the benefit of the attention layer.

---

> > > > ### Comment · Reviewer_giTQ · 2023-08-20
> > > >
> > > > I thank the authors for their explanation. I would suggest adding such example in the final version. If considered useful, also the comment about the equivalence with the model with $e_1^\top$ on the left might make the reading more comfortable. I confirm my score.

---

> > > > > ### Author Response · Authors · 2023-08-21
> > > > >
> > > > > Thank you again for your attention and the suggestions!

---

### Official Review · Reviewer_3Deb · 2023-07-04

**Soundness:** 4 excellent
**Presentation:** 4 excellent
**Contribution:** 3 good
**Rating:** 7
**Confidence:** 3

**Summary:**

The paper considers the representational and generalization properties single-layer scalar-valued transformer models with random key and query matrices and value vectors that can depend on those random matrices. They draw a comparison to the well-studied random-feature models for two-layer neural networks. Concretely:
* Theorem 1 proves that functions of the form $f_*(x_{0:N}) = \frac1N \sum_i F(x_0, x_i)$ can be efficiently represented under the random feature, with quadratic dependence on the input dimension $d$ and no dependence on the sequence length $N$.
* Theorem 2 extends this approximation-theoretic result to generalization by proving a generalization bound on the empirical risk-minimizing random feature transformer (among transformers with bounded value vectors) that fits a noiseless dataset. The proof follows from bounds on the Rademacher complexity of the family of functions that approximately represent the dataset.
* The paper gives several examples in Section 3.3 of functions of the above form and show that they admit much stronger learning rates for random feature attention models than for standard random feature models. Generally, standard random feature models have a substantial dependence on $N$, while random feature attention has no such dependence.
* Theorem 3 proves a generalization bound similar to Theorem 2, but in the regime where the random feature matrices (e.g. the product of the key and query matrices) is biased in favor of larger elements on the diagonal. (This is empirically motivated in the appendix by a finding that BERT weight matrices frequently concentrate mass on the diagonals.) They prove generalization bounds for a restricted family of target functions, where high-degree polynomials of $\langle x_0, x_i\rangle$ and low-degree polynomials of $x_0 \otimes x_i$ may be averaged together. They provide several examples showing how this model can reduce the error rates over the standard random feature attention model.
* Numerical experiments in Section 5 validate their theoretical results by comparing the error rates as a function of sample complexity of random feature MLPs, random feature attention, and biased random feature attention.

**Strengths:**

The work is novel, interesting, and relevant to the growing study of the theoretical properties of transformers. The work formalizes some intuitive advantages that attention models hold over standard MLPs in their ability to compute and aggregate pairwise functions of sequential inputs. Theoretical results are presented cleanly and the proofs that I read appeared correct. The work is creative, and draws inspiration for Theorem 3 from empirical observations. There is interesting follow-up work to be done on understanding the strengths and limitations of this model, especially on the optimization front.

**Weaknesses:**

While the bounds are interesting in their own right, the comparisons between random feature attention and MLP models focus on a few particular examples, whose generality is unclear. Moreover, the comparisons are between upper bounds for both models; ideally, the results would contrast with _lower_ bounds for random feature MLPs.

### Minor pointers
l244: "scaler" -> "scalar"

l657: $W$ in equation block should be $W_m$

**Questions:**

Would you mind explaining if you expect that the $\delta^{-1}$ dependence in Theorem 1 could be improved? As far as I am aware, other random feature approximation papers can often achieve a $\sqrt{\log \delta^{-1}}$ dependence by employing concentration bounds over Hilbert spaces instead of Markov or Chebyshev, as is employed in the proof of Lemma B.1.

I would be interested to know why in particular you chose to focus on the diagonally-biased weight initializations, when the plots in Figure 5 appear to make similarly strong cases in favor of selecting random feature matrices $W$ from distributions that impose either sparsity or low rank. (I am okay with these regimes not being included in the paper; I am mostly just interested in the choice.)

Would it be possible to offer a more concrete comparison with the Rademacher complexity generalization bounds in this work and single-layer covering numbers-based bound of [Edelman et al](https://arxiv.org/abs/2110.10090)? While the regimes are different, I think the paper would benefit from a brief comparison of the advantages and disadvantages of each.

**Limitations:**

The work makes modeling assumptions with substantial differences from standard transformer architectures, notably the random weights, the use of a single layer, the lack of softmax, and scalar rather than sequential outputs. The work is upfront those limitations.

While the random features regime for 2-layer neural networks is interesting for studying the representational and generalization properties of certain training regimes, it's well-known that the method (as well as all other kernel-based approaches) fall short when learning even simple learning problems like single-index models. (Or as was argued in Example 1 of this paper, how standard random feature models fall short on sequential tasks whose inputs depend exclusively on $x_0$.) Are there similar illustrative examples that random feature attention units fail to efficiently approximate?

Moreover, when providing examples that separate random feature MLPs and attention models, it may be helpful to account for a particular examples of a target function where the random feature MLP model offers a better error bound, or to make an argument that RFA models will always have superior rates under certain sequential assumptions.

---

> ### Author Rebuttal · Authors · 2023-08-09
>
> Thank you for your positive feedback and the suggestions on our paper. We would appreciate if you could champion our paper in the discussions!
>
> >While the bounds are interesting in their own right, the comparisons between random feature attention and MLP models focus on a few particular examples, whose generality is unclear. Moreover, the comparisons are between upper bounds for both models; ideally, the results would contrast with lower bounds for random feature MLPs.
>
> Our comparisons are indeed between upper bounds. Existing work has also derived lower bounds on the sample complexity of RFMLPs (e.g., learning degree polynomial by RFMLP requires $\Omega((dN)^p)$ samples, (Ghorbani et al., 2021)), which agrees with the upper bound for RFMLP we used though they apply to a special case with a uniform distributional assumption (input vector uniformly distributed on the sphere).
>
> We will make sure to emphasize this point and add a discussion of the lower bound in our revision.
>
> >Would you mind explaining if you expect that the $\delta^{-1}$
>  dependence in Theorem 1 could be improved? As far as I am aware, other random feature approximation papers can often achieve a
>  $\sqrt{\log \delta^{-1}}$ dependence by employing concentration bounds over Hilbert spaces instead of Markov or Chebyshev, as is employed in the proof of Lemma B.1.
>
> The $\delta^{-1}$ dependence comes from the Chebyshev bound in Lemma B.1, which in turns comes from the precondition $\mathbb{E}[\|v(W)\|_2^2]\le R^2$. Since we only assume a bounded second moment, the Chebyshev bound is likely the best achievable. Further, bounded second moment is likely the best we can do for expressing polynomials due to RKHS norm related arguments (cf. the proof of Lemma B.2).
>
> >I would be interested to know why in particular you chose to focus on the diagonally-biased weight initializations, when the plots in Figure 5 appear to make similarly strong cases in favor of selecting random feature matrices $W$ from distributions that impose either sparsity or low rank. (I am okay with these regimes not being included in the paper; I am mostly just interested in the choice.)
>
> We acknowledge the reviewer's point that other patterns such as low rank or sparsity may be present in Figure 5. However, the diagonal pattern appears the most predominant, which motivated our setting here.
>
> >Would it be possible to offer a more concrete comparison with the Rademacher complexity generalization bounds in this work and single-layer covering numbers-based bound of Edelman et al? While the regimes are different, I think the paper would benefit from a brief comparison of the advantages and disadvantages of each.
>
> We will incorporate a discussion of the work cited by the reviewer. However, we note that the settings differ substantially. The previous work considers full attention where $Q, K,$ and $V$ are all trainable, whereas we analyze random feature attention with only a trainable $V$. Consequently, the results of Edelman et al. bound a larger quantity than our result and thus are not directly comparable.
>
>
> >Limitations
>
> We appreciate the very detailed suggestions on the limitations of our work, such as target functions that cannot be approximated by RFA or those requiring more samples than RFMLP. We will think carefully about these limitations and make sure to add a discussion of them in our revision.

---

> > ### Comment · Reviewer_3Deb · 2023-08-16
> >
> > Thank you for addressing my concerns and promising to make updates to the paper accordingly. I continue to believe that the paper provides an interesting contrast with previous work on RFMLP approximation powers, and my score will remain the same.

---

> > > ### Author Response · Authors · 2023-08-18
> > > **Response**
> > >
> > > Thank you for your response and the support on our paper!

---

### Official Review · Reviewer_6d3X · 2023-07-06

**Soundness:** 3 good
**Presentation:** 3 good
**Contribution:** 2 fair
**Rating:** 5
**Confidence:** 3

**Summary:**

The paper examines the capabilities of a single-layer multi-head attention layer in a scenario where the Key and Query matrices are predetermined and randomly selected from a Gaussian distribution. The only modifiable component is the Value matrices, and when provided with a convex loss, the minimization problem becomes convex.   The authors establish expressivity results showing that, whether there is or not there is bias in the Key and Query matrices, the model can effectively learn a class of functions that exhibit permutation invariance to the Key vectors. Furthermore, they demonstrate that the sample complexity of their model is superior to that of two-layer random feature networks for the specific function class. The attention model investigated uses ReLu rather than the more common softmax attention

**Strengths:**

- Results are new and although not surprising require certain efforts to prove and could be a useful addition to the literature of random-feature -type models (more to that literature I think rather than to the attention/transformer literature)

- Paper is well written (but please correct the many typos) and the authors provide comprehensive explanations of limitations

**Weaknesses:**

- A major weakness is that instead of the common softmax function, the authors opt to use the ReLU function.
The use of ReLu is nonstandard in transformers and should be mentioned explicitly in the abstract and contributions. Relu and softmax actually have rather different properties and this should be clarified

- The fixed Key and Query matrices restrict the learning setting to a linear problem.

- The (as the authors admit) seemingly unnatural constraint set in (12) is rather "artificial". Why is such a constraint needed? Besides, knowledge of this K1,K2 requires bounds on B(f_*) making it rather impractical

- Thm 2 only applies for bounded loss. Is it possible to extend to say square-loss?

- Comparing RFA to RFMLP is based on comparing upper bounds to each other. Is it known if the latter bounds are tight?

- As the authors acknowledge, it is rather expected that RFA would beat RFMLP for the specific function class that involves correlations between key and query tokens. While the analysis is non-trivial it is questionable what is that we really learn from those bounds regarding attention?

- Even though the results are new and the proofs require efforts, the techniques are rather standard and is not clear if they are revealing of any special properties of attention? If so, this would be interesting to emphasize.

- There are a lot of typos

- My overall concern with the paper is not about the motivation of the setting (use of ReLu and random features) and also it is a bit unclear what the take home message is (other than a technically solid analysis in a mathematically interesting setting)


**Questions:**

- Why not normalize the model with 1/M rather than including this scaling afterwards on the v constraint set? Isn't that also more revealing of the "proof" strategy that you compare to expectation?
- Eventually, can the authors comment on where they see this study leading to? What specifically revealing does it show about attention and what are the authors thoughts on the use of random features in this setting?
- How would the results chance if instead of W being Gaussian, K, Q with lower inner dimension are Gaussian?
- If not mistaken I think I have seen in the literature some works experimentally validating performance of random feature attention (although with softmax). This might be worth taking a look as it might help the narrative (sorry that I don't remember on top of my head)
- Can you comment on how to derive the last inequality in line 736 of Lemma b3
- In Eq. (25), is it \sum_{i=1}^N or \sum_{j=1}^n?
- How you guys compare the results in line 263 and 264.
- Have you considered running your own experiments for softmax function instead of ReLU (to see the possible difference or implications for your results)?

---

> ### Author Rebuttal · Authors · 2023-08-09
>
> We thank the reviewer for the valuable feedback. We respond to the comments as follows.
>
> ### Response to questions regarding the setting and message:
>
> >A major weakness is that instead of the common softmax function, the authors opt to use the ReLU function.
>
> A significant portion of our results (such as the generalization bound, and expressing functions within the infinite-head random feature space (cf. Eq(26)) using finitely many heads) would not change if we change the ReLU activation to the softmax. However, the ReLU activation enables a more precise understanding of the infinite-head random feature space, which we show includes natural target functions such as polynomials. This property about ReLU has also been used in many existing work on random feature models, such as Arora et al. 2019 and the many follow-ups after.
>
> We agree that extending our analyses to random features with the softmax activation (especially studying the function space it can express) is an interesting question, which we would like to leave as future work.
>
> >The fixed Key and Query matrices restrict the learning setting to a linear problem.
>
> Our random feature model is linear in the parameter $(v_m)$ but **non-linear in the input $(x_i)_{i=0}^n$**. Such random feature models capture many non-trivial aspects of fully learnable neural networks, and have been a central topic of study in the deep learning theory literature (e.g. Daniely 2017, Mei & Montanari 2021, and the many references therein).
>
> > what is that we really learn from those bounds regarding attention?
>
> > Even though the results are new and the proofs require efforts, the techniques are rather standard and is not clear if they are revealing of any special properties of attention? If so, this would be interesting to emphasize.
>
> > My overall concern with the paper is not about the motivation of the setting (use of ReLu and random features) and also it is a bit unclear what the take home message is (other than a technically solid analysis in a mathematically interesting setting)
>
>
>
> Our results convey several new messages, including
> * The sample complexity of random-feature attention does not depend on the sequence length, which contrasts with the random-feature MLP model (Section 3);
> * A non zero-mean query-key matrix could further improve the sample complexity for learning certain functions of correlations (Section 4).
>
> Both results are not present in existing random features theory (for fully-connected neural networks). Please refer to our Additional Response to All Reviewers for more details.
>
>
> ### Response to technical questions:
>
> > The seemingly unnatural constraint set in (12) is rather "artificial.”
>
> (12) has two constraints, the total norm and the total norm squared. The total norm squared constraint is essential and natural (equivalent to an $L_2$ regularization on $(v_m)$). By contrast, the total norm condition is a technical condition merely for a slightly tightened sample complexity.
>
> > Thm 2 only applies for bounded loss. Is it possible to extend to say square-loss?
>
> For squared loss with unbounded labels, we believe a similar result would still hold by using standard machineries such as truncation. We assumed the loss is Lipschitz and bounded at $0$ for simplicity only.
>
> > Comparing RFA to RFMLP is based on comparing upper bounds to each other.
>
> Our comparisons are indeed between upper bounds. Existing work has also derived lower bounds on the sample complexity of RFMLPs (e.g., learning degree $p$ polynomial by RFMLP requires $\Omega((dN)^p)$ samples, (Ghorbani et al., 2021)), which agrees with the upper bound for RFMLP we used though they apply to a special case with a uniform distributional assumption (input vector uniformly distributed on the sphere).
>
> We will make sure to emphasize this point and add a discussion of the lower bound in our revision.
>
> > Why not normalize the model with 1/M
>
> We did not include any normalization for simplicity of the presentation only, and we agree that normalizing by $1/M$ can make the model more intuitive. We will carefully think about whether to add this normalization this in our revision.
>
> > How would the results chance if instead of W being Gaussian, K, Q with lower inner dimension are Gaussian?
>
> If $Q,K$ are Gaussian and have large inner dimensions, $W=Q^\top K$ would be have similar to a Gaussian and thus we don't expect our results to change much. However, if $Q,K$ have small inner dimensions, then $W$ would be low-rank and thus behave very differently.
>
> > How to derive the last inequality in line 736 of Lemma b3?
>
> The equation in line 736 of Lemma b3: We first drop the $\log(dM)/n$ term since $n\geq \log(dM)$. Then we use the bound for the expectation of the maximum of the sub-gaussian r.v.'s.
>
> >How you guys compare the results in line 263 and 264.
>
> The arguments can be found in the proof of Proposition C.2 (Line 919-930).

---

> > ### Comment · Reviewer_6d3X · 2023-08-14
> > **Use of ReLU**
> >
> > Thank you for your response.
> >
> > I continue to think that this is a good contribution to the literature on RF models , but one that bears limitations when phrased in the context of the transformer self-attention mechanism: (1) use of ReLU rather than softmax, (2) the requirement number of heads at least proportional to n (e.g. examples 1,2,4,5) (3) treating Q,K as single parameter matrix W ignoring the potentially low-rank structure. I understand that the use of ReLU makes the analysis more tractable. As the reviewers mention, this allows them to leverage existing techniques developed for MLPs. On the other hand, the standard self-attention module relies on softmax and it is not clear whether results for ReLU are also applicable (and to what extent) to softmax. I believe it would be particularly interesting extending the RF analysis to softmax and this probably requires new tools. In order to leave clear room for such extensions, I strongly believe that the use of ReLU in the paper should already be mentioned in the abstract and introduction (if not at the title).

---

> > > ### Author Response · Authors · 2023-08-16
> > > **Response on limitations**
> > >
> > > Thank you for the response and the constructive suggestions. We agree that the three limitations you pointed out are valid concerns that should be addressed, and we will aim to improve upon these aspects in future work.
> > >
> > > Here we would like to raise a few points about these limitations that may mitigate their impact in the current work to some extent.
> > >
> > > > In order to leave clear room for such extensions, I strongly believe that the use of ReLU in the paper should already be mentioned in the abstract and introduction (if not at the title).
> > >
> > > We agree that our use of a ReLU-based self-attention module differs from the standard softmax-based approach. As suggested, we will emphasize this in our abstract to leave room for extensions to softmax attention. However, we believe the ReLU attention is a pragmatically useful starting point. For example, on the practical side, some recent work has found that larger-scale ReLU-based transformers perform similarly as standard softmax-based ones [73].
> > >
> > > [73] K. Shen, J. Guo, X. Tan, S. Tang, R. Wang, and J. Bian. A study on relu and softmax in transformer. https://arxiv.org/abs/2302.06461
> > >
> > > > the requirement number of heads at least proportional to n (e.g. examples 1,2,4,5)
> > >
> > > Our general result (Theorem 2) does not require a hard lower bound like $M\ge O(n)$. Then, in our examples, we chose $M\ge O(n)$ simply to make the approximation error term (induced by finite $M$) in Theorem 2 smaller than the generalization term, so as to simplify the rate. Such a treatment (choosing a high-enough number of neurons to make the approximation error negligible) is also standard in the analysis of over-parametrized random feature / neural tangent kernel models, see e.g. Arora et al. (2019, Theorem 5.1).

---

> > > > ### Comment · Reviewer_6d3X · 2023-08-20
> > > >
> > > > Dear Authors,
> > > >
> > > > Regarding ReLU:
> > > > With regard to ReLU, while I lack familiarity with the specific reference, I see no reason to object to it may lead to practical improvements in certain scenarios enhancing your paper's direct relevance to such cases. However, the standard approach is softmax, so I appreciate the authors' commitment to acknowledging the use of ReLU in the abstract and ensuring transparency in the contribution. Additionally, a more in-depth discussion on the potential limitations stemming from the requirement on the number of heads > n would be valuable.
> > > >
> > > > Thank you for your attention and responses.

---

> > > > > ### Author Response · Authors · 2023-08-21
> > > > >
> > > > > Thank you again for your attention and all the suggestions!

---

### Official Review · Reviewer_5oTn · 2023-07-06

**Soundness:** 3 good
**Presentation:** 3 good
**Contribution:** 2 fair
**Rating:** 5
**Confidence:** 3

**Summary:**

This paper first studies the expressive power of a random-feature attention layer and then provides the generalization gap of the attention layer. A sample complexity bound is shown, which indicates a larger number of attention heads help the generalization. This paper also compares the results between attention and MLP layers given several target functions. Numerical experiments support the theory. I am willing to update my score after the rebuttal if my concerns are addressed with revisions in the manuscript.

-------------------------------------------------------------------
After rebuttal, I increased my score to 5. Please see comments below.

**Strengths:**

1. The paper is clear and well-written. The proof seems to be solid.
2. The random feature analysis of attention layers is new to this community. The analysis combines the expressive power and the generalization gap, which is better than some existing works.
3. The comparison between attention layers and MLP covers a lot of target functions, which is impressive.

**Weaknesses:**

1. Equation 3 is important for further derivation. From my understanding, Equation 3 indicates that the attention map is fixed for all data. I think attention layers usually have different attention maps for different data. I believe this is a big limitation of this work. Some clarification about "fixed attention" is needed to avoid misunderstanding.

2. An attention map fixed at random initialization makes the attention layer useless and meaningless. I think at least the attention layer should be trainable.

3. About the related works of generalization analysis of Transformers, the references discussed are too old. Here are some recent references. I would like to see a discussion of these works (Some of these works are concurrent works).
[1] Jelassi et al., 2022, "Vision Transformers provably learn spatial structure. "
[2] Li et al., 2023, "A Theoretical Understanding of Shallow Vision Transformers: Learning, Generalization, and Sample Complexity. "
[3] Oymak et al., 2023, "On the Role of Attention in Prompt-tuning. "
[4] Tarzanagh et al., 2023, "Max-Margin Token Selection in Attention Mechanism. "

**Questions:**

1. Is the comparison in Sections 3.3 and 4.2 a comparison between upper bounds (sufficient conditions) for RFA and RFMLP? I think so. If so, this should be mentioned. This is not very rigorous, but I am ok with it because it is difficult to compare an upper bound and a lower bound.

2. What does "permutation invariant of the target function" mean? Why is it needed?

**Limitations:**

There is no negative societal impact of their work.

---

> ### Author Rebuttal · Authors · 2023-08-09
>
> We thank the reviewer for the valuable comments and suggestions.
>
> > Equation 3 is important for further derivation. From my understanding, Equation 3 indicates that the attention map is fixed for all data. I think attention layers usually have different attention maps for different data. I believe this is a big limitation of this work. Some clarification about "fixed attention" is needed to avoid misunderstanding.
>
> Could the reviewer kindly clarify what "fixed for all data" means in the context of Equation 3? We want to make sure we fully understand your perspective to make appropriate clarifications in the paper. As in standard practice, our attention matrices $(Q_m, K_m, V_m)$ are the same for all data points (cf. Equation 1).
>
> > An attention map fixed at random initialization makes the attention layer useless and meaningless. I think at least the attention layer should be trainable.
>
> Our attention layer is still learnable since the value matrices $V_m$ are learnable. We only freeze the inner layer (the query-key matrices). This corresponds to the Random Feature setting, which is widely considered in deep learning theory.
>
> > About the related works of generalization analysis of Transformers, the references discussed are too old. Here are some recent references. I would like to see a discussion of these works (Some of these works are concurrent works). [1] Jelassi et al., 2022, "Vision Transformers provably learn spatial structure. " [2] Li et al., 2023, "A Theoretical Understanding of Shallow Vision Transformers: Learning, Generalization, and Sample Complexity. " [3] Oymak et al., 2023, "On the Role of Attention in Prompt-tuning. " [4] Tarzanagh et al., 2023, "Max-Margin Token Selection in Attention Mechanism. "
>
> We appreciate the reviewer for suggesting these works, and we will make sure to incorporate and discuss these studies in our revised version. A key distinction between these works and ours is that they only focus on a small class of target functions with special properties, whereas our work covers a large generic class of target functions (polynomials of input tokens of any degree).
>
> >Is the comparison in Sections 3.3 and 4.2 a comparison between upper bounds (sufficient conditions) for RFA and RFMLP? I think so. If so, this should be mentioned. This is not very rigorous, but I am ok with it because it is difficult to compare an upper bound and a lower bound.
>
> Our comparisons are indeed between upper bounds. Existing work has also derived lower bounds on the sample complexity of RFMLPs (e.g., learning degree $p$ polynomial by RFMLP requires $\Omega((dN)^p)$ samples, (Ghorbani et al., 2021)), which agrees with the upper bound for RFMLP we used though they apply to a special case with a uniform distributional assumption (input vector uniformly distributed on the sphere).
>
> We will make sure to emphasize this point and add a discussion of the lower bound in our revision.
>
> >What does "permutation invariant of the target function" mean? Why is it needed?
>
> For any function $f(X_1,\ldots,X_n)$ of $n$ tokens, it is permutation invariant if $f(X_{\sigma(1)},\ldots,X_{\sigma(n)}) = f(X_1,\ldots,X_n)$ for any permutation $\sigma:[n]\to[n]$. We consider permutation invariant target functions, as attention layers can only fit these functions (due to the structure of attention heads). We will clarify the meaning of "permutation invariance" in our revision.

---

> > ### Comment · Reviewer_5oTn · 2023-08-10
> > **Thank you for the response**
> >
> > Thank the author for the response.
> >
> > I am sorry for the confusion. It is not fixed attention "for all the data. " What I want to emphasize is the fixed attention. Because $W_m$ is fixed during the training, which means the attention map is fixed. So this question is similar to my second question. Your answer to my second question says $V_m$ is trainable. However, I feel a trainable attention map is essential to learning Transformer. I know this setting can hardly be changed because of the random feature framework. I can consider increasing the score if the author can answer these two alternative questions properly. The main point is that I want to figure out how the random feature analysis is useful for studying the generalization of the Transformer.
> >
> > 1. What does it imply for $x_0$ and $x_i$ if $\sigma(<W_m, x_0x_i^\top>)$ is activated, and what does it imply when it is not activated? The $\sigma(\cdot)$ you use is ReLLU, right? I want to know, although $W_m$ is randomly initialized and fixed, can it characterize some meaningful relationship between the query and key?
> >
> > 2. Can you provide an intuition of the comparison between Transformer and MLP, i.e., what is the reason that Transformer is better than MLP in terms of generalization? I think the logic should be that it comes from the self-attention layer because it is the major difference between these two architectures. However, I cannot see it from the theory in this paper because the self-attention layer seems not important in the analysis.
> >
> > I am overall satisfied with the response to other questions. A larger range of target functions is a good contribution.

---

> > > ### Author Response · Authors · 2023-08-11
> > > **Response to the further questions**
> > >
> > > Thank the reviewer for the prompt response and the clarification to our questions.
> > >
> > > To answer the two alternative questions about random feature attention:
> > > 1. As we chose $\sigma(\cdot)$ to be ReLU, $\sigma(\langle W_m, x_0x_i^\top\rangle)$ is activated iff $x_i^\top W_mx_0>0$, i.e. $x_i$ and $x_0$ has a positive correlation when transformed by $W_m$. A higher correlation yields a higher attention score.
> > >    * In a simpler scenario where $W_m=U_m^\top V_m$ with $U_m,V_m$ being orthogonal matrices, this is equivalent to $(U_Mx_i)^\top V_mx_0 > 0$, i.e. $x_i$ and $x_0$ have a positive correlation when rotated by $U_m$ and $V_m$ respectively.
> > >    * In the general case where $W_m$ is a random matrix with SVD $W_m=U_m^\top D_m V_m$, the condition requires the (diagonally scaled) vectors $D_m^{1/2}U_mx_i$ and $D_m^{1/2}V_mx_0$ to have a positive correlation.
> > >
> > >    Thus, different randomly initialized attention heads $(W_m)_{m\in[M]}$ induce "correlation tests" with different rotations and scalings, which could characterize meaningful similarity relationships between the query $x_0$ and key $x_i$.
> > >
> > > 2. Transformers (in our case single-layer attention models) generalize better than MLPs as the attention models naturally admit a permutation-invariant structure between tokens, i.e. ${\rm Attn}(x_0; x_1, \dots, x_N) = {\rm Attn}(x_0; x_{\sigma(1)},\dots,x_{\sigma(N)})$ where $\sigma:[N]\to[N]$ is any permutation. As a result, the sample complexity of learning with attention models does not scale with the sequence length when learning such permutation-invariant target functions.
> > > Concretely, for fitting target functions of the form Eq. (7) (which are permutation invariant), the sample complexity of attention model only depends polynomially on the individual token dimension $d$ but not the sequence length $N$, whereas the sample complexity of MLPs depend polynomially on $dN$. (See Examples 1-3 for the concrete comparisons.)
> > >
> > > We agree both questions are important for understanding the role of our random-feature attention model in terms of understanding transformers. We will expand on these points in our revision.

---

> > > > ### Comment · Reviewer_5oTn · 2023-08-15
> > > >
> > > > Thank you for the response. Sorry for my late reply.
> > > >
> > > > I am satisfied with your two responses. One thing that needs to be mentioned is that in your new response 2, it is not enough only to say that the difference is a permutation-invariant structure because I think CNN can also handle the permutation of features. A better answer is to incorporate your explanation of Response 1 into Response 2. I mean, there should also be a distinction between CNN and Transformers. **If you have further clarification on this question, please let me know.**
> > > >
> > > > I have increased my score to 5. My feeling about this paper is that although I think random feature analysis may not be a very suitable analytical tool for Transformers (maybe I am wrong), the authors have tried their best to explore the extension to Transformers in this direction. The analysis is complete. This should be notified, and I appreciate this effort. My main concern is that random feature analysis is usually only good for analyzing one-hidden-layer fully connected networks rather than Transformers, from my understanding. After the rebuttal, the new response 1 gives some new insights. I think it makes sense. I also appreciate the authors' effort during the rebuttal.

---

> > > > > ### Author Response · Authors · 2023-08-18
> > > > > **Response**
> > > > >
> > > > > Thank you for your response and increasing the score. We briefly comment on the additional questions as follows.
> > > > >
> > > > > > CNN can also handle the permutation of features... there should also be a distinction between CNN and Transformers.
> > > > >
> > > > > There is indeed a distinction between CNNs and transformers on what permutation invariance functions they fit, even if we take simple "1d" CNNs where the input is a $d$-dimensional vector (instead of a $d\times d$ image). For example, [58] showed that 1d CNN favors to fit function of type
> > > > > $f(x) = \sum_{k \in [d]} g(x_{(k)})$,
> > > > > where $x_{(k)} = (x_k, \ldots, x_{k+q-1})$ is the $k$-th contiguous block of length $q$ within $x$, and $g$ is a low degree polynomial (cf. their equation (CYC-LOC)). Using a certain CNN architecture (in the Neural Tangent Kernel regime), the sample complexity only depends on $q$ instead of $d$.
> > > > >
> > > > > On the other hand, random-feature transformers favor to fit function of type $\{ F(x_0, x_i): \sum_{r, s \ge 0} \langle x_0^{\otimes r} \otimes x_i^{\otimes s}, f_{r, s} \rangle \}$ (cf. our equation (7)).
> > > > >
> > > > > In words, among the class of permutation invariant functions, CNN favors "local" functions without long range dependence, whereas transformers favors correlation functions with long range dependence.
> > > > >
> > > > > [58] Learning with convolution and pooling operations in kernel methods. Theodor Misiakiewicz, Song Mei.
> > > > >
> > > > > > Random feature analysis may not be a very suitable analytical tool for Transformers … My main concern is that random feature analysis is usually only good for analyzing one-hidden-layer fully connected networks rather than Transformers.
> > > > >
> > > > > We agree that RF analysis is quite fruitful when analyzing one-hidden-layer fully connected networks. RF analyses of CNN, RNN, and GNN also provide intuitions on the inductive bias of these architectures. Extending the analysis to transformers is a natural scientific problem, and we agree that certain simplifications of transformers are required in our paper for RF analysis to apply. For our paper, we are just hoping to faithfully present our attempts of this question to the community.

---

> > > > > > ### Comment · Reviewer_5oTn · 2023-08-18
> > > > > >
> > > > > > Thank you for your response.

---

### Author Rebuttal · Authors · 2023-08-09

**Additional Response to All Reviewers**

We thank all reviewers again for their valuable feedback on our work.

Here we would like to highlight our contributions again, which we believe were missed by some reviewers:

- The random features (RF) model is an important and widely-studied model in deep learning theory (e.g; Daniely 2017, Mei & Montanari 2021, and the many references therein), widely studied for fully connected neural network. This paper **initiates the study of random feature attention models and provides a set of end-to-end learning results**, which we believe should be of broad interest to the community and could inspire follow-up works.

- Our Section 4 considers the case where the $W$ matrices in random-feature attention are initialized as identity plus standard Gaussian, going beyond the standard Gaussian initialization. This setting was motivated by the empirical weight patterns of *pretrained* BERT models. We show that such initialization has non-trivial advantages (in sample complexity upper bounds) over standard Gaussian initialization, for learning certain functions depending on the correlation of tokens. This is a new message that **did not have an analog in existing random feature theory** to our best knowledge.

- Our paper introduces several new techniques, such as sharp analyses of the Rademacher complexity for random-feature attention models (Line 188-193), as well as approximating functions of correlations using identity-biased random-feature attention (Line 283-288 & concrete statement in Lemma C.2). We believe these techniques **could be useful for future work on analyzing attention**.

We would be grateful if the reviewers could reconsider the evaluation taking into account our contributions. We will highlight these points more clearly in our revision. Additionally, we appreciate the reviewers for catching the typos, which we will fix in the revision.

---

### Decision · Program_Chairs · 2023-09-21

**Decision:**

Accept (poster)

**Comment:**

This paper explores what a single attention layer learns through random feature analysis. The analysis provides valuable insights and is of interest to the community. The paper has received generally positive reviews from the reviewers, and the rebuttal has addressed most of the concerns. One remaining issue is the use of ReLU instead of softmax. I understand that this choice is primarily driven by the challenges in analyzing the softmax function in a general context, and the authors have referenced a study [73] suggesting that ReLU could be comparable to softmax (although this comparison has yet to be validated across a wider range of models and datasets). One potential improvement, as also highlighted by one of the reviewers, is to conduct softmax experiments within the paper's simulated environment and gather additional data points to support the claim.

[73] K. Shen, J. Guo, X. Tan, S. Tang, R. Wang, and J. Bian. A study on relu and softmax in
496 transformer. arXiv preprint arXiv:2302.06461, 2023.